# Scalable Energy-Based Models via Adversarial Training: Unifying Discrimination and Generation

**Xuwang Yin**[*]
Independent
xuwangyin@gmail.com

**Claire Zhang**
MIT
clairefz@mit.edu

**Julie Steele**
MIT
jssteele@mit.edu

**Nir Shavit**
MIT
shanir@csail.mit.edu

**Tony T. Wang**
MIT
twang6@mit.edu

## ABSTRACT

Simultaneously achieving robust classification and high-fidelity generative modeling within a single framework presents a significant challenge. Hybrid approaches, such as Joint Energy-Based Models (JEM), interpret classifiers as EBMs but are often limited by the instability and poor sample quality inherent in training based on Stochastic Gradient Langevin Dynamics (SGLD). We address these limitations by proposing a novel training framework that integrates adversarial training (AT) principles for both discriminative robustness and stable generative learning. The proposed method introduces three key innovations: (1) the replacement of SGLD-based JEM learning with a stable, AT-based approach that optimizes the energy function through a Binary Cross-Entropy (BCE) loss that discriminates between real data and contrastive samples generated via Projected Gradient Descent (PGD); (2) adversarial training for the discriminative component that enhances classification robustness while implicitly providing the gradient regularization needed for stable EBM training; and (3) a two-stage training strategy that addresses normalization-related instabilities and enables leveraging pretrained robust classifiers, generalizing effectively across architectures. Experiments on CIFAR-10/100 and ImageNet demonstrate that our approach: (1) is the first EBM-based hybrid to scale to high-resolution datasets with high training stability, simultaneously achieving state-of-the-art discriminative and generative performance on ImageNet 256×256; (2) uniquely combines generative quality with adversarial robustness, enabling faithful counterfactual explanations; and (3) functions as a competitive standalone generative model, matching state-of-the-art autoregressive models (VAR-d16) and surpassing strong diffusion baselines (ADM-G, LDM-4-G), while additionally supporting diverse image synthesis tasks and compositional generation within a single model.

## 1 INTRODUCTION

Deep learning models have traditionally been developed with either discriminative or generative objectives in mind, rarely excelling at both simultaneously (Ng and Jordan, 2001; Jebara, 2004; Lasserre et al., 2006; Xie et al., 2016; Grathwohl et al., 2019). Discriminative models are optimized for classification or regression tasks but lack the ability to model data distributions, while generative models can synthesize new data samples but may underperform on downstream classification tasks (Ng and Jordan, 2001; Jebara, 2004). Recent research has explored unifying these approaches through joint discriminative-generative modeling frameworks that aim to combine the predictive power of discriminative approaches with the rich data understanding of generative models (Xie et al., 2016; Lazarow et al., 2017; Jin et al., 2017; Du and Mordatch, 2019; Grathwohl et al., 2019; Chen

---

[*]Correspondence to xuwangyin@gmail.com. Code: https://github.com/xuwangyin/DAT.

et al., 2019; Guo et al., 2023; Deja et al., 2023). Such unification holds the promise of grounding classification decisions in the model's understanding of the data distribution—but realizing this potential requires more than simply combining discriminative and generative objectives.

Among these unification efforts, Energy-Based Models (EBMs) have emerged as a promising framework due to their flexibility and theoretical connections to both paradigms. In particular, Joint Energy-Based Models (JEM) (Grathwohl et al., 2019) demonstrated that standard classifier architectures could be reinterpreted to simultaneously function as EBMs, enabling both high-accuracy classification and reasonable sample generation. However, a critical limitation of JEM and similar approaches is their reliance on Markov Chain Monte Carlo (MCMC) methods such as Stochastic Gradient Langevin Dynamics (SGLD; Welling and Teh 2011) for training the generative component. SGLD-based EBM learning suffers from significant training instabilities, computational inefficiency, and often produces poor-quality samples (Grathwohl et al., 2019; Duvenaud et al., 2021; Du and Mordatch, 2019; Nijkamp et al., 2019), limiting the practical adoption of these hybrid models.

We address these limitations by introducing **Dual Adversarial Training (DAT)**, a novel framework that leverages adversarial training (AT) principles for both discriminative robustness and stable generative learning within a unified JEM-based architecture. Our approach employs a dual application of adversarial training: (1) standard AT for the discriminative component to achieve robustness against adversarial perturbations, and (2) an AT-based energy function learning strategy for the generative component that replaces unstable SGLD-based JEM learning.

Our key technical contributions include:

1. **A stable AT-based alternative to SGLD-based JEM learning.** We replace the unstable SGLD-based JEM learning with an adversarial training approach that optimizes the energy function through a Binary Cross-Entropy loss that discriminates between real data and contrastive samples generated via PGD (Madry et al., 2017). This addresses the training instabilities of JEM, enabling reliable convergence and significantly improved sample quality.

2. **Robustness and implicit regularization from adversarial training.** We incorporate adversarial training for the discriminative component, which not only enhances classification robustness but also eliminates the need for the explicit $R_1$ gradient penalty (Mescheder et al., 2018) required by previous AT-EBM frameworks (Yin et al., 2022), simplifying the training procedure and avoiding constraints on model expressiveness.

3. **Two-stage training strategy.** We introduce a two-stage training strategy that leverages pre-trained robust classifiers and addresses normalization-related instabilities, generalizing across architectures with batch normalization (ResNet) and layer normalization (ConvNeXt).

Experiments on CIFAR-10/100 and ImageNet demonstrate the effectiveness and scalability of our approach, establishing three advances in hybrid modeling:

1. **First EBM-based hybrid that scales to high-resolution complex datasets.** Prior EBM-based hybrid approaches could not scale beyond low resolution or achieve competitive generative performance on ImageNet-level datasets due to SGLD instability. Our approach is the first to overcome these limitations, achieving competitive generative quality and strong classification performance on ImageNet 256×256 with high training stability, demonstrating that EBM-based hybrid models can scale reliably to complex, high-resolution datasets.

2. **Generative capability with adversarial robustness enables counterfactual explanations.** Our approach uniquely combines state-of-the-art generative quality with adversarial robustness, enabling the model to generate visual counterfactual explanations using the exact same energy function that determines its classification decisions. We demonstrate that our counterfactuals are substantially more perceptually realistic and semantically faithful to target class than those from non-robust or robustness-only methods.

3. **Competitive and flexible generative model.** When evaluated on generation quality, our approach (with ConvNeXt-Large) matches the autoregressive model VAR-d16 and surpasses diffusion models on ImageNet 256×256, while achieving higher throughput than diffusion models. Beyond this competitive quality, our approach is versatile across diverse image synthesis tasks (Santurkar et al., 2019) and uniquely supports compositional generation (Du et al., 2020).

These results show that adversarial training provides an effective and scalable foundation for energy-based generative learning, enabling hybrid models that need not compromise on any dimension.

## 2 RELATED WORK

**Joint discriminative-generative modeling** The pursuit of joint discriminative-generative modeling, or hybrid modeling, aims to combine the predictive power of discriminative approaches with the rich data understanding of generative models within a single framework. This line of research is motivated by the potential to improve classifier robustness, calibration, and out-of-distribution detection (Grathwohl et al., 2019; Du and Mordatch, 2019), while also enabling tasks like sample generation (e.g., for counterfactual explanation (Deja et al., 2023)) and semi-supervised learning (Kingma et al., 2014). A significant thrust in this area involves Energy-Based Models (EBMs) (LeCun et al., 2006). Early work by Xie et al. (2016) showed how generative ConvNets could be derived from discriminative ones, framing them as EBMs. Du and Mordatch (2019) scaled EBM training to complex high-dimensional image datasets and showed that the same energy function can be used for discriminative tasks such as out-of-distribution detection and robust classification, without introducing task-specific objectives. Grathwohl et al. (2019) introduced JEM, which explicitly reinterprets standard classifiers as EBMs over the joint distribution of data and labels $p(x, y)$, allowing simultaneous classification and generation. Yang et al. (2023) incorporated sharpness-aware minimization (SAM) to smooth energy landscapes and removed data augmentation from the EBM loss term to improve both classification accuracy and generation quality of JEM. Guo et al. (2023) proposed EGC, which employs Fisher divergence within a diffusion framework to learn an unconditional score function $\nabla \log p(x)$ and a conditional classifier $p(y|x)$ for unified classification and generation, thereby circumventing the training instability and scalability limitations of traditional EBMs.

Alternative architectural approaches have also been explored for joint modeling. Rather than energy-based formulations, joint diffusion models (Deja et al., 2023) attach classifiers directly to diffusion model UNet encoders for joint end-to-end training. Another distinct approach is "introspective learning," where a single model functions as both a generator and a discriminator through an iterative self-evaluation process, developed across works by Lazarow et al. (2017), Jin et al. (2017), and Lee et al. (2018). Flow-based models have also been explored for hybrid tasks; for instance, Residual Flows (Chen et al., 2019) utilized invertible ResNet and showed competitive performance in joint generative and discriminative settings. These diverse approaches underscore the continued effort to create models that jointly leverage both discriminative and generative learning.

**Joint Energy-Based Models (JEM)** Grathwohl et al. (2019) showed that the logits of a standard classifier can be reinterpreted as defining an energy function over the joint distribution $p(x, y)$, enabling simultaneous classification and generation within a single architecture. Their hybrid training objective combines cross-entropy for $p(y|x)$ with an SGLD-based EBM objective for $p(x)$, improving calibration, OOD detection, and adversarial robustness over standard training. Subsequent works improved JEM's training stability: JEM++ (Yang and Ji, 2021) introduced proximal SGLD and informative initialization, while Robust-JEM (Korst and Asadulaev, 2022) incorporated adversarial training into the discriminative component. Yang et al. (2023) applied sharpness-aware minimization to smooth energy landscapes and decoupled data augmentation from the EBM loss. However, all these methods fundamentally rely on SGLD-based sampling for the generative component, inheriting its instability, and remain limited to CIFAR-scale ($32 \times 32$) datasets.

**Adversarial training and energy-based models** Several works have revealed deep connections between adversarial robustness and energy-based modeling. Zhu et al. (2021) showed that adversarial training implicitly flattens the energy landscape around real data, and proposed JEAT for joint classification and generation. Wang et al. (2022a) unified adversarial training and contrastive learning under an EBM framework, showing that PGD-generated adversarial examples serve as implicit negative samples (see also Mirza et al., 2024). Separately, Yin et al. (2022) proposed AT-EBM, which replaces SGLD with PGD-based contrastive sampling and a BCE loss for learning the energy function, achieving more stable training and competitive generation, though limited to unconditional generation with an explicit $R_1$ gradient penalty. Augustin et al. (2020) proposed RATIO, combining in-distribution AT with OOD adversarial training for robust confidence calibration and OOD detection, also enabling visual counterfactuals through $\ell_2$ robustness. Santurkar et al. (2019) demonstrated that robust classifiers can serve as primitives for diverse image synthesis tasks via gradient-based optimization. Our work synthesizes these lines of research by incorporating AT-based EBM learning into the JEM framework for conditional generative modeling, with implicit $R_1$ regularization from adversarial training eliminating the need for explicit gradient penalties. See Appendix A.1 for extended discussion.

## 3 METHOD

### 3.1 JOINT ENERGY-BASED MODEL

Our approach builds upon the JEM framework (Grathwohl et al., 2019), which reinterprets the outputs of a standard discriminative classifier as an energy-based model (EBM) over the joint distribution of data $x$ and labels $y$. Given a classifier network that produces logits $f_\theta(x) \in \mathbb{R}^K$ for $K$ classes, JEM defines the joint energy function as:

$$E_\theta(x, y) = -f_\theta(x)[y] \tag{1}$$

where $f_\theta(x)[y]$ is the logit corresponding to class $y$. This energy function can be normalized to obtain a joint probability density:

$$p_\theta(x, y) = \frac{\exp(-E_\theta(x, y))}{Z(\theta)} = \frac{\exp(f_\theta(x)[y])}{Z(\theta)} \tag{2}$$

where $Z(\theta) = \sum_{y'} \int \exp(f_\theta(x')[y'])dx'$ is the partition function (an intractable global normalizing constant). By marginalizing out the label $y$, a marginal density over the input data $x$ can be obtained:

$$p_\theta(x) = \sum_y p_\theta(x, y) = \frac{\sum_y \exp(f_\theta(x)[y])}{Z(\theta)} \tag{3}$$

Thus, a valid energy function for $p_\theta(x)$ is given by:

$$E_\theta(x) = -\log \sum_y \exp(f_\theta(x)[y]) \tag{4}$$

A JEM is trained by maximizing the joint log-likelihood $\log p_\theta(x, y)$ over labeled training datapoints $(x, y)$ drawn from an empirical joint distribution $p_{\text{data}}(x, y)$. The joint log-likelihood is typically factorized as $\log p_\theta(y|x) + \log p_\theta(x)$. The conditional term $\log p_\theta(y|x)$ can be maximized by minimizing the standard cross-entropy classification loss. The marginal term $\log p_\theta(x)$ is optimized using the EBM gradient (LeCun et al., 2006):

$$\nabla_\theta \mathbb{E}_{x \sim p_{\text{data}}(x)}[\log p_\theta(x)] = \mathbb{E}_{x \sim p_{\text{data}}(x)}[-\nabla_\theta E_\theta(x)] - \mathbb{E}_{x \sim p_\theta(x)}[-\nabla_\theta E_\theta(x)] \tag{5}$$

where $p_{\text{data}}(x)$ is the empirical marginal distribution obtained by marginalizing $y$ from $p_{\text{data}}(x, y)$. This gradient decreases the energy of real data samples while increasing the energy of model-generated samples. At equilibrium when $p_\theta(x) = p_{\text{data}}(x)$, these terms balance and the gradient becomes zero.

To approximate the expectation $\mathbb{E}_{x \sim p_\theta(x)}[\cdot]$, samples are drawn from $p_\theta(x)$ using SGLD (Welling and Teh, 2011), which starts from an initial distribution $p_0(x)$ (e.g., uniform noise) and iteratively applies the update rule:

$$x_{t+1} = x_t - \frac{\alpha}{2}\nabla_x E_\theta(x_t) + \xi_t, \quad \text{where } \xi_t \sim \mathcal{N}(0, \alpha) \tag{6}$$

### 3.2 LEARNING JEM WITH ADVERSARIAL TRAINING

The JEM framework successfully integrates generative modeling into classifiers, but its reliance on SGLD and EBM gradient (Eq. 5) causes significant training instabilities (Grathwohl et al., 2019; Duvenaud et al., 2021) and results in poor sample quality. We address these limitations by replacing the SGLD-based JEM with an adversarial training (AT) approach inspired by AT-EBM (Yin et al., 2022).

Concretely, we replace the standard EBM gradient (Eq. 5) with a stabilized formulation:

$$\mathbb{E}_{x \sim p_{\text{data}}(x)}[-\nabla_\theta E_\theta(x)] - \mathbb{E}_{x \sim p_\theta(x)}[-\nabla_\theta E_\theta(x)]$$
$$\implies \mathbb{E}_{x \sim p_{\text{data}}(x)}[-\alpha(x)\nabla_\theta E_\theta(x)] - \mathbb{E}_{x \sim p_\theta(x)}[-\beta(x)\nabla_\theta E_\theta(x)] \tag{7}$$

where $\alpha(x) = 1 - \sigma(-E_\theta(x))$ and $\beta(x) = \sigma(-E_\theta(x))$ are data-dependent scaling factors, and $\sigma$ denotes the logistic sigmoid function. This formulation preserves the structural form of Eq. 5

while introducing adaptive scaling factors that modulate gradient contributions according to the model's current energy values. These scaling factors stabilize training by providing automatic gradient regularization (Yin et al., 2022): when $-E_\theta(x)$ takes extreme values, the sigmoid saturation drives the corresponding scaling factor ($\alpha$ for $p_{\text{data}}$ samples, $\beta$ for contrastive samples) toward zero, attenuating gradient contributions and preventing numerical overflow and underflow. In contrast, the standard EBM gradient (Eq. 5) is unconstrained and permits $-E_\theta(x)$ to grow unbounded, resulting in numerical instability. The corresponding training objective, whose gradient with respect to $\theta$ recovers Eq. 7, can be written as a Binary Cross-Entropy (BCE) loss:

$$\mathcal{L}_{\text{BCE}}(\theta) = -\mathbb{E}_{x \sim p_{\text{data}}(x)}[\log(\sigma(-E_\theta(x)))] - \mathbb{E}_{x \sim p_\theta(x)}[\log(1 - \sigma(-E_\theta(x)))] \tag{8}$$

This gradient formulation stabilizes training at the cost of limiting the EBM to modeling the support of $p_{\text{data}}$ rather than learning the full density. We provide a formal characterization of the learned distribution in Section A.2, where we show that the optimal solution under the joint discriminative-generative objective learns $f_\theta^*(x)[y] = \log p_{\text{data}}(y|x)$ on the support with constant marginal energy $E_\theta^*(x) = 0$.

In addition to the above gradient reformulation, we follow Yin et al. (2022) in replacing JEM's SGLD sampling with PGD, initializing from an auxiliary out-of-distribution dataset $p_{\text{ood}}$ (e.g., the 80 Million Tiny Images dataset (Torralba et al., 2008) for CIFAR-10). Concretely, contrastive samples are generated by performing $T$ steps of normalized gradient descent on $E_\theta(x)$:

$$x_{t+1} = x_t - \eta \frac{\nabla_x E_\theta(x_t)}{||\nabla_x E_\theta(x_t)||_2}, \quad t = 0, 1, \ldots, T-1 \tag{9}$$

The PGD procedure transforms OOD images toward the data distribution during training. At test time, the same mechanism produces samples with competitive FID scores (Section 4.3) and enables counterfactual generation from existing images (Section 4.3.2). While OOD initialization yields the best generation quality, we show that DAT can also be trained from pure random noise, eliminating the dependence on auxiliary datasets (Appendix C.5).

### 3.3 CLASSIFIER ROBUSTNESS AND IMPLICIT REGULARIZATION

**Classifier robustness.** While our AT-based approach improves the generative capabilities of JEM, the discriminative component still exhibits weaker adversarial robustness compared to standard AT classifiers. To address this limitation, we complement the generative improvements by incorporating adversarial training for the discriminative term $p_\theta(y|x)$.

For each input sample $x$ with label $y$, we find an adversarial example $x_{adv}$ within an $\epsilon$-ball $B(x, \epsilon)$ around $x$ that maximizes the classification loss:

$$x_{adv} = \underset{x' \in B(x,\epsilon)}{\arg\max} \mathcal{L}_{\text{CE}}(\theta; x', y) \tag{10}$$

where $\mathcal{L}_{\text{CE}}(\theta; x', y)$ is the standard cross-entropy loss and $B(x, \epsilon)$ is an $\ell_p$-norm ball. Similar to our generative component, we approximate this optimization using the PGD attack, generating adversarial examples through iterative gradient steps within the constraint set. The classification term is then defined as:

$$\mathcal{L}_{\text{AT-CE}}(\theta) = \mathbb{E}_{(x,y) \sim p_{\text{data}}(x,y)} [-\log p_\theta(y|x_{adv})] \tag{11}$$

**Implicit regularization.** Incorporating AT for the classifier not only ensures robust accuracy but also yields an additional benefit for the generative component. The original AT-EBM framework required explicit $R_1$ gradient penalties (Mescheder et al., 2018) for training stability; we find that AT eliminates this need. Building on Roth et al. (2020), we show that AT implicitly bounds the $R_1$ penalty (Section A.3). We empirically validate the effect of AT on $R_1$ gradients: AT maintains bounded $R_1$ gradients throughout training, while standard training exhibits gradient explosion (Figure 2).

### 3.4 DUAL AT FOR JOINT MODELING

Our complete approach applies adversarial training to both the generative and discriminative components, resulting in the combined objective:

$$\mathcal{L}(\theta) = \mathcal{L}_{\text{AT-CE}}(\theta) + \mathcal{L}_{\text{BCE}}(\theta) \tag{12}$$

where $\mathcal{L}_{\text{AT-CE}}(\theta)$ is the robust classification loss from Eq. 11, and $\mathcal{L}_{\text{BCE}}(\theta)$ is the AT-based generative loss from Eq. 8. This combination corresponds to the factorization of the joint log-likelihood in the original JEM formulation: $\log p_\theta(x, y) = \log p_\theta(y|x) + \log p_\theta(x)$. The joint objective simultaneously enhances the model's discriminative robustness and generative capabilities, addressing the key limitations of the original JEM framework; full algorithmic details are provided in Appendix A.4.

Our approach shares conceptual similarities with RATIO (Augustin et al., 2020), which also combines adversarially robust classification with adversarial perturbations applied to out-of-distribution data:

$$\mathcal{L}_{\text{RATIO}}(\theta) = \mathcal{L}_{\text{AT-CE}}(\theta) + \lambda \mathbb{E}_{x \sim p_{\text{ood}}(x)} \left[ \max_{x' \in B(x, \epsilon_o)} \mathcal{L}_{\text{CE}}(\theta; x', \mathbf{1}/K) \right] \tag{13}$$

Despite this structural similarity, the approaches differ fundamentally in their objectives. RATIO's secondary term attacks OOD samples to maximize classifier confidence, then penalizes this confidence via cross-entropy against a uniform distribution, explicitly targeting robust OOD detection. In contrast, our $\mathcal{L}_{\text{BCE}}(\theta)$ leverages AT-based energy function learning (Yin et al., 2022), using PGD to generate contrastive samples from OOD data and employing BCE loss to shape the energy landscape. While RATIO focuses primarily on reducing confidence in OOD regions, our approach prioritizes learning an energy function that enables high-quality sample generation alongside robust classification.

## 3.5 TWO-STAGE TRAINING

Neural network architectures typically incorporate normalization layers to stabilize and speed up training: ResNet (He et al., 2016) uses batch normalization (BN) (Ioffe and Szegedy, 2015), while modern architectures like ConvNeXt (Liu et al., 2022) and Vision Transformers (Dosovitskiy et al., 2021; Vaswani et al., 2017) use layer normalization (Ba et al., 2016). Training energy-based joint models presents challenges with normalization layers. In particular, batch normalization has been identified as problematic for EBM training (Grathwohl et al., 2019; Yin et al., 2022; Zhao et al., 2020). Consistent with these findings, we observe that enabling BN during joint training destabilizes the optimization of the generative modeling term $\mathcal{L}_{\text{BCE}}$, leading to oscillating losses and failure to converge.

To address these challenges while maintaining the benefits of normalization during discriminative training, we propose a two-stage training strategy that generalizes effectively across architectures:

- **Stage 1: Discriminative training.** We first train the network with its original normalization configuration, optimizing only the robust classification objective $\mathcal{L}_{\text{AT-CE}}$ (Eq. 11). This stage is equivalent to standard adversarial training and leverages normalization layers to achieve faster convergence and strong robust classification performance. Notably, this stage can be skipped when pretrained robust classifiers are available, making our approach immediately applicable to existing robust models.

- **Stage 2: Joint training.** After robust discriminative training, we modify the normalization behavior when necessary and continue training with the complete objective $\mathcal{L}(\theta) = \mathcal{L}_{\text{AT-CE}}(\theta) + \mathcal{L}_{\text{BCE}}(\theta)$ (Eq. 12). For architectures with batch normalization (ResNet, WRN), we disable BN by setting BN modules to eval mode, which freezes the BN statistics computed during Stage 1. For architectures with layer normalization (ConvNeXt), we maintain the normalization as-is.

This strategy not only addresses the incompatibility between batch normalization and EBM training, but also enables leveraging pretrained robust classifiers to reduce training costs (see Appendix B.2 for detailed computational analysis). As demonstrated in Section 4.3, Stage 2 improves the generative modeling performance of pretrained robust classifiers with minimal impact on the robust accuracy established in Stage 1 (see Appendix B.3 for training dynamics). The two-stage training strategy works effectively for both ResNet and ConvNeXt models, making it applicable to modern scalable architectures such as Vision Transformers (Dosovitskiy et al., 2021; Singh et al., 2023; Peebles and Xie, 2023a).

## 4 EXPERIMENTS

### 4.1 TRAINING SETUP

**Datasets and architectures.** We evaluate our approach on CIFAR-10, CIFAR-100 (Krizhevsky et al., 2009), and ImageNet (Deng et al., 2009). For CIFAR-10/100 experiments, we use WRN-34-10 (Zagoruyko and Komodakis, 2016) following the official RATIO implementation (Augustin et al., 2020). For ImageNet experiments, we use ResNet-50 (He et al., 2016), WRN-50-4 (Zagoruyko and Komodakis, 2016), and ConvNeXt-Large with ConvStem (Singh et al., 2023).

**Two-stage training.** Since Stage 1 training is equivalent to standard adversarial training, we use pretrained standard AT checkpoints when available: a standard AT checkpoint from the RATIO codebase (Augustin et al., 2020) for CIFAR-10, pretrained ImageNet ResNet-50 and WRN-50-4 models (Salman et al., 2020), and pretrained ConvNeXt-Large with ConvStem (Singh et al., 2023) (originally trained for $\ell_\infty = 4/255$ robustness), while training our own CIFAR-100 model following Augustin et al. (2020). For Stage 2 training, we initialize from the Stage 1 model and continue joint training. For ResNet and WRN architectures, we set the BN modules to eval mode (which disables BN while preserving the BN statistics computed during Stage 1). Complete training hyperparameters can be found in Appendix B.1.1.

**Data augmentation.** Strong data augmentations are necessary for classifier robustness (Rebuffi et al., 2021; Gowal et al., 2020) but can distort the data distribution in ways detrimental to generative modeling. Following Yang et al. (2023), we use separate augmentation strategies for Stage 2 training: strong augmentations for $\mathcal{L}_{\text{AT-CE}}$ and basic transformations for $\mathcal{L}_{\text{BCE}}$. Yang et al. (2023) found that augmentations such as random cropping with padding introduce artifacts (e.g., black borders) in generated samples and therefore excluded them from generative training. We observe that this is not a limitation in our framework—even with random cropping and padding applied, our generated samples do not exhibit such artifacts (Appendix B.1.2).

**Out-of-distribution data.** Following RATIO (Augustin et al., 2020), we use the 80 million tiny images (Torralba et al., 2008) as the OOD dataset ($p_{\text{ood}}$) for CIFAR-10/100 experiments. For ImageNet, as there are no established OOD datasets, we follow OpenImage-O (Wang et al., 2022b) and construct an OOD dataset from the Open Images training set (Krasin et al., 2016). We randomly sample 350K images, restricting our selection to those whose labels do not overlap with any ImageNet classes, yielding 300K samples for training and 50K for FID evaluation.

### 4.2 EVALUATION METRICS

We measure both classification and generative modeling performance. For classification, we report clean accuracy and robust accuracy against $\ell_2$ attacks ($\epsilon = 0.5$ for CIFAR-10/100 and $\epsilon = 3.0$ for ImageNet) computed using AutoAttack (Croce and Hein, 2020). For generative modeling, we evaluate sample diversity and visual fidelity using Fréchet Inception Distance (FID) (Heusel et al., 2017) and Inception Score (IS) (Salimans et al., 2016). We focus on conditional generation, which consistently outperforms unconditional generation across all datasets (Table 9); details of the generation setup are provided in Appendix B.4.

To measure counterfactual quality, we apply targeted PGD attacks to training samples across a range of perturbation budgets and compute class-wise FID between the resulting counterfactuals and real samples of each target class. This differs from generative sample evaluation, where PGD is applied to OOD inputs rather than in-distribution data.

### 4.3 RESULTS

We evaluate DAT on CIFAR-10, CIFAR-100, and ImageNet 256×256 (Tables 1 and 2). As detailed below, our approach achieves the first successful scaling of EBM-based hybrids to high resolutions while simultaneously achieving robust classification and high-fidelity generation.

**First EBM-based hybrid to scale to high-resolution datasets with adversarial robustness.** Prior EBM-based hybrid approaches (JEM, SADA-JEM) are not explicitly optimized for adversarial robustness. On CIFAR-10, these methods achieve significantly lower robust accuracy than standard AT: JEM achieves 40.5% and SADA-JEM achieves 31.93%, compared to 75.73% for standard AT

(Table 1). Our approach addresses this limitation, achieving 75.75% robust accuracy—comparable to standard AT—while improving generative quality over prior EBM hybrids: FID 9.12 versus 38.4 (JEM) and 9.41 (SADA-JEM). Beyond robustness, prior EBM-based hybrids could not scale beyond low resolution or achieve competitive generative performance on ImageNet-level datasets. Our approach is the first to overcome this limitation: on ImageNet 256×256, our ConvNeXt-Large model achieves FID 3.29 (Table 2) with classification performance comparable to standard AT (Appendix C.6), demonstrating that EBM-based hybrid models can scale reliably to complex, high-resolution datasets.

**Unique combination of generation quality and adversarial robustness.** While other scalable hybrids exist, they do not achieve both strong adversarial robustness and state-of-the-art generative quality. The diffusion-based EGC achieves 13.56% robust accuracy on ImageNet compared to our 56.40%, with worse FID (6.05 vs. 3.29) (Table 2). Similarly, RATIO targets robustness but not generation quality (FID 21.96 vs. our 9.12 on CIFAR-10). Qualitatively, Figures 6, 7, and 8 show that our method produces visually superior samples with fewer artifacts compared to RATIO and standard AT. This unique combination of generative quality and robustness enables our model to produce substantially higher-quality counterfactual explanations than both non-robust and robustness-only models (Section 4.3.2).

**Competitive as a standalone generative model.** When evaluated purely on generation quality, our approach achieves performance competitive with state-of-the-art specialized generative models. On ImageNet 256×256, DAT with ConvNeXt-L achieves FID 3.29, matching the state-of-the-art autoregressive model VAR-d16 (FID 3.30) while using fewer parameters (198M vs. 310M) and outperforming leading diffusion models including ADM-G (FID 4.59, 608M parameters) and LDM-4-G (FID 3.60, 400M parameters) (Table 2). The model also achieves relatively strong IS performance (310.2), likely due to PGD-based sampling explicitly optimizing for classifier confidence. Figure 9 shows representative samples demonstrating the visual quality achieved by our approach. Beyond quality, our approach achieves significantly higher throughput than diffusion models: $\sim29\times$ faster than ADM-G and $\sim5\times$ faster than LDM-4-G (Table 7).

### 4.3.1 ANALYSIS AND ABLATIONS

**Noise initialization.** DAT can be trained without any OOD data by initializing PGD from pure random noise, eliminating the dependence on auxiliary datasets. Noise-initialized DAT matches OOD initialization on both clean and robust accuracy while achieving reasonable generation quality (Appendix C.5).

**Generative-discriminative trade-off.** Our experiments reveal that the number of PGD iterations $T$ (Eq. 9) controls the balance between discriminative and generative performance. On CIFAR-10, increasing $T$ from 40 to 50 improves FID from 9.12 to 7.57 at the cost of standard and robust accuracy. A similar trend is observed across other datasets. In Appendix C.8.3, we investigate this tension, showing that the generative objective aligns representations with $p_{\text{data}}$, which can come at the cost of robustness. We further show that this trade-off can be explicitly tuned through loss weighting in addition to PGD iterations.

**Effect of model capacity and architecture.** Our experiments on ImageNet demonstrate the benefits of increased model capacity and modern architectures. Scaling from ResNet-50 (26M parameters) to WRN-50-4 (223M parameters) yields consistent improvements across both discriminative and generative metrics. Beyond capacity, using state-of-the-art architectures also provides clear benefits: ConvNeXt-L (198M parameters) substantially outperforms WRN-50-4 (223M parameters) in both accuracy and generation quality despite having fewer parameters, demonstrating the importance of architectural design alongside model scale.

**Component contributions.** We isolate the contributions of our key components, showing that the generative loss and decoupled augmentation are essential for high-fidelity synthesis (Appendix C.8.1).

**OOD data efficiency.** We also demonstrate high data efficiency, achieving strong performance even with limited auxiliary OOD data (Appendix C.8.2).

**Robustness and stability.** Beyond adversarial robustness, DAT maintains corruption robustness comparable to standard AT (Appendix C.7) and generalizes to $\ell_\infty$ training (Appendix C.6). We also demonstrate high reproducibility with zero training divergence across all runs (Appendix C.9).

**Computational efficiency.** Finally, we analyze computational cost, showing that our two-stage training incurs only modest overhead (1.05–1.56×) relative to standard AT, and that our model achieves significantly higher inference throughput than diffusion models (Appendix B.2).

Table 1: Classification and generative modeling results on CIFAR-10 and CIFAR-100.

| Method | Acc% ↑ | Robust Acc% ↑ | IS ↑ | FID ↓ |
|---|---|---|---|---|
| *CIFAR-10 hybrid models* | | | | |
| Residual Flow (Chen et al., 2019) | 70.3 | – | 3.6 | 46.4 |
| Glow (Kingma and Dhariwal, 2018) | 67.6 | – | 3.92 | 48.9 |
| IGEBM (Du and Mordatch, 2019) | 49.1 | – | 8.3 | 37.9 |
| JEM (Grathwohl et al., 2019) | 92.9 | 40.5 | 8.76 | 38.4 |
| VERA (Grathwohl et al., 2021) | 93.2 | – | 8.11 | 30.5 |
| JEM++ (Yang and Ji, 2021) | 94.1 | – | 8.11 | 38.0 |
| JEAT (Zhu et al., 2021) | 85.16 | – | 8.80 | 38.24 |
| Robust-JEM (Korst and Asadulaev, 2022) | – | – | 8.71 | 41.17 |
| SADA-JEM (Yang et al., 2023) | 95.5 | 31.93 | 8.77 | 9.41 |
| WEAT (Mirza et al., 2024) | 83.36 | – | 8.97 | 30.74 |
| EGC (Guo et al., 2023) | 95.9 | – | 9.43 | 3.30 |
| Joint-Diffusion (Deja et al., 2023) | 96.4 | – | – | 6.4 |
| RATIO (Augustin et al., 2020) | 92.23 | 76.25 | 9.61 | 21.96 |
| Standard AT (Augustin et al., 2020) | 92.43 | 75.73 | 9.58 | 28.41 |
| DAT ($T = 40$) | 91.92 | **75.75** | 9.92 | **9.12** |
| DAT ($T = 50$) | 90.72 | **74.65** | 9.86 | **7.57** |
| *CIFAR-10 conditional generative models* | | | | |
| SNGAN (Miyato et al., 2018) | – | – | 8.59 | 25.5 |
| BigGAN (Brock et al., 2018) | – | – | 9.22 | 14.73 |
| StyleGAN2 (Karras et al., 2020a) | – | – | 9.53 | 6.96 |
| StyleGAN2 ADA (Karras et al., 2020b) | – | – | 10.24 | 3.49 |
| EDM (Karras et al., 2022) | – | – | – | 1.79 |
| *CIFAR-100 hybrid models* | | | | |
| Joint-Diffusion (Deja et al., 2023) | 77.6 | – | – | 16.8 |
| SADA-JEM (Yang et al., 2023) | 75.0 | – | 11.63 | 14.4 |
| EGC (Guo et al., 2023) | 77.9 | – | 11.50 | 4.88 |
| RATIO (Augustin et al., 2020) | 71.58 | 47.74 | 9.28 | 24.17 |
| Standard AT (Augustin et al., 2020) | 72.16 | 47.78 | 9.54 | 23.59 |
| DAT ($T = 45$) | 65.76 | **45.94** | 10.99 | **10.73** |
| DAT ($T = 50$) | 60.12 | **42.55** | 11.12 | **9.53** |

Table 2: Classification and generative modeling results on ImageNet 256×256.

| Method | Acc% ↑ | Robust Acc% ↑ | FID ↓ | IS ↑ | Params | Steps |
|---|---|---|---|---|---|---|
| *Hybrid models* | | | | | | |
| EGC (Guo et al., 2023) | 78.90 | 13.56 | 6.05 | 231.3 | 543M (U-Net) | 1000 |
| Standard AT (Salman et al., 2020) | 64.91 | 39.96 | 15.12 | 286.2 | 26M (ResNet-50) | 13 |
| DAT ($T = 15$) | 61.31 | 39.96 | 6.87 | 322.65 | 26M (ResNet-50) | 14 |
| DAT ($T = 30$) | 55.96 | 37.14 | 5.28 | 319.3 | 26M (ResNet-50) | 14 |
| Standard AT (Salman et al., 2020) | 71.25 | 45.86 | 37.33 | 260.2 | 223M (WRN-50-4) | 12 |
| DAT ($T = 30$) | 64.45 | 45.84 | 6.23 | 341.0 | 223M (WRN-50-4) | 17 |
| DAT ($T = 65$) | 58.78 | 40.74 | 4.94 | 358.0 | 223M (WRN-50-4) | 19 |
| Standard AT (Singh et al., 2023) | 78.25 | 33.38 | 44.46 | 27.32 | 198M (ConvNeXt-L-CvSt) | 0 |
| DAT ($T = 110$) | 75.78 | 56.40 | 3.29 | 310.2 | 198M (ConvNeXt-L-CvSt) | 36 |
| *Conditional generative models* | | | | | | |
| BigGAN-deep (Brock et al., 2018) | – | – | 6.95 | 203.6 | 340M (ResNet) | 1 |
| ADM-G (Dhariwal and Nichol, 2021) | – | – | 4.59 | 186.7 | 608M (U-Net) | 250 |
| LDM-4-G (Rombach et al., 2022) | – | – | 3.60 | 247.7 | 400M (U-Net) | 250 |
| DiT-XL/2-G (Peebles and Xie, 2023b) | – | – | 2.27 | 278.2 | 675M (Transformers) | 250 |
| VAR-d16 (Tian et al., 2024) | – | – | 3.30 | 274.4 | 310M (Transformers) | 10 |
| VAR-d30-re (Tian et al., 2024) | – | – | 1.73 | 350.2 | 2.0B (Transformers) | 10 |

### 4.3.2 COUNTERFACTUAL GENERATION, OOD DETECTION, AND CALIBRATION

**Counterfactual generation.** Visual counterfactual explanations (VCEs) reveal what minimal semantic changes would flip a classifier's decision. Standard classifiers cannot generate meaningful VCEs because their input gradients lack semantic structure—gradient-based modifications produce adversarial noise rather than interpretable features. Prior VCE methods were therefore restricted to adversarially robust models (Augustin et al., 2020; Boreiko et al., 2022). DVCE (Augustin et al., 2022) overcomes this limitation by guiding an unconditional diffusion model using gradients from an auxiliary adversarially robust classifier, projecting these onto a cone around the target classifier's gradient. This enables VCEs for arbitrary classifiers but requires multiple external dependencies: a pretrained diffusion model, a robust classifier, and careful hyperparameter tuning.

In contrast, DAT is *intrinsically explainable*: it generates VCEs directly by gradient descent on the joint energy $E_\theta(x, y)$, requiring no external models. Because our joint objective explicitly learns the energy landscape of the joint distribution $p(x, y)$, the resulting gradients $\nabla_x E_\theta(x, y)$ naturally point toward semantically valid configurations. Therefore, our model's improved generative capability directly translates to higher-quality counterfactual explanations.

Figure 1 compares counterfactual quality across different models while accounting for classifier confidence. Our approach consistently generates counterfactuals with lower FIDs than baseline methods when achieving similar target class confidence. For instance, when RATIO reaches 0.89 confidence in the target class (at $\epsilon = 8$), its corresponding FID is 43.18. Our DAT model achieves a similar confidence level at $\epsilon = 4$ with a significantly better FID of 25.53. This demonstrates that, for a comparable level of certainty that the counterfactual represents the target class, our generated samples are substantially more faithful to the true visual characteristics of that class, indicating more plausible counterfactuals. We provide visualizations of counterfactuals in Appendix C.2.

**OOD detection.** Our approach generally underperforms RATIO on OOD detection. Ablation studies show this gap persists even when using identical aggressive augmentation for both the generative and discriminative components, indicating it stems from fundamental objective differences rather than the use of milder augmentation for the generative term: RATIO explicitly optimizes for OOD detection while our generative loss prioritizes learning accurate energy functions for generation. The complete details can be found in Appendix C.3.

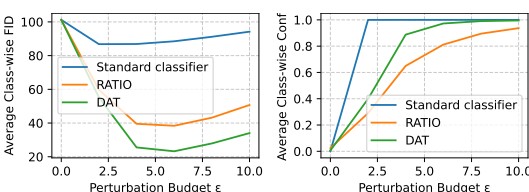

Figure 1: Counterfactual FIDs and classifier confidences under different perturbations.

**Calibration.** Our model's calibration performance is dataset-dependent, with detailed results provided in Appendix C.4. While the model is well-calibrated on CIFAR-10, outperforming the standard AT and RATIO baselines, it exhibits higher overconfidence on CIFAR-100 and ImageNet. The results suggest that prioritizing generative quality may come at the cost of calibration.

## 5 CONCLUSION

We presented Dual Adversarial Training (DAT), a framework that replaces SGLD-based EBM learning with adversarial training, resolving the long-standing stability issues that have limited joint energy-based models. By integrating AT for both the discriminative and generative components within a JEM architecture, DAT achieves competitive results across classification, robustness, and generation on CIFAR-10/100 and ImageNet 256×256, while producing faithful counterfactual explanations. These results suggest that adversarial training—beyond its well-known role in robustness—provides an effective and scalable foundation for energy-based generative learning.

### ACKNOWLEDGMENTS

This work was supported in part by Advanced Micro Devices, Inc. under the AMD University Program's AI & HPC Cluster and in part by a Lightspeed grant. TW was supported by a Vitalik Buterin PhD Fellowship.

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

APPENDIX OVERVIEW

**Contents**

# A  THEORY & METHOD

## A.1  EXTENDED RELATED WORK

**Connections between adversarial robustness and energy-based models** Zhu et al. (2021) reinterpret adversarially trained classifiers as joint energy-based models, showing that adversarial training implicitly flattens the energy landscape around real data by reducing the energy of nearby high-energy adversarial examples. They identify $E_\theta(x, y)$ as the key energy term for conditional generation and propose JEAT, which employs energy-based adversarial perturbations and SGLD sampling for likelihood estimation and generation. Mirza et al. (2024) extend this analysis by decomposing the cross-entropy loss as $\mathcal{L}_{\text{CE}}(x, y; \theta) = E_\theta(x, y) - E_\theta(x)$, revealing that untargeted attacks increase the joint energy $E_\theta(x^*, y)$ (reducing classifier confidence of class $y$) while decreasing the marginal energy $E_\theta(x^*)$. They show that robust overfitting corresponds to divergence between $E_\theta(x)$ and $E_\theta(x^*)$, and that state-of-the-art robust models achieve better generalization by smoothing the marginal energy landscape around natural data. Wang et al. (2022a) proposed a unified Contrastive Energy-based Model (CEM) framework that interprets adversarial training as biased maximum likelihood estimation of an energy-based model $p_\theta(x, y) = \exp(f_\theta(x, y))/Z(\theta)$. Unlike JEM (Grathwohl et al., 2019) which samples negative examples from random noise via Langevin dynamics, CEM shows that PGD-generated adversarial perturbations from real data serve as implicit negative samples, providing more stable training without requiring random noise or OOD data. The framework unifies supervised (P-CEM) and unsupervised (NP-CEM) scenarios, revealing connections between adversarial training, contrastive learning, and energy-based modeling, and enables improved sampling algorithms that achieve improved generative performance over prior EBM-based methods.

**Learning EBMs with adversarial training** Yin et al. (2022) explored an alternative approach to learning EBMs by leveraging the mechanism of Adversarial Training (AT). They established a connection between the objective of binary AT (discriminating real data from adversarially perturbed out-of-distribution data) and the SGLD-based maximum likelihood training commonly used for EBMs. Specifically, they showed that the binary classifier learned via AT implicitly defines an energy function that models the support of the data distribution, assigning low energy to in-distribution regions and high energy to out-of-distribution (OOD) regions. The PGD attack used in AT to generate adversarial samples from OOD data was interpreted as a non-convergent sampler that produces contrastive data, analogous to MCMC sampling in EBM training. Although the resulting energy function can only capture the support rather than recover the exact density, their model achieves competitive image generation performance compared to explicit EBMs. Notably, this AT-based EBM learning approach is more stable than traditional MCMC-based EBM training and demonstrated strong performance in worst-case out-of-distribution detection, similar to methods like RATIO (Augustin et al., 2020). However, AT-EBM focuses on unconditional generative modeling and employs an explicit $R_1$ gradient penalty to stabilize training, which can constrain model expressiveness. Our work incorporates AT-based EBM learning into the JEM framework to perform conditional generative modeling with implicit $R_1$ regularization from adversarial training, using ancestral sampling from the conditional distribution $p(x|y)$ rather than the marginal distribution $p(x)$.

**Improving joint energy-based models** Building on the original JEM framework (Grathwohl et al., 2019), several works have explored techniques to improve training stability and performance. Yang and Ji (2021) (JEM++) introduced multiple training improvements: (1) Proximal SGLD that constrains samples within an $L_p$-norm ball of previous samples via gradient clamping for improved stability; (2) YOPO-inspired acceleration (PYLD) that reduces redundant backpropagation by exploiting the coupling between samples and first-layer weights; (3) Informative initialization from a class-conditional Gaussian mixture distribution estimated from training data, which accelerates SGLD convergence, improves stability, and enables batch normalization. Korst and Asadulaev (2022) (Robust-JEM) further enhanced JEM++ by incorporating adversarial training into the discriminative component, empirically observing improved training stability. At inference time, they propose a "combined inference" approach where initial samples from PGD adversarial attacks are refined using SGLD, improving generative performance. However, both JEM++ and Robust-JEM fundamentally still rely on SGLD for sampling and MLE-based objectives for the generative component, inheriting SGLD's intrinsic instability issues. While these works introduced valuable techniques for improving SGLD stability, they did not fundamentally resolve the instability of SGLD-based training and both remain limited to CIFAR-scale ($32\times32$) datasets. Our approach departs fundamentally from this line of work by: (1) providing mathematical analysis and empirical evidence showing that adversarial

training offers implicit $R_1$ regularization (Section A.3); (2) replacing the MLE-based objective with BCE-based gradients; (3) using deterministic PGD-based sampling instead of stochastic Langevin dynamics; and (4) introducing a two-stage training strategy to address the incompatibility between batch normalization and EBM training. This enables scaling to high-resolution ImageNet synthesis (256×256) with state-of-the-art generative quality.

**In- and out-distribution adversarial robustness** Addressing the multifaceted challenge of creating models that are simultaneously accurate, robust, and reliable on out-of-distribution (OOD) data, Augustin et al. (2020) proposed RATIO (Robustness via Adversarial Training on In- and Out-distribution). Their approach combines standard adversarial training (AT) on the in-distribution data, aimed at improving robustness against adversarial examples, with a form of AT on OOD data, which enforces low and uniform confidence predictions within a neighborhood around OOD samples. The combined objective trains the model to maintain correct, robust classifications for in-distribution data while actively discouraging high-confidence predictions for OOD inputs, even under adversarial manipulation. Augustin et al. (2020) demonstrated that RATIO achieves state-of-the-art $\ell_2$ robustness on datasets like CIFAR-10, often with less degradation in clean accuracy compared to standard AT alone. Furthermore, they showed that RATIO yields reliable OOD detection performance, particularly in worst-case scenarios where OOD samples are adversarially perturbed to maximize confidence. Their work also highlighted that the $\ell_2$ robustness fostered by RATIO enables the generation of meaningful visual counterfactual explanations directly in pixel space, where optimizing confidence towards a target class results in the emergence of corresponding class-specific visual features.

**Robust classifiers for image synthesis and manipulation** Santurkar et al. (2019) demonstrated that adversarially robust classifiers can serve as powerful primitives for diverse image synthesis tasks. The core insight of their work is that the process of adversarial training—which optimizes the worst-case loss over an $\ell_2$ perturbation set rather than expected loss—compels a model to learn more perceptually aligned and human-interpretable feature representations by preventing reliance on imperceptible artifacts. Based on this insight, Santurkar et al. (2019) showed that simple gradient ascent on class scores from such robust classifiers enables a unified framework for image generation, inpainting, image-to-image translation, super-resolution, and interactive manipulation—tasks typically requiring specialized GAN architectures or complex generative models.

## A.2 OPTIMAL SOLUTION TO THE JOINT OBJECTIVE

We formally characterize the distribution learned by our joint objective (Eq. 8). Our analysis builds on the theoretical framework from Yin et al. (2022) (AT-EBM) and extends it to the conditional modeling setting by deriving the optimal class logits under the joint objective.

**Optimal solution for class logits $f_\theta^*(x)[y]$.** Following Yin et al. (2022), the generative component of our objective can be expressed as a maximin optimization problem. Let $D(x) = \sigma(-E_\theta(x))$ where $E_\theta(x) = -\log \sum_y \exp(f_\theta(x)[y])$ is the marginal energy function. For the theoretical analysis below, we assume (1) the PGD attack in Eq. 9 converges to the global minimum of $E_\theta(x)$, (2) the model has sufficient capacity, and (3) infinite training data from $p_{\text{data}}(x, y)$. Under these assumptions, minimizing $\mathcal{L}_{\text{BCE}}(\theta)$ implicitly solves:

$$\max_D \min_{p_T} U(D, p_T) = \mathbb{E}_{x \sim p_{\text{data}}(x)}[\log D(x)] + \mathbb{E}_{x \sim p_T}[\log(1 - D(x))] \tag{14}$$

where $p_T$ represents the distribution of samples after PGD attack initialized from the auxiliary out-of-distribution dataset $p_{\text{ood}}$.

Under these assumptions, the optimal solution to Eq. 14 is characterized by Proposition 1 of Yin et al. (2022), which shows that at optimum $U(D^*, p_T^*) = -\log(4)$ with:

1. $D^*(x) = \frac{1}{2}$ for all $x \in \text{Supp}(p_{\text{data}}(x))$
2. $D^*(x) \leq \frac{1}{2}$ for all $x \notin \text{Supp}(p_{\text{data}}(x))$
3. $p_T^*$ is supported on $\{x : D(x) = \frac{1}{2}\}$

where $\text{Supp}(p_{\text{data}}(x))$ denotes the support of the marginal data distribution.

The above result characterizes only the marginal energy $E_\theta(x)$, leaving the individual class logits $f_\theta(x)[y]$ underdetermined. We now derive their optimal values by incorporating the discriminative objective $\mathcal{L}_{\text{AT-CE}}$.

**Proposition A.1** (Optimal class logits)**.** *Under the assumptions stated above, at the optimal solution to the joint objective $\mathcal{L}(\theta) = \mathcal{L}_{AT\text{-}CE}(\theta) + \mathcal{L}_{BCE}(\theta)$, the class logits satisfy on the support:*

$$f_\theta^*(x)[y] = \log p_{data}(y|x) \quad \text{for all } x \in \text{Supp}(p_{data}(x)) \tag{15}$$

*Proof.* From the AT-EBM result above, on the support we have $D^*(x) = \sigma(-E_\theta^*(x)) = \frac{1}{2}$, which implies:

$$E_\theta^*(x) = 0 \implies -\log \sum_y \exp(f_\theta^*(x)[y]) = 0 \implies \sum_y \exp(f_\theta^*(x)[y]) = 1 \tag{16}$$

The conditional distribution is defined as:

$$p_\theta(y|x) = \frac{\exp(f_\theta(x)[y])}{\sum_{y'} \exp(f_\theta(x)[y'])} \tag{17}$$

Substituting the constraint from Eq. 16:

$$p_\theta^*(y|x) = \frac{\exp(f_\theta^*(x)[y])}{1} = \exp(f_\theta^*(x)[y]) \tag{18}$$

The adversarial cross-entropy objective $\mathcal{L}_{\text{AT-CE}}$ minimizes $-\log p_\theta(y|x)$ over worst-case perturbations of $(x, y) \sim p_{\text{data}}(x, y)$. Under our capacity assumption, at optimality on the support this reduces to standard cross-entropy behavior, yielding $p_\theta^*(y|x) = p_{\text{data}}(y|x)$. Since the optimal predictive distribution naturally satisfies the normalization constraint $\sum_y p_{\text{data}}(y|x) = 1$, the solution to the discriminative objective is fully compatible with the marginal energy constraint $\sum_y \exp(f_\theta^*(x)[y]) = 1$ derived from the generative objective. Therefore:

$$\exp(f_\theta^*(x)[y]) = p_{\text{data}}(y|x) \tag{19}$$

Taking logarithms gives Eq. 15. $\qquad \square$

**On-support behavior.** Proposition A.1 implies that on the support: (1) the joint energy equals the negative conditional log-probability $E_\theta^*(x, y) = -f_\theta^*(x)[y] = -\log p_{\text{data}}(y|x)$, (2) the marginal energy is constant $E_\theta^*(x) = 0$, and (3) the unnormalized marginal density is constant, $p_\theta^*(x) \propto \exp(-E_\theta^*(x)) = 1$. This confirms that the model assigns *equal density to all points on the support*, rather than learning the true data density $p_{\text{data}}(x)$. For datasets with deterministic labels where $p_{\text{data}}(y|x) = \delta_{y, y_{\text{true}}(x)}$, this implies $f_\theta^*(x)[y] = 0$ for the true class and $f_\theta^*(x)[y] = -\infty$ otherwise.

**Off-support behavior.** For $x \notin \text{Supp}(p_{\text{data}}(x))$, the optimal solution to Eq. 14 constrains $D^*(x) \leq \frac{1}{2}$, which implies $E_\theta^*(x) \geq 0$ and thus $\sum_y \exp(f_\theta^*(x)[y]) \leq 1$. Since $\mathcal{L}_{\text{AT-CE}}$ is only computed on the data support and its adversarial perturbations, the individual class logits are underdetermined for points far from the support. From the constraint $\sum_y \exp(f_\theta^*(x)[y]) \leq 1$, we have $f_\theta^*(x)[y] \leq 0$ for all classes $y$, and consequently $E_\theta^*(x, y) = -f_\theta^*(x)[y] \geq 0$.

**Comparison.** For datasets with deterministic labels, this creates a hierarchical energy structure:

- Valid pairs ($x \in \text{Supp}, y = y_{\text{true}}$): The joint energy is exactly $E_\theta^*(x, y) = 0$.
- On-support, incorrect labels ($x \in \text{Supp}, y \neq y_{\text{true}}$): The joint energy diverges to $E_\theta^*(x, y) = \infty$.
- Off-support (OOD) ($x \notin \text{Supp}$, any $y$): The joint energy is bounded below, $E_\theta^*(x, y) \geq 0$, but the exact value is underdetermined.

Thus, the learned energy function $E_\theta(x, y)$ enables robust classification (via $\arg\min_y E_\theta(x, y)$), generation (via minimizing $E_\theta(x, y)$ over $x$), and OOD detection (via thresholding $\min_y E_\theta(x, y)$).

**Finite-step vs. convergent PGD.** The theoretical analysis above assumes the PGD attack converges to the global minimum of $E_\theta(x)$, which leads to the maximin formulation in Eq. 14. In practice, however, we use finite-step PGD (Eq. 9), which does not explicitly solve the inner minimization problem, creating a crucial gap between theory and practice. Our experiments demonstrate that our approach achieves diverse, high-quality generation, suggesting that rather than converging to energy minima, finite-step PGD behaves like a sampler in practice: starting from different initializations in the OOD dataset, it explores different regions of the energy landscape, leading to diverse generated samples.

**Summary and implications.** Our formal analysis reveals that our joint objective learns a fundamentally different quantity than MLE-based methods. While MLE-based JEM theoretically learns the full density $p_\theta(x)$ with a valid partition function (though in practice short-run SGLD fails to achieve this), our approach explicitly learns the *support* of the data distribution with $E_\theta(x) = 0$ on the support. The optimal class logits $f_\theta^*(x)[y] = \log p_{\text{data}}(y|x)$ (Proposition A.1) reveal how the joint objective uniquely determines the solution on the support.

This support-based characterization has important implications. The constant marginal energy $E_\theta^*(x) = 0$ on the support means the model assigns equal unnormalized density to all points on the support, theoretically discarding frequency information about the data distribution. This represents a significant theoretical limitation: the model cannot distinguish between common and rare examples within the support, and thus cannot perform density estimation tasks that require modeling relative frequencies.

Despite this limitation, the support-based formulation provides clear advantages: superior training stability compared to SGLD-based methods, robust classification, and effective OOD detection—benefits that do not require density information. Additionally, our strong empirical generation results demonstrate that the finite-step PGD dynamics discussed above act as an effective sampler, capturing density variations despite the theoretical prediction of uniformity. Overall, the support-based approach provides a practical and stable framework for joint modeling, trading full density estimation for superior training stability and strong empirical performance.

## A.3 IMPLICIT $R_1$ REGULARIZATION

Building on the operator norm equivalence established by Roth et al. (2020), we show that adversarial training implicitly bounds the $R_1$ gradient penalty, eliminating the need for explicit gradient regularization. Consider a classifier $f : \mathbb{R}^d \to \mathbb{R}^K$ producing logits for $K$ classes.

$R_1$ regularization directly penalizes large gradients of the true class logit:

$$\mathcal{L}_{R_1} = \mathbb{E}_{(x,y)\sim p_{\text{data}}} \left[ \|\nabla_x f_y(x)\|_2^2 \right] \tag{20}$$

In practice, adversarial training uses cross-entropy loss to enforce consistent predictions:

$$\mathcal{L}_{\text{AT}} = \mathbb{E}_{(x,y)\sim p_{\text{data}}} \left[ \max_{\|\delta\|_p \leq \epsilon} L(f(x+\delta), y) \right] \tag{21}$$

**Adversarial training as operator norm regularization.** Roth et al. (2020) proved that $\ell_p$-norm constrained adversarial training with an $\ell_q$-norm loss on the logits is *equivalent* to data-dependent $(p, q)$-operator norm regularization:

$$\mathcal{L}_{\text{AT}} \equiv \mathcal{L}_{\text{standard}} + \lambda(\epsilon) \cdot \|J_f(x)\|_{p,q} \tag{22}$$

where $\lambda(\epsilon)$ is an implicit regularization coefficient that scales linearly with the attack budget $\epsilon$. Here $J_f(x) \in \mathbb{R}^{K \times d}$ denotes the input-Jacobian of the logits, and the $(p, q)$-operator norm is defined as:

$$\|J_f(x)\|_{p,q} := \max_{\|v\|_p=1} \|J_f(x)v\|_q \tag{23}$$

This measures the maximal signal amplification when propagating a norm-bounded input perturbation through the linearized network. For $\ell_2$-constrained adversarial training, the $(2, 2)$-operator norm reduces to the spectral norm $\sigma_{\max}(J_f(x))$, the largest singular value of the Jacobian:

$$\mathcal{L}_{\text{AT}} \equiv \mathcal{L}_{\text{standard}} + \lambda(\epsilon) \cdot \sigma_{\max}(J_f(x)) \tag{24}$$

The optimal perturbation direction aligns with the dominant right singular vector of $J_f(x)$.

**Connection to $R_1$ regularization.** Since adversarial training regularizes $\sigma_{\max}(J_f(x))$, we trace how this bound propagates to $\|\nabla_x f_y(x)\|_2$, which $R_1$ penalizes. By definition, the spectral norm satisfies $\sigma_{\max}(J_f(x)) = \sigma_{\max}(J_f(x)^T) = \max_{\|v\|_2=1} \|J_f(x)^T v\|_2$. Let $e_y \in \mathbb{R}^K$ be the standard basis vector for class $y$. The product $J_f(x)^T e_y$ extracts the $y$-th column of $J_f(x)^T$, which corresponds to the gradient $\nabla_x f_y(x)$. Since $\|e_y\|_2 = 1$, we have:

$$\|\nabla_x f_y(x)\|_2 = \|J_f(x)^T e_y\|_2 \leq \max_{\|v\|_2=1} \|J_f(x)^T v\|_2 = \sigma_{\max}(J_f(x)) \tag{25}$$

Thus, adversarial training's regularization of $\sigma_{\max}(J_f(x))$ upper-bounds the $R_1$ gradient penalty $\|\nabla_x f_y(x)\|_2$.

**Bound on all per-class gradients.** This analysis reveals that adversarial training provides a stronger regularization than $R_1$ alone. While $R_1$ penalizes only $\|\nabla_x f_y(x)\|_2$, adversarial training simultaneously bounds all per-class gradient norms:

$$\|\nabla_x f_k(x)\|_2 \leq \sigma_{\max}(J_f(x)), \quad \forall k \in \{1, \ldots, K\} \tag{26}$$

This includes the probability-weighted mean gradient $\sum_k p_k \nabla_x f_k(x)$, since its norm is bounded by $\max_k \|\nabla_x f_k(x)\|_2$.

**Effective regularization strength scales with attack budget.** An important consequence of this framework is that the effective regularization strength scales with the adversarial attack budget $\epsilon$. From the equivalence established above, the adversarial training objective implicitly regularizes $\sigma_{\max}(J_f(x))$ with strength $\lambda(\epsilon)$. For the network to maintain bounded total loss, the regularization term must remain bounded: $\lambda(\epsilon) \cdot \sigma_{\max}(J_f(x)) \lesssim C$ for some constant $C$. Since $\lambda(\epsilon)$ scales linearly with $\epsilon$, this implies:

$$\sigma_{\max}(J_f(x)) \lesssim \frac{C}{\lambda(\epsilon)} \tag{27}$$

Combining with our direct bound:

$$\|\nabla_x f_y(x)\|_2 \leq \sigma_{\max}(J_f(x)) \lesssim \frac{C}{\lambda(\epsilon)} \tag{28}$$

Thus, the $R_1$ gradient norm scales inversely with the regularization strength. Larger attack budgets $\epsilon$ induce stronger implicit regularization $\lambda(\epsilon)$, leading to lower $R_1$ gradients.

**Connection to GAN regularization.** GAN training commonly employs spectral normalization (Miyato et al., 2018) on weight matrices and $R_1$ gradient penalties (Mescheder et al., 2018) for stability. The weight spectral norm provides a global upper bound on the Jacobian spectral norm: $\sigma(J_f(x)) \leq \prod_\ell \sigma(W^\ell)$ (Yoshida and Miyato, 2017). However, Roth et al. (2020) show this bound can be loose as it ignores data-dependent activation patterns. AT's equivalence to data-dependent spectral norm regularization is more targeted, directly controlling sensitivity at actual data points. Combined with our analysis showing this bounds $R_1$, AT provides a principled implicit alternative to both explicit GAN regularizers.

**Empirical validation.** We validate this theoretical analysis by tracking $R_1$ gradient norms during Stage 2 joint training on ImageNet 256×256 (ResNet-50, $T = 15$). Figure 2 compares two settings: (1) adversarial training on the discriminative loss ($\mathcal{L}_{\text{AT-CE}}$), and (2) standard training on the discriminative loss. As predicted by the analysis above, the adversarial training curve remains bounded and stable throughout training, while standard training exhibits gradient explosion—with $R_1$ values increasing by orders of magnitude in later stages. While the exact equivalence from Roth et al. (2020) holds under technical conditions (small $\epsilon$ relative to ReLU cell size, infinite attack step size limit), our empirical observations are consistent with the theoretical predictions. This confirms that implicit regularization from adversarial training suffices to maintain bounded $R_1$ gradients, thereby enabling stable energy-based model training.

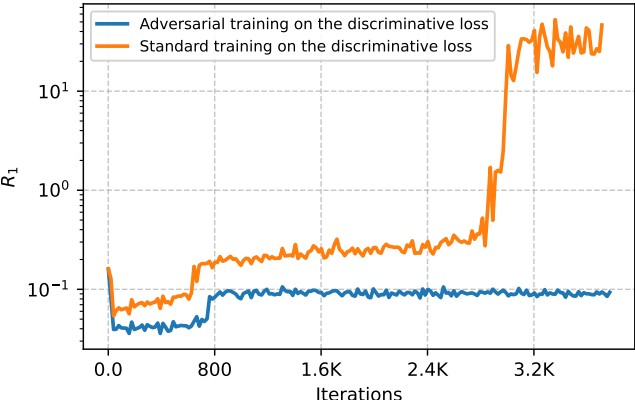

Figure 2: $R_1$ gradient norm during Stage 2 joint training on ImageNet 256×256 (ResNet-50, $T = 15$). Y-axis is in log scale. Adversarial training maintains bounded gradients while standard training exhibits gradient explosion.

## A.4 Training algorithm

The complete training procedure for our combined objective (Eq. 12) is detailed in Algorithm 1. We note that to train the generative component $\mathcal{L}_{\text{BCE}}$, we sample from $p_\theta(x)$ to estimate $\mathbb{E}_{x \sim p_\theta(x)}[-\nabla_\theta E_\theta(x)]$ in Eq. 5. In the context of JEM, there are broadly two strategies for drawing samples from $p_\theta(x)$ (Grathwohl et al., 2019):

1. Direct sampling from the marginal distribution using gradient-based MCMC (e.g., SGLD or PGD) on the marginal energy $E_\theta(x) = -\log \sum_y \exp(f_\theta(x)[y])$, as implied by Eq. 9.

2. Ancestral sampling, which first draws a label $y \sim p_{\text{data}}(y)$, then samples $x \sim p_\theta(x|y)$ by running gradient-based MCMC on the joint energy $E_\theta(x, y) = -f_\theta(x)[y]$.

Although both approaches yield unbiased estimates, we find ancestral sampling to be practically superior for training stability, possibly because it leverages the classifier's existing strong class representations to provide better mode coverage and mixing properties, while direct sampling from the marginal distribution often diverges.

Consequently, our implementation adopts ancestral sampling when generating contrastive samples (Algorithm 1). Specifically, we first sample a label $y' \sim p_{\text{data}}(y)$, then generate a contrastive sample $x_T$ by performing $T$ iterations of PGD on the negative joint energy function $-E_\theta(x, y')$, starting from an initial sample $x_0 \sim p_{\text{ood}}$. This class-conditional contrastive sample $x_T$ is then used in the $\mathcal{L}_{\text{BCE}}$ objective (Eq. 8), whose gradient (Eq. 7) provides an approximation to the EBM gradient (Eq. 5).

---

**Algorithm 1** DAT training: Given network logits $f_\theta$, in-distribution dataset $p_{\text{data}}$, auxiliary out-of-distribution dataset $p_{\text{ood}}$, classification AT bound $\epsilon$, PGD iterations $T$, PGD step size $\eta$

---

1: **while** not converged **do**
2:    Sample $(x, y) \sim p_{\text{data}}(x, y)$, apply aggressive augmentation to $x$
3:    Sample $\hat{x} \sim p_{\text{data}}(x)$, $x_0 \sim p_{\text{ood}}(x)$, apply mild augmentation to $\hat{x}$ and $x_0$
4:    Solve $x_{adv} = \arg\max_{x' \in B(x,\epsilon)} \mathcal{L}_{\text{CE}}(\theta; x', y)$ via PGD attack
5:    $\mathcal{L}_{\text{AT-CE}}(\theta) = -\log p_\theta(y|x_{adv})$                      ▷ Robust classification loss
6:    Initialize $x_t \leftarrow x_0$ for $t = 0$, sample $y' \sim p_{\text{data}}(y)$
7:    **for** $t \in \{1, \ldots, T\}$ **do**                      ▷ Generate contrastive sample for EBM
8:        $g = \nabla_x E_\theta(x_{t-1}, y')$                      ▷ Energy gradient
9:        $x_t \leftarrow x_{t-1} - \eta \cdot g/||g||_2$                      ▷ Normalized gradient descent step
10:    **end for**
11:    $\mathcal{L}_{\text{BCE}}(\theta) = -\log(\sigma(-E_\theta(\hat{x}))) - \log(1 - \sigma(-E_\theta(x_T)))$                      ▷ Generative modeling loss
12:    $\mathcal{L}(\theta) = \mathcal{L}_{\text{AT-CE}}(\theta) + \mathcal{L}_{\text{BCE}}(\theta)$
13:    Compute parameter gradients $\nabla_\theta \mathcal{L}(\theta)$ and update $\theta$
14: **end while**

---

# B TRAINING & EVALUATION

## B.1 IMPLEMENTATION DETAILS

### B.1.1 HYPERPARAMETERS

We follow the two-stage training approach described in Section 3.5. Table 3 summarizes the hyperparameters for both stages across datasets.

For Stage 1, we use a pretrained CIFAR-10 model from RATIO (Augustin et al., 2020) and pretrained ImageNet ResNet-50 and WRN-50-4 models from Salman et al. (2020), while training a CIFAR-100 model following RATIO with the hyperparameters in Table 3. For ConvNeXt-Large experiments on ImageNet, we use the pretrained checkpoint from Singh et al. (2023). We select the EMA model with the best robust test accuracy as the final Stage 1 model.

For Stage 2, we initialize from the Stage 1 model. For ResNet and WRN, we disable batch normalization by fixing all BN layers to evaluation mode. During this stage, we optimize the complete objective $\mathcal{L}(\theta) = \mathcal{L}_{\text{AT-CE}}(\theta) + \mathcal{L}_{\text{BCE}}(\theta)$ using fixed learning rates as specified in Table 3. For the discriminative component $\mathcal{L}_{\text{AT-CE}}(\theta)$, CIFAR and ImageNet ResNet/WRN models continue to use the same adversarial training parameters as Stage 1, while ConvNeXt transitions from $\ell_\infty$ to $\ell_2$ perturbations with adjusted PGD parameters (see Table 4). The generative component $\mathcal{L}_{\text{BCE}}(\theta)$ employs the parameters detailed in Table 5.

We select the Stage 2 checkpoint with the best FID score for the final evaluation reported in Section 4.3.

Table 3: Training hyperparameters for both stages. Epochs for Stage 2 are estimated based on the number of in-distribution images seen by the discriminative component during training.

| | CIFAR-10/100 | ImageNet | ImageNet |
|---|---|---|---|
| Architecture | WRN-34-10 | ResNet-50/WRN-50-4 | ConvNeXt-L-CvSt |
| BatchNorm | Enabled (Stage 1), Disabled (Stage 2) | Enabled (Stage 1), Disabled (Stage 2) | N/A |
| LayerNorm | N/A | N/A | Enabled |
| Optimizer | SGD with Nesterov | SGD with Nesterov | AdamW |
| Weight decay | $5 \times 10^{-4}$ | $1 \times 10^{-4}$ (Stage 1), $5 \times 10^{-4}$ (Stage 2) | 0.05 |
| Batch size | 128 | 512 | 756 (Stage 1), 512 (Stage 2) |
| EMA | Enabled | Enabled | Enabled |
| LR (Stage 1) | 0.1 (cosine schedule) | 0.1 (step decay at epochs 30, 60, 90) | 0.001 (cosine decay with warm-up) |
| LR (Stage 2) | 0.001 (CIFAR-10), 0.009 (CIFAR-100) | 0.001 | 0.0003 |
| Epochs (Stage 1) | 300 | 90 | 100 |
| Epochs (Stage 2) | 26 (CIFAR-10), 30 (CIFAR-100) | 0.78 (ResNet-50), 0.39 (WRN-50-4) | 1.09 |

Table 4: Adversarial training parameters for $\mathcal{L}_{\text{AT-CE}}$ (identical across stages for CIFAR and ImageNet ResNet/WRN).

| | CIFAR-10/100 | ImageNet (ResNet/WRN) | ImageNet (ConvNeXt-L-CvSt) |
|---|---|---|---|
| PGD steps | 10 | 2 | 2 |
| PGD step size | 0.1 | 2.0 | 2/255 ($\ell_\infty$, Stage 1), 2.0 ($\ell_2$, Stage 2) |
| Perturbation bound | $\ell_2, \epsilon = 0.5$ | $\ell_2, \epsilon = 3.0$ | $\epsilon = 4/255$ ($\ell_\infty$, Stage 1), $\epsilon = 3.0$ ($\ell_2$, Stage 2) |

Table 5: Adversarial training parameters for $\mathcal{L}_{\text{BCE}}$ (Stage 2 only).

| | CIFAR-10/100 | ImageNet (ResNet/WRN) | ImageNet (ConvNeXt-L-CvSt) |
|---|---|---|---|
| Max PGD steps ($T$) | 40/45/50 | 15/30/65 | 110 |
| PGD step size | 0.1 | 2.0 | 3.0 |
| $\ell_2$ perturbation bound | None (unconstrained) | None (unconstrained) | None (unconstrained) |
| OOD data source | 80M Tiny Images (Torralba et al., 2008) | Open Images (Krasin et al., 2016) | Open Images (Krasin et al., 2016) |

### B.1.2 DATA AUGMENTATION

As described in Section 4.1, we use separate data augmentation pipelines for the discriminative and generative components. Table 6 summarizes the augmentation strategies for Stage 2 training. For CIFAR-10/100 and ImageNet ResNet/WRN, the pretrained models were trained with the same augmentations as Stage 2 $\mathcal{L}_{\text{AT-CE}}$ (from Augustin et al. (2020) for CIFAR and Salman et al. (2020) for ImageNet). For ImageNet ConvNeXt-L-CvSt, the pretrained model from Singh et al. (2023) was trained with heavier augmentations (RandAugment + MixUp + CutMix + Random Erasing) than the Stage 2 $\mathcal{L}_{\text{AT-CE}}$ strategy in Table 6. Examples of CIFAR-10 augmentations are shown in Figure 3.

Figure 4 shows CIFAR-10 Stage 2 training curves under various augmentation strategies for $\mathcal{L}_{\text{BCE}}$ (AutoAugment with Cutout is used throughout for $\mathcal{L}_{\text{AT-CE}}$). The choice of augmentation for the generative component also influences discriminative performance: robust test accuracy declines when using no augmentation. The best FID is achieved with no augmentation or random cropping with padding, both of which minimally distort $p_{\text{data}}$. Overall, we find random cropping with padding provides the optimal balance between discriminative and generative performance.

Table 6: Stage 2 data augmentation strategies for discriminative and generative components.

| Dataset | Component | Augmentation Strategy |
|---|---|---|
| CIFAR-10/100 | $\mathcal{L}_{\text{AT-CE}}$ | AutoAugment + Cutout + RandomHorizontalFlip() |
| | $\mathcal{L}_{\text{BCE}}$ | RandomCrop(32, padding=4) + RandomHorizontalFlip() |
| ImageNet | $\mathcal{L}_{\text{AT-CE}}$ | RandomResizedCrop(256) + RandomHorizontalFlip() |
| | $\mathcal{L}_{\text{BCE}}$ | Resize(256) + CenterCrop(256) + RandomHorizontalFlip() |

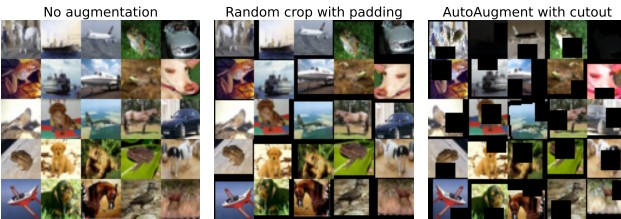

Figure 3: CIFAR-10 samples under different augmentation strategies.

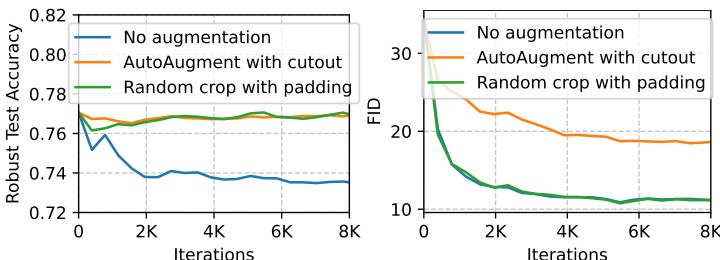

Figure 4: CIFAR-10 training curves under different data augmentations during Stage 2 joint training.

### B.2 COMPUTATIONAL EFFICIENCY

**Training overhead relative to standard AT.** We estimate computational cost as follows: a forward pass costs 1 unit, an input-gradient backward (for PGD) costs 1 unit, and a full backward (for parameter updates) costs 2 units. This gives 2 units per PGD step and 3 units per training update. Under this model, Stage 1 (standard AT) costs $(2K + 3)B$ units per iteration: $2KB$ for the $K$-step PGD attack plus $3B$ for the training update. Stage 2 adds $T$ generative PGD steps ($2TB$ units) and processes three sample types per iteration (adversarial in-distribution, adversarial OOD, and clean in-distribution), tripling the training update cost to $9B$ units, for a total of $(2K + 2T + 9)B$ units per iteration. With $E_1$ and $E_2$ epochs for each stage, the total overhead relative to standard AT is:

$$\text{Overhead} = 1 + \frac{E_2}{E_1} \cdot \frac{2K + 2T + 9}{2K + 3} \qquad (29)$$

This provides an upper bound, as our curriculum learning gradually increases $T$ from a small initial value. Table 7 reports overhead for all configurations: 1.41–1.56× for CIFAR and 1.05–1.36× for ImageNet. Despite Stage 2's higher per-iteration cost, its short duration (less than 1 epoch for most ImageNet configurations) results in minimal additional cost.

**Absolute training time.** Table 7 reports Stage 2 effective training times on AMD MI210/MI250/MI300 accelerators. For baseline models reported in V100-days (Rombach et al., 2022), we convert to MI300-hours using benchmark-based ratios.[1] Our ImageNet 256×256 ConvNeXt-L Stage 2 training takes 20 hours on 8×MI300, which is ∼5× faster than LDM-4-G (110 MI300-equivalent hours). Note that Stage 1 uses a standard AT checkpoint, so total training time depends on whether a pre-trained robust model is available.

**Inference efficiency.** For classification, DAT requires only a single forward pass, identical to standard classifiers. For generation, our models require 13–36 sampling steps (Table 8)—an order of magnitude fewer than diffusion models (250 steps). Table 7 reports the generation throughput, showing ∼5× higher throughput than LDM-4-G while achieving better FID.

Table 7: Computational cost and performance metrics of DAT models. Training overhead computed relative to standard AT using Eq. 29. Stage 2 training times are effective durations excluding evaluation. Baseline training times converted from V100-days; throughput measured on a single MI300 accelerator.

| Model | Params | FID $\downarrow$ | IS $\uparrow$ | Training overhead | Training time (wall-clock hours) | Throughput (img/s) |
|---|---|---|---|---|---|---|
| *CIFAR-10* | | | | | | |
| DAT (WRN-34-10, $T = 40$) | 46M | 9.12 | 9.96 | 1.41× | 10 (4×MI210, Stage 2) | 39 |
| DAT (WRN-34-10, $T = 50$) | 46M | 7.57 | 9.86 | 1.49× | 10 (4×MI210, Stage 2) | 40 |
| *CIFAR-100* | | | | | | |
| DAT (WRN-34-10, $T = 45$) | 46M | 10.70 | 10.83 | 1.52× | 12 (4×MI210, Stage 2) | 40 |
| DAT (WRN-34-10, $T = 50$) | 46M | 9.53 | 11.12 | 1.56× | 12 (4×MI210, Stage 2) | 39 |
| *ImageNet 256×256* | | | | | | |
| DAT (ResNet-50, $T = 15$) | 26M | 6.87 | 317.7 | 1.05× | 2.4 (4×MI210, Stage 2) | 33 |
| DAT (ResNet-50, $T = 30$) | 26M | 5.28 | 319.3 | 1.09× | 3.8 (4×MI210, Stage 2) | 33 |
| DAT (WRN-50-4, $T = 30$) | 223M | 6.23 | 341.0 | 1.05× | 4.7 (4×MI250, Stage 2) | 14 |
| DAT (WRN-50-4, $T = 65$) | 223M | 4.94 | 358.0 | 1.09× | 8.2 (4×MI250, Stage 2) | 13 |
| DAT (ConvNeXt-L-CvSt, $T = 110$) | 198M | 3.29 | 310.2 | 1.36× | 20 (8×MI300, Stage 2) | 5 |
| BigGAN-deep (Brock et al., 2018) | 340M | 6.95 | 203.6 | — | 52-104 (8×MI300-eq) | — |
| ADM-G (Dhariwal and Nichol, 2021) | 608M | 4.59 | 186.7 | — | 390 (8×MI300-eq) | ∼0.17 |
| LDM-4-G (Rombach et al., 2022) | 400M | 3.60 | 247.7 | — | 110 (8×MI300-eq) | ∼0.96 |

---

[1]MI300/V100 $\approx$ 7.4× based on AMD (2024); Lambda Labs (2024).

## B.3 TRAINING DYNAMICS

Figure 5 shows the Stage 2 training curves across all datasets. FID improves rapidly and stabilizes, while robust accuracy is largely preserved. For ConvNeXt-L, which transitions from $\ell_\infty$ to $\ell_2$ perturbations in Stage 2, $\ell_2$ robust accuracy increases during training.

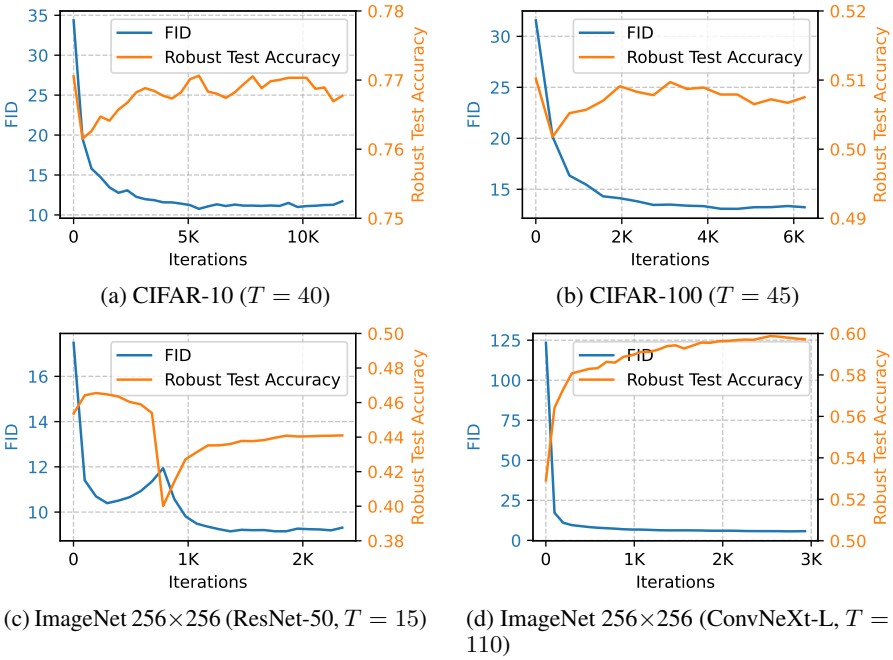

(a) CIFAR-10 ($T = 40$)

(b) CIFAR-100 ($T = 45$)

(c) ImageNet 256×256 (ResNet-50, $T = 15$)

(d) ImageNet 256×256 (ConvNeXt-L, $T = 110$)

Figure 5: Training curves from Stage 2 joint training demonstrating substantial FID improvements while maintaining robust test accuracy (evaluated via PGD attacks; FID measured using 10K generated samples).

### B.4 EVALUATION METHODOLOGY

We evaluate generative performance using FID and IS. FID is computed between 50K class-balanced generated samples and the full training set, while IS is computed on the same set of 50K generated samples.

**Conditional generation.** We generate an equal number of samples for each class. To generate samples for a given class $y$, we first sample an OOD data point $x$ from the OOD data source, and then perform $T$ steps of PGD according to:

$$x_{t+1} = x_t + \eta \frac{\nabla_x(-E_\theta(x_t, y))}{||\nabla_x(-E_\theta(x_t, y))||_2} \tag{30}$$

where $T$ is the number of PGD steps and $\eta$ is the step size (see Table 8).

**Unconditional generation.** For unconditional generation, we sample from the marginal distribution using PGD (Eq. 9).

Table 9 compares conditional and unconditional FIDs across datasets; conditional generation consistently outperforms.

Table 8: Sample generation parameters for FID and IS evaluation. The number of PGD steps for each model and dataset combination is determined through grid search.

| Model | Dataset | PGD steps ($T$) | Step size | OOD data source |
|---|---|---|---|---|
| DAT | CIFAR-10 ($T = 40$) | 33 | 0.2 | 80M Tiny Images |
| | CIFAR-10 ($T = 50$) | 35 | 0.2 | 80M Tiny Images |
| | CIFAR-100 ($T = 45$) | 32 | 0.2 | 80M Tiny Images |
| | CIFAR-100 ($T = 50$) | 33 | 0.2 | 80M Tiny Images |
| | ImageNet (ResNet-50, $T = 15$) | 13 | 8.0 | Open Images |
| | ImageNet (ResNet-50, $T = 30$) | 14 | 8.0 | Open Images |
| | ImageNet (WRN-50-4, $T = 30$) | 17 | 8.0 | Open Images |
| | ImageNet (WRN-50-4, $T = 65$) | 19 | 8.0 | Open Images |
| | ImageNet (ConvNeXt-L-CvSt, $T = 110$) | 36 | 8.0 | Open Images |
| RATIO | CIFAR-10 | 31 | 0.2 | 80M Tiny Images |
| | CIFAR-100 | 14 | 0.2 | 80M Tiny Images |
| Standard AT | CIFAR-10 | 22 | 0.2 | 80M Tiny Images |
| | CIFAR-100 | 15 | 0.2 | 80M Tiny Images |
| | ImageNet (ResNet-50) | 13 | 8.0 | Open Images |
| | ImageNet (WRN-50-4) | 11 | 8.0 | Open Images |
| | ImageNet (ConvNeXt-L-CvSt) | 0 | 8.0 | Open Images |

Table 9: Conditional vs. unconditional generation FIDs.

| | CIFAR-10 | CIFAR-100 | ImageNet 224×224 (ResNet50, $T = 15$) |
|---|---|---|---|
| Conditional generation | 9.07 | 10.70 | 6.64 |
| Unconditional generation | 20.57 | 13.56 | 18.67 |

# C RESULTS & ANALYSES

## C.1 QUALITATIVE SAMPLES

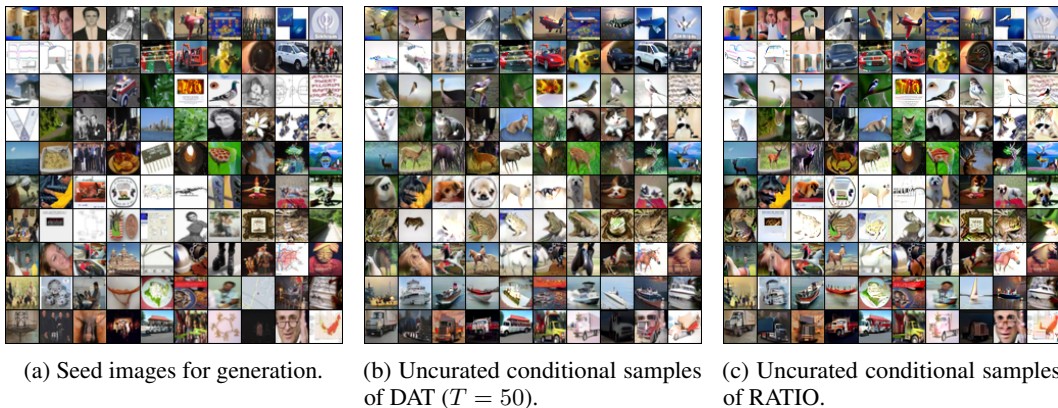

(a) Seed images for generation.

(b) Uncurated conditional samples of DAT ($T = 50$).

(c) Uncurated conditional samples of RATIO.

Figure 6: CIFAR-10 class-conditional generation results.

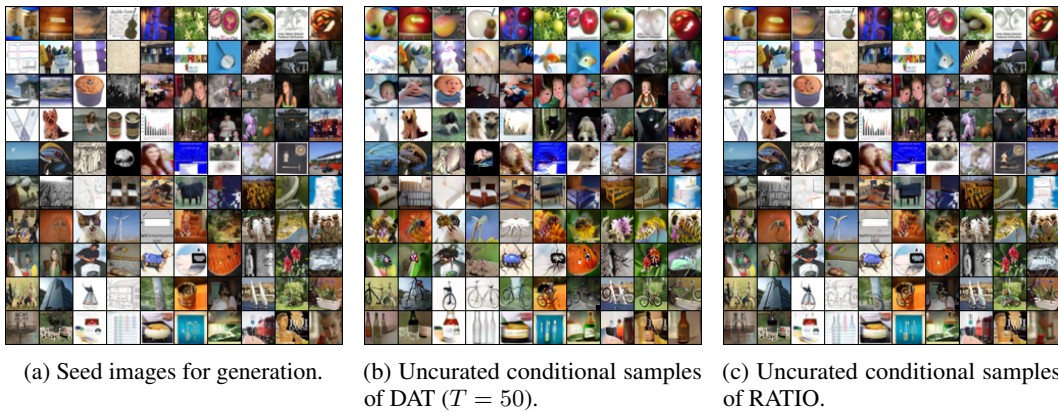

(a) Seed images for generation.

(b) Uncurated conditional samples of DAT ($T = 50$).

(c) Uncurated conditional samples of RATIO.

Figure 7: CIFAR-100 conditional generation results.

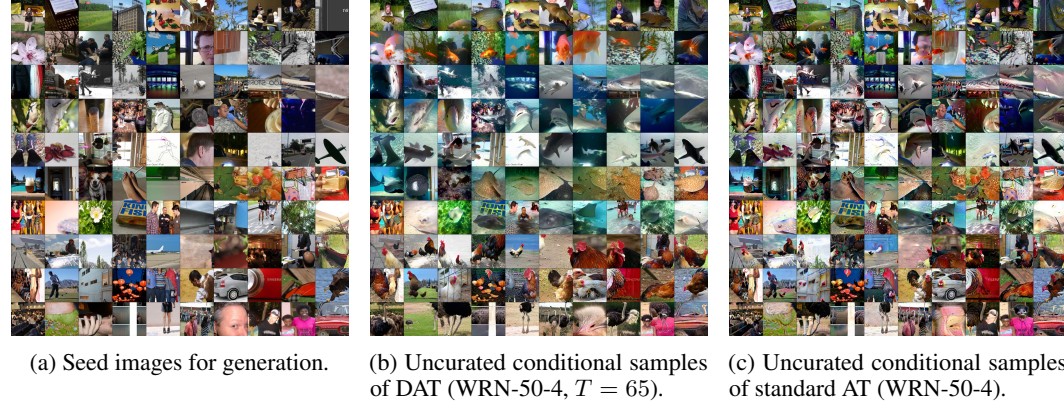

(a) Seed images for generation.

(b) Uncurated conditional samples of DAT (WRN-50-4, $T = 65$).

(c) Uncurated conditional samples of standard AT (WRN-50-4).

Figure 8: ImageNet class-conditional generation results for the first 10 classes: tench, goldfish, great white shark, tiger shark, hammerhead, electric ray, stingray, cock, hen, ostrich (at $256 \times 256$ resolution).

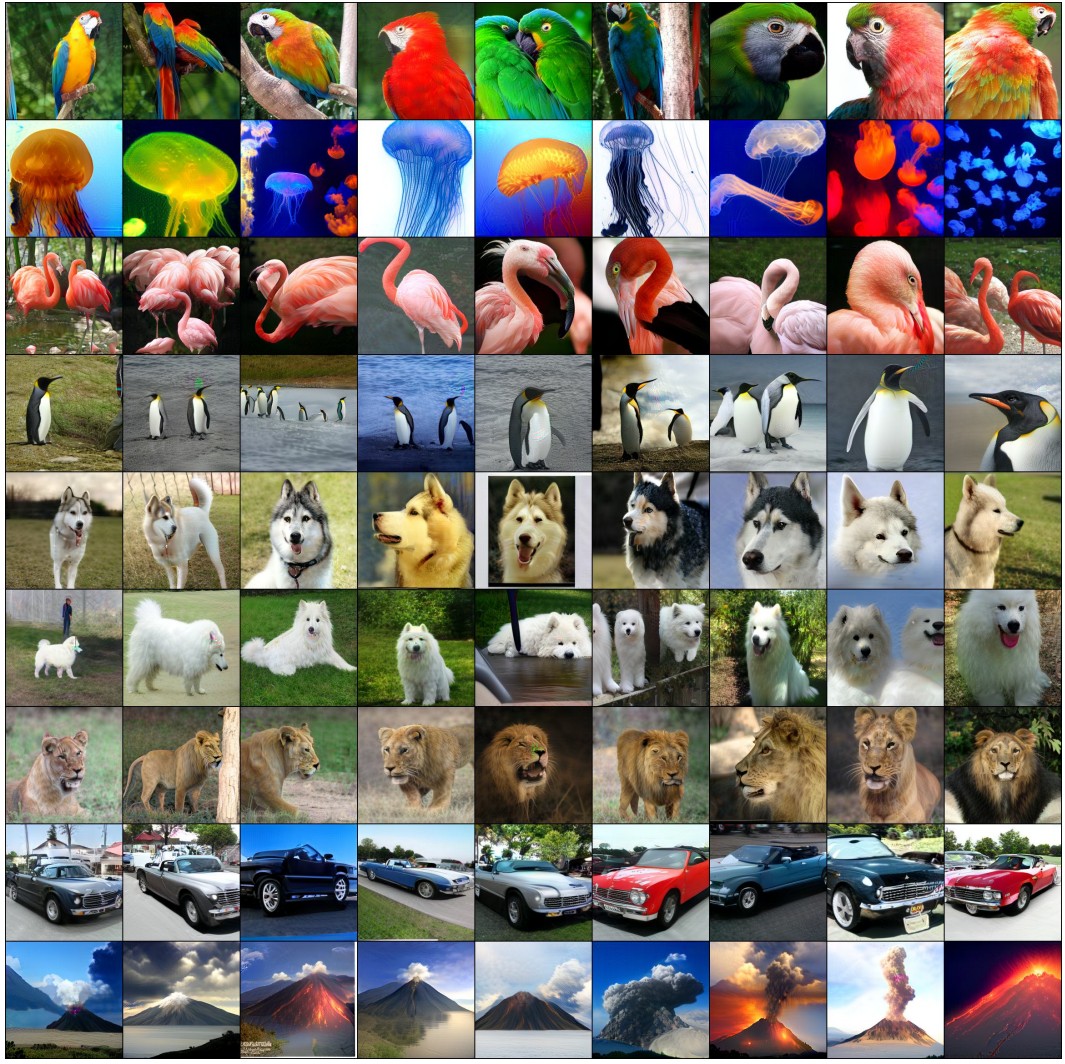

Figure 9: Selected ImageNet conditional generation results for class 88 (macaw), 107 (jellyfish), 130 (flamingo), 145 (king penguin), 248 (husky), 258 (Samoyed), 291 (lion), 511 (convertible), and 980 (volcano). Generated with DAT ConvNeXt-L-CvSt at 256×256.

## C.2 Visual counterfactuals

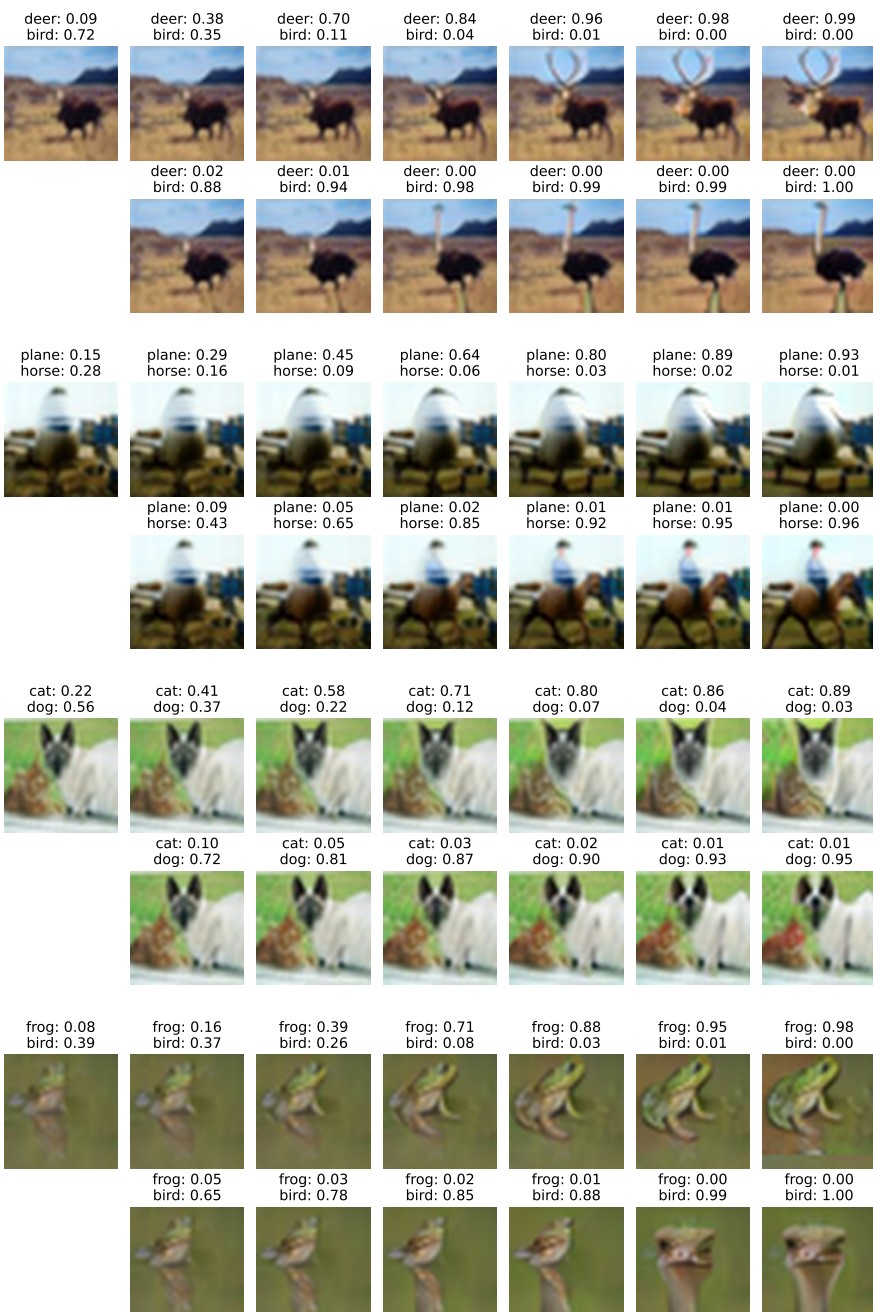

Figure 10: CIFAR-10 counterfactual examples at $\epsilon = 0.5, 1.0, 1.5, 2.0, 2.5, 3.0$. Each panel shows counterfactuals with classifier confidences for the correct class (top row) and a target class (bottom row). As $\epsilon$ increases, counterfactuals progressively resemble the target class distribution with increasing target class confidence.

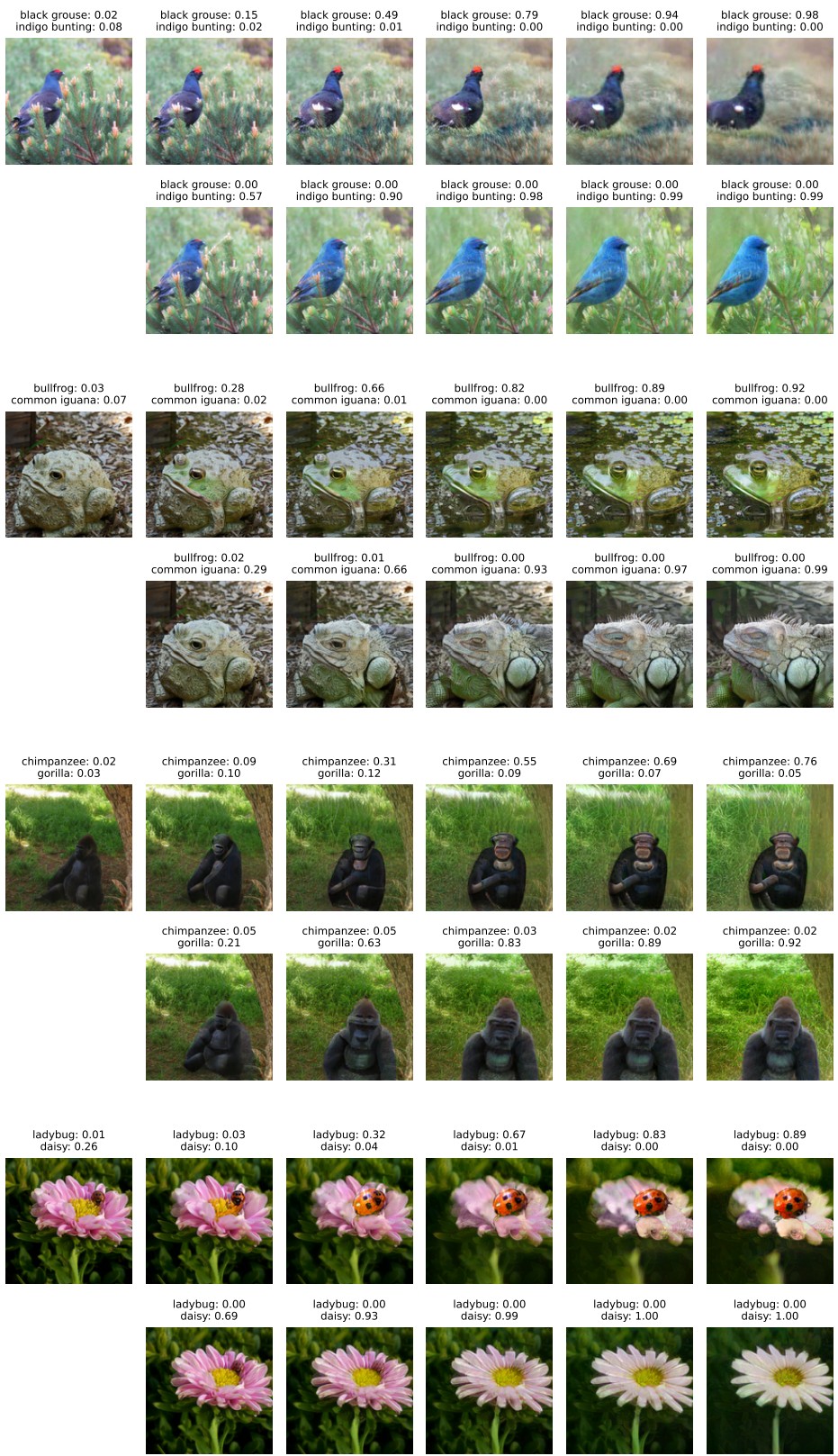

Figure 11: ImageNet counterfactual examples at $\epsilon = 10, 20, 30, 40, 50$.

## C.3   OOD DETECTION

Using models trained with 224×224 generation, we evaluate both standard and worst-case (adversarial) OOD detection. For standard OOD detection, we measure AUROC between in-distribution test samples and clean OOD samples. For worst-case detection, we evaluate against adversarially perturbed OOD samples optimized to maximize the detection score. Results are computed using all in-distribution test samples and 1024 out-of-distribution samples. To generate adversarial OOD samples, we use an $\ell_2$ perturbation budget of 1.0 for CIFAR-10/100 and 3.0 for ImageNet.

**Energy-based detection.** We use an energy-based function $s_\theta(x) = -E_\theta(x)$, which is proportional to $\log p_\theta(x)$ up to an additive constant. To find adversarial OOD inputs for this function, we use PGD to maximize the negative energy:

$$x_{adv} = \underset{x' \in B(x, \epsilon_o)}{\arg\max} -E_\theta(x') \tag{31}$$

where $x$ is a clean OOD input and $B(x, \epsilon_o)$ represents an $\ell_2$-ball of radius $\epsilon_o$ centered at $x$.

**Maximum confidence detection.** We use a maximum confidence function $s_\theta(x) = \max_y p_\theta(y|x)$ that uses the confidence in the most likely class (also used by RATIO (Augustin et al., 2020)). For this detection function, following RATIO (Augustin et al., 2020), we generate adversarial OOD inputs by maximizing the cross-entropy loss against a uniform distribution:

$$x_{adv} = \underset{x' \in B(x, \epsilon_o)}{\arg\max} \mathcal{L}_{\text{CE}}(\theta; x', \mathbf{1}/K) \tag{32}$$

where $\mathbf{1}/K$ represents a uniform distribution over all $K$ classes. Maximizing this loss encourages the model to produce a non-uniform (confident) prediction, thereby maximizing the detection function.

Table 10 compares the two OOD detection functions. The results reveal complementary strengths: the energy-based function $(-E_\theta(x))$ achieves near-perfect AUROC on uniform noise detection, while the maximum confidence function $(\max_y p_\theta(y|x))$ performs better on natural image OOD datasets. Based on these findings, we adopt the maximum confidence score for subsequent comparisons with other methods.

Table 11 presents comparative results across different baselines. Notably, our DAT model achieves OOD detection performance comparable to standard AT on natural image datasets (CIFAR-100, SVHN), despite incorporating an additional OOD dataset during training. This suggests that the generative component improves generation quality without significantly affecting OOD detection beyond standard AT.

Compared to RATIO, our model exhibits lower OOD detection performance across most datasets. To investigate whether this gap stems from our use of milder augmentations for the generative component, we trained an ablation model that applies RATIO's aggressive augmentation strategy to both loss terms. The results show that this variant performs similarly to our standard DAT model and still underperforms RATIO. This finding indicates that the performance gap is not primarily caused by the augmentation strategy but rather by the fundamental differences in the training objectives: RATIO's loss explicitly optimizes for OOD detection, while our generative loss prioritizes learning an accurate energy function for generation. A natural extension would be combining both objectives to preserve generation quality while improving OOD detection.

Table 10: Comparison of OOD detection functions on CIFAR-10.

| Method | CIFAR-100 | | SVHN | | Uniform noise | |
|---|---|---|---|---|---|---|
| | Clean | Adversarial | Clean | Adversarial | Clean | Adversarial |
| DAT $(\max_y p_\theta(y|x))$ | 0.8709 | 0.6480 | 0.9609 | 0.8334 | 0.8922 | 0.8257 |
| DAT $(-E_\theta(x))$ | 0.8484 | 0.6647 | 0.8011 | 0.6046 | 0.9995 | 0.9983 |

Table 11: OOD detection performance (AUROC) with CIFAR-10 as ID dataset (JEM results from Augustin et al. (2020)). All methods use the maximum confidence detection function $s_\theta(x) = \max_y p_\theta(y|x)$.

| Method | CIFAR-100 | | SVHN | | Uniform noise | |
|---|---|---|---|---|---|---|
| | Clean | Adversarial | Clean | Adversarial | Clean | Adversarial |
| JEM | 0.8760 | 0.1920 | 0.8930 | 0.0730 | 0.1180 | 0.0250 |
| Standard AT | 0.8759 | 0.6364 | 0.9625 | 0.8306 | 0.8501 | 0.7902 |
| DAT ($T = 40$, uniform aug) | 0.8751 | 0.6261 | 0.9642 | 0.8303 | 0.9546 | 0.9254 |
| DAT ($T = 40$) | 0.8709 | 0.6480 | 0.9609 | 0.8334 | 0.8922 | 0.8257 |
| RATIO | 0.9157 | 0.7516 | 0.9843 | 0.9130 | 0.9999 | 0.9999 |

Table 12: OOD detection performance (AUROC) with CIFAR-100 as ID dataset.

| Method | CIFAR-10 | | SVHN | | Uniform noise | |
|---|---|---|---|---|---|---|
| | Clean | Adversarial | Clean | Adversarial | Clean | Adversarial |
| Standard AT | 0.7430 | 0.4093 | 0.8700 | 0.4863 | 0.7858 | 0.5048 |
| RATIO | 0.7320 | 0.3795 | 0.8439 | 0.4356 | 0.7769 | 0.5881 |
| DAT ($T = 45$) | 0.7027 | 0.5145 | 0.8271 | 0.5823 | 0.4024 | 0.2283 |

Table 13: OOD detection performance (AUROC) with ImageNet as ID dataset.

| Method | CIFAR-10 | | SVHN | | Uniform noise | |
|---|---|---|---|---|---|---|
| | Clean | Adversarial | Clean | Adversarial | Clean | Adversarial |
| Standard AT (ResNet-50) | 0.7235 | 0.5304 | 0.9239 | 0.8089 | 0.8678 | 0.8377 |
| DAT (ResNet-50, $T = 15$) | 0.6599 | 0.4870 | 0.8813 | 0.7754 | 0.6899 | 0.6268 |

## C.4 CALIBRATION

Figures 12 to 14 show reliability diagrams for CIFAR-10, CIFAR-100, and ImageNet, using models trained with 224×224 generation.

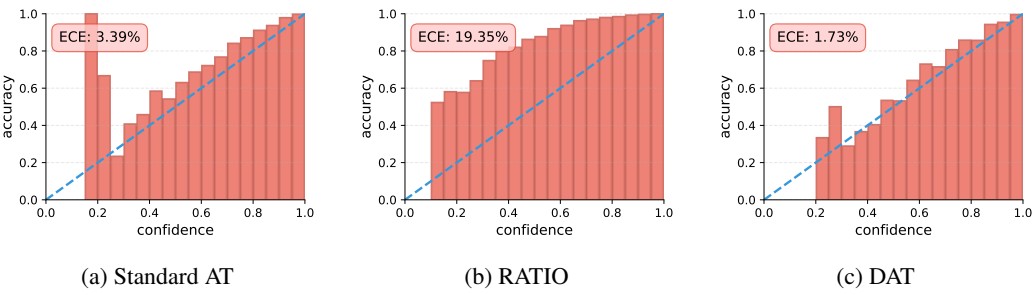

(a) Standard AT        (b) RATIO        (c) DAT

Figure 12: Calibration diagrams on CIFAR-10 (without temperature scaling).

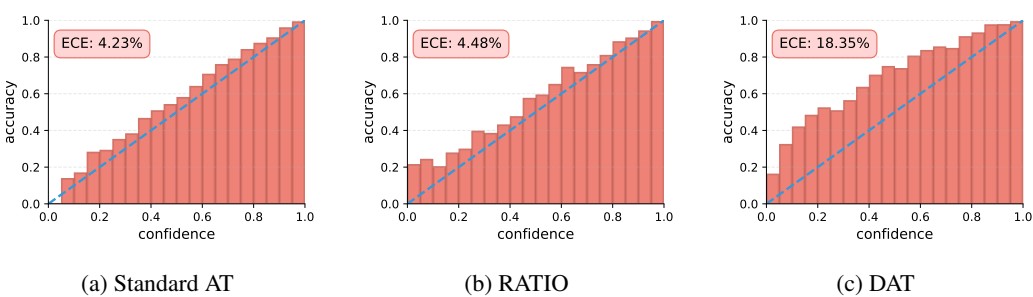

(a) Standard AT        (b) RATIO        (c) DAT

Figure 13: Calibration diagrams on CIFAR-100 (without temperature scaling).

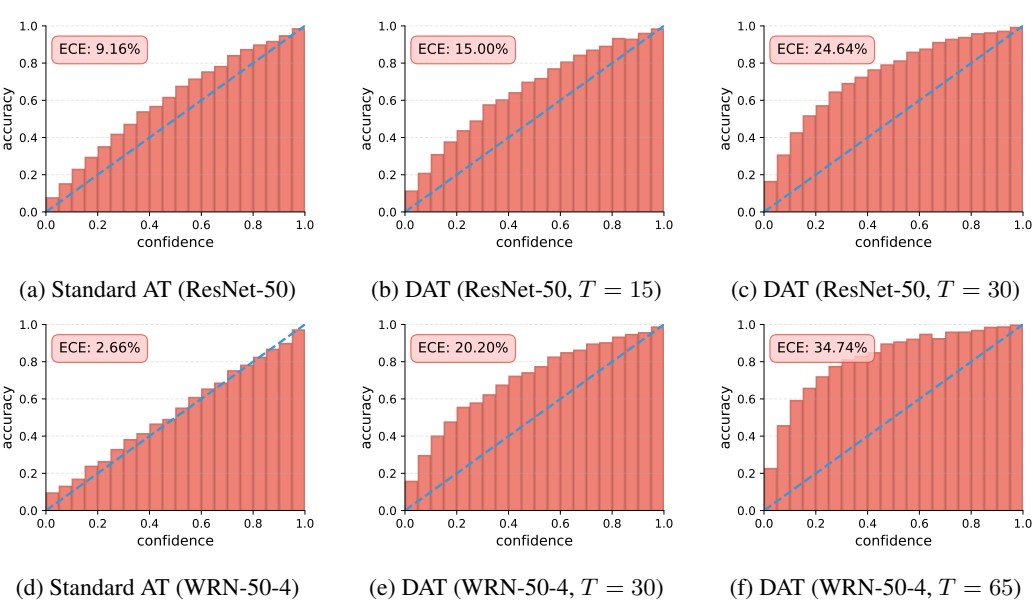

(a) Standard AT (ResNet-50)    (b) DAT (ResNet-50, $T = 15$)    (c) DAT (ResNet-50, $T = 30$)

(d) Standard AT (WRN-50-4)    (e) DAT (WRN-50-4, $T = 30$)    (f) DAT (WRN-50-4, $T = 65$)

Figure 14: Calibration diagrams on ImageNet (without temperature scaling).

## C.5 NOISE INITIALIZATION

We investigate whether DAT can be trained without any OOD data—initializing PGD from pure random noise instead of OOD images. We find that the DAT framework provides sufficient stability for training to succeed even from pure noise. Noise-initialized DAT matches OOD initialization on classification and robust accuracy and achieves reasonable generation quality, though OOD initialization still yields better FID and IS (Table 14). Figure 15 shows class-conditional samples generated from pure noise on CIFAR-10, demonstrating that the framework can produce recognizable, class-consistent images without any auxiliary dataset.

This experiment uses the same training methodology and hyperparameters as OOD initialization (Appendix B.1.1), with two modifications. To compensate for the greater distance from noise to the data manifold, we increase the PGD step size from 0.1 to 0.4 and the maximum number of PGD steps from 50 to 105. We also increase the batch size from 128 to 256, since batch size 128 induces transient classification collapse under noise initialization.

Table 14: Noise vs. OOD initialization on CIFAR-10 (WRN-34-10). Noise-initialized results are reported as mean $\pm$ standard deviation over three random seeds.

| Method | Acc% $\uparrow$ | Robust Acc% $\uparrow$ | IS $\uparrow$ | FID $\downarrow$ |
|---|---|---|---|---|
| DAT (OOD init, $T = 50$) | 90.72 | 74.65 | 9.86 | 7.57 |
| DAT (noise init, $T = 105$) | $92.05 \pm 0.68$ | $75.03 \pm 0.60$ | $8.83 \pm 0.23$ | $11.98 \pm 0.62$ |

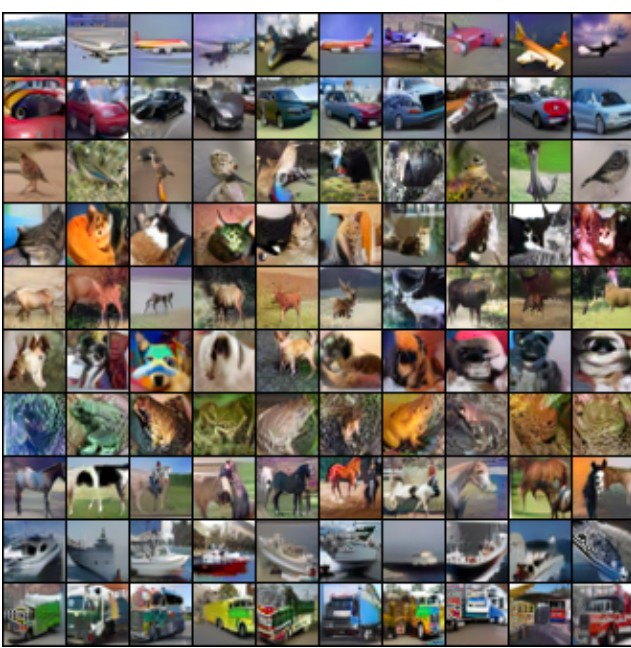

Figure 15: Random class-conditional samples generated by noise-initialized DAT on CIFAR-10 (WRN-34-10). Each row corresponds to one class. PGD is initialized from uniform noise $x_0 \sim \mathcal{U}(0, 1)$ with no auxiliary OOD dataset.

## C.6 $\ell_\infty$ GENERALIZATION

All our models in Tables 1 and 2 are trained with $\ell_2$-based adversarial attacks for both the discriminative and generative components. To demonstrate that our approach generalizes to $\ell_\infty$ adversarial training, we train on ImageNet 256×256 with ConvNeXt-L using $\ell_\infty$ attacks: for discriminative training, we use 2-step PGD with $\epsilon = 4/255$ and step size 2/255; for generative training, we use step size 3/255 with $T = 110$ maximum steps. All other hyperparameters (optimizer, learning rate, weight decay, EMA, batch size) match our $\ell_2$ configuration exactly (see Table 3). FID is evaluated on samples generated using $\ell_\infty$-based PGD with 36 steps and step size 0.03 ($\approx 8/255$).

Table 15 presents the results. Compared to Standard AT, our $\ell_\infty$-trained DAT model trades modest reductions in clean accuracy (76.58% vs 78.25%) and robust accuracy (57.94% vs 59.40%) for significantly superior generation quality (FID 4.11 vs 44.46, IS 320.7 vs 27.32). This demonstrates that DAT successfully achieves joint discriminative-generative modeling under $\ell_\infty$ training, maintaining competitive robustness while enabling high-quality generation.

Comparing our $\ell_\infty$- and $\ell_2$-trained DAT models shows each model specializes in its training norm, achieving superior robustness under that norm while maintaining comparable clean accuracy and generation quality. However, we observe a notable difference in visual quality: $\ell_2$-trained models produce smooth generated images, while $\ell_\infty$-trained models exhibit high-frequency noise artifacts (grainy appearance with scattered bright pixels). This artifact stems from the different constraint geometries—$\ell_\infty$ perturbations allow independent bounded changes to each pixel, favoring per-pixel variations, whereas $\ell_2$ perturbations enforce a global constraint that naturally penalizes high-frequency noise. These results confirm that our approach successfully generalizes to the $\ell_\infty$ setting, though the choice of norm significantly influences the perceptual quality of generated samples despite similar FID scores.

Table 15: $\ell_\infty$ and $\ell_2$ results on ImageNet 256×256 with ConvNeXt-L. Standard AT uses the $\ell_\infty$-trained ($\epsilon = 4/255$) checkpoint from Singh et al. (2023). Both DAT models initialize from the same Stage 1 checkpoint and differ only in Stage 2 perturbation norm. All models are evaluated under both $\ell_\infty$ ($\epsilon = 4/255$) and $\ell_2$ ($\epsilon = 3.0$) attacks using AutoAttack (Croce and Hein, 2020).

| Method | Training Norm | Clean Acc% ↑ | Robust Acc ($\ell_\infty$ 4/255) ↑ | Robust Acc ($\ell_2$ 3.0) ↑ | FID ↓ | IS ↑ |
|---|---|---|---|---|---|---|
| Standard AT | $\ell_\infty$ | 78.25 | 59.40 | 33.38 | 44.46 | 27.32 |
| DAT ($T = 110$) | $\ell_\infty$ | 76.58 | 57.94 | 33.40 | 4.11 | 320.7 |
| DAT ($T = 110$) | $\ell_2$ | 75.73 | 51.90 | 56.40 | 3.29 | 310.2 |

## C.7 CORRUPTION ROBUSTNESS

To evaluate whether the joint training objective maintains robustness under distribution shift, we assess our CIFAR-10 models on CIFAR-10-C (Hendrycks and Dietterich, 2019), a benchmark testing robustness to 15 common corruption types across 5 severity levels.

As shown in Table 16, DAT achieves mCE of 19.84% ($T = 40$) and 21.84% ($T = 50$), compared to 19.63% for standard AT. While corruption robustness slightly degrades as $T$ increases, DAT remains comparable to standard AT across all corruption types.

Table 16: Corruption robustness on CIFAR-10. Error rates (%) are averaged across 5 severity levels; mCE denotes mean corruption error across all 15 corruption types.

| Metric | Standard AT | DAT ($T = 40$) | DAT ($T = 50$) |
|---|---|---|---|
| *Noise corruptions (error %)* | | | |
|     Gaussian noise | 21.30 | 20.52 | 20.63 |
|     Shot noise | 17.14 | 16.36 | 16.86 |
|     Impulse noise | 23.80 | 23.54 | 24.72 |
| *Blur corruptions (error %)* | | | |
|     Defocus blur | 11.08 | 11.69 | 13.13 |
|     Glass blur | 15.38 | 14.74 | 17.06 |
|     Motion blur | 13.65 | 14.27 | 15.84 |
|     Zoom blur | 11.72 | 12.39 | 13.89 |
| *Weather corruptions (error %)* | | | |
|     Snow | 12.19 | 12.19 | 13.62 |
|     Frost | 12.44 | 12.13 | 13.60 |
|     Fog | 24.75 | 26.67 | 29.45 |
|     Brightness | 8.26 | 8.60 | 9.61 |
| *Digital corruptions (error %)* | | | |
|     Contrast | 33.31 | 32.89 | 36.24 |
|     Elastic transform | 12.34 | 13.06 | 14.80 |
|     Pixelate | 9.51 | 9.65 | 11.08 |
|     JPEG compression | 9.56 | 9.73 | 11.14 |
| Mean corruption error (mCE) ↓ | 19.63 | 19.84 | 21.84 |
| Clean Acc (%) ↑ | 92.43 | 91.86 | 90.72 |
| Robust Acc (%) ↑ | 75.73 | 75.66 | 74.65 |
| FID ↓ | 28.41 | 9.07 | 7.57 |

## C.8 Ablation studies

We ablate individual model components, analyze OOD data efficiency, and investigate the trade-off between discriminative and generative performance.

### C.8.1 Component ablation

To analyze the contribution of each component, we conduct an ablation study on CIFAR-10 with the following variants:

- Standard AT: Baseline without generative component.
- DAT with uniform augmentation: Same aggressive augmentation for both objectives.
- DAT with decoupled augmentation: Aggressive augmentation for $\mathcal{L}_{\text{AT-CE}}$, mild augmentation for $\mathcal{L}_{\text{BCE}}$.

The results in Table 17 show the impact of the AT-based generative loss and decoupled augmentation. Adding the generative loss reduces FID from 33.04 to 15.35, while robust accuracy remains comparable. Decoupled augmentation further reduces FID to 9.07.

Since both our approach and RATIO extend a standard AT baseline with an objective that leverages out-of-distribution (OOD) data, it is natural to compare their effect on generation quality. The RATIO objective, which is formulated for robust OOD detection, reduces the FID from 33.04 to 21.96. In contrast, our generative objective provides a much larger improvement, lowering the FID to 15.35. This confirms that a dedicated generative loss is more effective for sample quality than an OOD detection loss.

Table 17: Effect of generative loss and augmentation on CIFAR-10.

| Method | Acc% ↑ | Robust Acc% ↑ | FID ↓ |
|---|---|---|---|
| Standard AT | 92.34 | 75.73 | 33.04 |
| DAT (uniform aug) | 92.68 | 75.93 | 15.35 |
| DAT (decoupled aug) | 91.86 | 75.66 | 9.07 |
| RATIO (Augustin et al., 2020) | 92.23 | 76.25 | 21.96 |

### C.8.2 OOD data efficiency

The out-of-distribution (OOD) dataset is a critical component of our training framework, as it provides initialization points for generating contrastive samples in the generative loss. Intuitively, a more diverse OOD dataset provides better coverage of the input space, allowing PGD to discover a broader range of spurious modes in the energy landscape. These modes are then suppressed as training progresses.

We ablate OOD dataset size on ImageNet with DAT ResNet-50 ($T = 15$), varying from 1K to 300K samples. As shown in Table 18, FID improves modestly from 6.96 with 1K samples to 6.64 with 300K samples. Classification accuracy remains stable across all dataset sizes, with similar robustness levels, indicating that the OOD dataset size primarily affects generation quality rather than discriminative performance.

These results demonstrate notable data efficiency, with only modest improvements when scaling from 1K to 300K OOD samples. One factor is data augmentation: we employ RandomResizedCrop with scale=(0.08, 1.0) and aspect ratio=(0.75, 1.33), which can crop as little as 8% of the original image with varying aspect ratios, potentially amplifying the effective diversity of each sample. To investigate the contribution of augmentation, we include a baseline using 1K OOD samples without data augmentation. While augmentation provides clear benefits—improving FID from 8.00 to 6.96— even without augmentation, our approach substantially outperforms standard AT in generation quality (FID 8.00 vs. 15.97).

Table 18: Impact of OOD dataset size on ImageNet performance for DAT ResNet-50 ($T = 15$) with 224×224 generation.

| Method | Acc% ↑ | Robust Acc% ↑ | FID ↓ | IS ↑ |
|---|---|---|---|---|
| Standard AT | 62.83 | 34.44 | 15.97 | 274.90 |
| DAT 1K w/o aug | 57.50 | 33.80 | 8.00 | 320.64 |
| DAT 1K | 57.56 | 34.22 | 6.94 | 324.23 |
| DAT 10K | 57.82 | 34.70 | 6.84 | 320.78 |
| DAT 100K | 58.19 | 34.88 | 6.70 | 322.10 |
| DAT 300K | 57.88 | 34.84 | 6.64 | 339.55 |

### C.8.3 DISCRIMINATIVE-GENERATIVE TRADE-OFF

Our experiments—varying PGD steps $T$ (Section 4.3), loss weights (see below), and augmentation strategies (Appendix B.1.2)—reveal a trade-off between generative and discriminative performance. The augmentation ablation sheds light on this: varying only the generative augmentation, no augmentation achieves better FID but worse robust accuracy, while random cropping maintains similar FID but improves robustness. This sensitivity to the generative pipeline's data distribution suggests that model representations adapt to the data used for generative modeling. The large FID improvement of DAT over standard AT further supports this, as generation relies on gradients through the shared representation. This creates a tension: the generative objective encourages representations aligned with $p_{\text{data}}$ for high-fidelity generation, but this alignment can come at the cost of robustness. While this tension is inherent, the trade-off can be tuned through PGD iterations $T$ and loss weighting.

**Control via loss weighting.** To demonstrate this controllability, we perform experiments on CIFAR-10 with three weighting configurations for the composite objective $\mathcal{L}(\theta) = \lambda_1 \mathcal{L}_{\text{AT-CE}}(\theta) + \lambda_2 \mathcal{L}_{\text{BCE}}(\theta)$: (1) standard loss with $\lambda_1 = \lambda_2 = 1.0$; (2) emphasize generative with $\lambda_1 = 0.6, \lambda_2 = 1.4$; and (3) emphasize classification with $\lambda_1 = 1.4, \lambda_2 = 0.6$. The results in Table 19 confirm that the balance between generative and discriminative performance can be tuned by adjusting the loss term weights. Emphasizing the generative component improves FID at the cost of slightly reduced classification performance, and vice versa. Notably, our standard, unweighted loss corresponds to the natural factorization of the joint log-likelihood in the original JEM formulation: $\log p_\theta(x, y) = \log p_\theta(y|x) + \log p_\theta(x)$. This suggests that equal weighting is a principled default that performs well without requiring additional hyperparameter tuning.

Table 19: Trading off generative and discriminative performance by weighting loss terms.

| Method | Acc% ↑ | Robust Acc% ↑ | FID ↓ |
|---|---|---|---|
| Standard loss | 91.88 | 75.73 | 9.09 |
| Emphasize generative modeling | 91.16 | 75.11 | 8.77 |
| Emphasize classification | 92.52 | 75.97 | 10.02 |

## C.9 REPRODUCIBILITY AND STABILITY

Table 20 reports mean and standard deviation across five independent runs with different random seeds. Performance is consistent across all runs, with zero training divergences observed.

Table 20: Reproducibility of DAT across datasets (five runs with different random seeds).

| Dataset | Acc% ↑ | Robust Acc% ↑ | IS ↑ | FID ↓ |
|---|---|---|---|---|
| CIFAR-10 (WRN-34-10, $T = 40$) | $91.92 \pm 0.09$ | $75.75 \pm 0.07$ | $9.92 \pm 0.05$ | $9.12 \pm 0.05$ |
| CIFAR-100 (WRN-34-10, $T = 45$, LR=0.009) | $65.76 \pm 0.75$ | $45.94 \pm 0.48$ | $10.99 \pm 0.29$ | $10.73 \pm 0.25$ |
| ImageNet 256×256 (ResNet-50, $T = 15$) | $61.31 \pm 0.16$ | $39.96 \pm 0.41$ | $322.65 \pm 2.28$ | $6.87 \pm 0.05$ |

