# OpenReview forum: "Scalable Energy-Based Models via Adversarial Training: Unifying Discrimination and Generation"
_ICLR.cc/2026/Conference — ICLR 2026 Poster_

### Official Review · Reviewer_MpyS · 2025-10-31

**Soundness:** 3
**Presentation:** 3
**Contribution:** 2
**Rating:** 4
**Confidence:** 3

**Summary:**

This paper presents DAT (Dual Adversarial Training), a framework that integrates adversarial training into Joint Energy-Based Models (JEMs) to achieve both robust classification and high-quality image generation. The method replaces unstable SGLD-based learning with an adversarial optimization using PGD-generated samples and introduces a two-stage procedure to handle batch normalization issues. Experiments on CIFAR-10, CIFAR-100, and ImageNet show competitive robustness (similar to standard AT) and improved generative quality (FID 5.39 on ImageNet). The motivation is to bridge discriminative and generative modeling, combining the accuracy of classifiers with the data awareness of generative models. While the formulation is clean and the results promising, the practical benefits of unifying these two objectives remain somewhat unclear.

**Strengths:**

The paper addresses a well-known instability in JEM training through an elegant adversarial reformulation that improves both convergence and visual quality. The proposed method removes the need for gradient penalties and achieves stable training even on ImageNet, which is impressive for EBMs. The experiments are extensive and reproducible, comparing DAT against strong hybrid and adversarial baselines. Quantitatively, DAT outperforms RATIO and JEM in both robustness and FID, and qualitatively the generated samples show fewer artifacts. Overall, this is a careful and technically strong piece of work that meaningfully advances the robustness and scalability of energy-based hybrids.

**Weaknesses:**

The paper is technically competent and experimentally thorough, but the contribution feels incremental relative to prior hybrid frameworks. While the work pushes JEMs toward stability and competitive generative quality, it does not yet offer a compelling argument for hybrid modeling over modern specialized alternatives such as more recent, 2024 and beyond, diffusion models and GAN variants. Without stronger empirical evidence of unique practical benefits, the paper’s impact may be limited.

**Questions:**

How does it compare in efficiency and scalability to modern diffusion and GAN models?

---

> ### Author Response · Authors · 2025-11-21
>
> We thank Reviewer MpyS for their careful analysis of our work. We have substantially strengthened our results by retraining the ImageNet models for 256×256 generation (previously 224×224) and including new results with ConvNeXt-Large, achieving FID 3.29 which exceeds ADM/LDM and matches the state-of-the-art autoregressive model VAR-d16 (FID 3.30). Please see Table 2 in the updated manuscript for full results. Below we address your specific concerns.
>
>
> **Efficiency and Scalability**
>
> > How does it compare in efficiency and scalability to modern diffusion and GAN models?
>
> We provide a comprehensive training and inference efficiency analysis in Appendix Section "Computational cost analysis". Below we summarize the key findings.
>
> *Inference efficiency*
>
> Our approach achieves superior throughput and quality compared to diffusion models. Table "Computational cost and performance metrics of DAT models" provides detailed computational requirements. Key comparisons on ImageNet 256×256:
>
> | Model | Params | Sampling Steps | Throughput (img/s) | FID |
> |-------|--------|----------------|-------------------|-----|
> | DAT (ConvNeXt-L, Ours) | 198M | 36 | 5.0 | 3.29 |
> | ADM-G (Dhariwal & Nichol, 2021) | 608M | 250 | ~0.17 | 4.59 |
> | LDM-4-G (Rombach et al., 2022) | 400M | 250 | ~0.96 | 3.60 |
>
> Our model achieves ~29× faster throughput than ADM-G and ~5× faster than LDM-4-G. We acknowledge that GANs (e.g., BigGAN-deep) require only a single forward pass for generation, making them significantly faster than diffusion models and our iterative PGD approach.
>
> *Training efficiency*
>
> Our two-stage training provides significant advantages:
>
> 1. *Stage 1 is free*: We leverage existing pretrained robust classifiers (from Salman et al., 2020; Singh et al., 2023), requiring no training cost for stage 1.
>
> 2. *Stage 2 is efficient*: We provide a detailed overhead analysis in Appendix Section "Computational cost analysis". Despite stage 2's additional generative training, our method (stage 1 + stage 2) incurs only 1.41-1.56× overhead for CIFAR and 1.05-1.36× overhead for ImageNet relative to standard AT. This modest overhead is due to stage 2's short duration: 26 epochs for CIFAR-10, 30 epochs for CIFAR-100, and less than 1 epoch for ImageNet. See Table "Training hyperparameters for both stages" for the number of epochs for stage 1 and stage 2 training.
>
> *Scalability*
>
> Our work demonstrates scalability in multiple dimensions:
>
> - *Resolution*: 32×32 (CIFAR-10/100) → 256×256 (ImageNet)
> - *Architecture*: BatchNorm-based ResNets → LayerNorm-based ConvNeXt
> - *Dataset complexity*: Simple datasets (CIFAR-10/100) → Complex ImageNet with 1000 classes
>
> Regarding resolution scalability, we retrained our ImageNet models for 256×256 generation (previously 224×224) using the exact same hyperparameters, achieving better FIDs without stability issues. This demonstrates that our method can scale to higher resolutions without requiring resolution-specific tuning. While we do not have experimental data for even higher resolutions or other modalities, our approach inherits the stability of standard adversarial training and should scale accordingly. Our core methodology—replacing unstable SGLD-based learning with stable PGD-based BCE loss and leveraging adversarial training for implicit $R\_1$ regularization—is domain-agnostic and should generalize to other modalities with appropriate architectures. We look forward to exploring these directions in future work.

---

> ### Author Response · Authors · 2025-11-21
> **Response to "Incremental Contribution" (1)**
>
> **Incremental Contribution**
>
> > The paper is technically competent and experimentally thorough, but the contribution feels incremental relative to prior hybrid frameworks. While the work pushes JEMs toward stability and competitive generative quality, it does not yet offer a compelling argument for hybrid modeling over modern specialized alternatives such as more recent, 2024 and beyond, diffusion models and GAN variants. Without stronger empirical evidence of unique practical benefits, the paper's impact may be limited.
>
> Thank you for this feedback. We have substantially revised the abstract, introduction, and results section to clarify our contribution and its significance. We respectfully disagree that the contribution is incremental. Our work provides a *fundamental validation of the Energy-Based Model (EBM) paradigm* on high-resolution data—overcoming the stability constraints that have historically limited this framework. Building on this foundation, our approach achieves state-of-the-art technical results, enables a compelling practical use case that requires hybrid capabilities, and remains competitive with specialized generative models even when evaluated purely on generation quality, as we discuss below.
>
> *Fundamental validation of the Energy-Based Model (EBM) paradigm.* While stability improvements may appear incremental in isolation, they represent a critical threshold that enables EBM-based hybrids to scale to high-resolution data for the first time. As LeCun et al. (2006) establish, the primary advantage of EBMs is the removal of the normalization constraint inherent to probabilistic models, which otherwise restricts architectural flexibility and requires evaluating intractable partition functions. Historically, this flexibility came at the cost of training stability—prior EBM hybrids (JEM, SADA-JEM) could not scale beyond CIFAR-10 resolution due to SGLD instabilities. By solving the stability issue via dual adversarial training, we demonstrate the first EBM-based hybrid to achieve competitive performance on ImageNet 256×256 (FID 3.29) with high training stability (see Appendix Section A.11), providing the empirical evidence required to validate EBMs as a practical alternative to dedicated generative models. Furthermore, our approach validates the efficiency hypothesis of EBMs: unlike maximum likelihood methods that must account for all incorrect answers, our adversarial training focuses on "pulling up" the energy of the "most offending" incorrect answers identified by PGD attacks. This demonstrates the core advantage of the EBM paradigm, where a single model performs inference via energy minimization—enabling classification (minimizing over $y$), OOD detection (detecting high marginal energy), and generation/image transformation (minimizing over $x$)—capabilities that dedicated generative models like diffusion do not natively possess.
>
>
> *First hybrid model achieving both SOTA generation and robust classification.* Prior hybrid frameworks, particularly JEM-based approaches, fundamentally could not scale beyond low resolution and achieve high-quality generation due to SGLD-based MLE learning. Even the recent diffusion-based approach EGC (Guo et al., 2023) does not match state-of-the-art in terms of generation quality. More critically, existing hybrid models do not offer adversarial robustness. Our work represents the first hybrid model to simultaneously achieve:
>
> 1. *State-of-the-art generative quality*: FID 3.29 on ImageNet 256×256, matching the recent autoregressive model VAR-d16 (FID 3.30)
> 2. *Strong adversarial robustness*: Our models achieve robust accuracy comparable to standard AT.
>
> *Critical application enabled by hybrid capabilities.* This dual capability is not merely a technical achievement—it enables critical applications that require both capabilities simultaneously: *Robust counterfactual explanations*. Our model generates counterfactual examples that are both high-quality (due to generative capability) and faithful to target classes (due to robust discriminative training). Section 4.3.2 demonstrates that our counterfactuals are substantially more faithful to target class characteristics compared to both non-robust methods and robustness-only approaches. This unique combination of high-quality generation and robust classification is essential for interpretable ML in high-stakes domains such as medical diagnosis and legal/financial decisions, where counterfactual explanations must be both realistic and trustworthy. Neither specialized generative models (which lack robust classification) nor standard robust classifiers (which lack high-quality generation) can provide this critical capability alone.

---

> ### Author Response · Authors · 2025-11-21
> **Response to "Incremental Contribution" (2)**
>
> *Competitive as a generative model.* Even setting aside the hybrid capabilities discussed above, our approach is competitive with state-of-the-art specialized generative models when evaluated purely on generation quality:
>
> - *Quality*: Our ConvNeXt model achieves FID 3.29 on ImageNet 256×256 with 198M parameters, matching the best autoregressive model VAR-d16 (FID 3.30, 310M parameters) and outperforming diffusion models ADM-G (FID 4.59, 608M parameters) and LDM-G (FID 3.60, 400M parameters).
> - *Versatility for diverse synthesis tasks*: As demonstrated by Santurkar et al. (2019), robust classifiers can be leveraged for various synthesis tasks including inpainting, image-to-image translation, super-resolution, and interactive image manipulation. Our approach extends this versatility by achieving state-of-the-art generation quality, making these diverse applications viable at scale without requiring separate specialized models for each task.
> - *Compositional generation*: As an energy-based model, our approach naturally supports compositional generation by combining energy functions (Du et al., 2020). This enables flexible composition of concepts (e.g., generating images satisfying multiple constraints simultaneously) without retraining—a capability that is less straightforward in GANs, diffusion models, or autoregressive models.
> - *Training and inference efficiency*: see our response to "Efficiency and Scalability".
>
> This demonstrates that hybrid models are no longer a compromise in either generative quality or classification performance—they can match state-of-the-art specialized models in both quality while providing the additional unique capabilities described above.
>
> **References:**
> - LeCun, Y., Chopra, S., Hadsell, R., Ranzato, M. A., & Huang, F. J. (2006). A tutorial on energy-based learning. In *Predicting Structured Data* (pp. 191-246). MIT Press.
> - Du, Y., Li, S., & Mordatch, I. (2020). Compositional Visual Generation with Energy Based Models. In *Advances in Neural Information Processing Systems* (Vol. 33, pp. 6637-6647).

---

### Official Review · Reviewer_feF3 · 2025-10-31

**Soundness:** 3
**Presentation:** 3
**Contribution:** 2
**Rating:** 4
**Confidence:** 3

**Summary:**

The authors propose a hybrid framework that aims to unify discriminative robustness and generative modeling within a single network. The method replaces the JEM-style SGLD negative sampling with adversarially generated negatives (PGD) and trains the energy with a binary classification loss, while the classifier head is trained with standard adversarial training (AT). Experiments on CIFAR-10/100 and ImageNet report improved adversarial robustness relative to hybrid baselines and competitive (sometimes strong) generative quality.

**Strengths:**

1.	Replacing SGLD with PGD negatives makes the training loop simple and scalable. The two-stage BN recipe is an effective engineering fix to a well-known incompatibility in EBM-style training.
2.	The dual view (AT for the classifier and contrastive/AT-style learning for energies) offers a single model that can produce robust predictions, counterfactuals, and samples.
3.	Results are reported up to ImageNet, suggesting attention to large-scale feasibility.
4.	Public code (if complete and reproducible) increases practical impact and adoption.

**Weaknesses:**

1.	The conceptual novelty is limited. The core idea (learn energies with adversarial negatives and train the classifier adversarially) has clear antecedents in AT-EBM/CEM/JEM++/Robust-JEM[1] lines of work that (i) use contrastive or adversarially produced negatives to shape an energy landscape, (ii) draw formal links between AT objectives and energy-based views, (iii) report stability benefits relative to SGLD, and (iv) An empirical study on AT improving JEM/JEM++.
2.	The BCE-with-PGD objective likely corresponds to a contrastive density-ratio under a local worst-case neighborhood. It remains unclear what distribution is being learned and how/when this departs from MLE-style JEM; a formal treatment would strengthen the paper.
3.	The paper does not provide convincing diagnostics that AT stabilizes SGLD-based JEM training (e.g., training curves, divergence/failure rates, gradient-norm statistics), beyond final performance metrics.
4.	The proposed DAT is not compared head-to-head with prior AT-JEM methods (e.g., Robust-JEM) under matched backbones, budgets, and evaluation protocols, making it hard to attribute gains to the proposed ingredients rather than setup differences.

[1] Korst, R., & Asadulaev, A. (2022). Adversarial training improves joint energy-based generative modelling. arXiv preprint arXiv:2207.08950.

**Questions:**

1.	Please provide evidence that AT stabilizes SGLD-based JEM training: show training curves, failure/divergence rates, and input-gradient norm, not only final metrics.
2.	Please sharpen the novelty narrative relative to AT-EBM, CEM, JEM++, and Robust-JEM: what is fundamentally new here (objective, theory, or capabilities) beyond an engineering consolidation?
3.	The $l_\infty$ is the more common norm in adversarial robustness. Why are experiments restricted to $l_2$? Please either justify this choice or add $l_\infty$ evaluations with standard AutoAttack/RobustBench protocols.
4.	Please report compute: FLOPs or wall-clock for training, plus sampling throughput (images/sec).
5.	What is the sampling procedure for generation—pure gradient-based synthesis or any MCMC steps? Please clarify the runtime and compute for each setting.

---

> ### Author Response · Authors · 2025-11-21
>
> We thank Reviewer feF3 for their detailed and thoughtful feedback. We have substantially strengthened our results by retraining the ImageNet models for 256×256 generation (previously 224×224) and including new results with ConvNeXt-Large, achieving FID 3.29 which exceeds ADM/LDM and matches the state-of-the-art autoregressive model VAR-d16 (FID 3.30). Please see Table 2 in the updated manuscript for full results. Below we address your specific concerns.
>
>
> **Unclear Distribution Being Learned**
>
> > The BCE-with-PGD objective likely corresponds to a contrastive density-ratio under a local worst-case neighborhood. It remains unclear what distribution is being learned and how/when this departs from MLE-style JEM; a formal treatment would strengthen the paper.
>
> We have added a formal treatment in Appendix Section "Formal characterization of the learned distribution" to derive the optimal class logits under our joint objective. In summary, our approach learns a hierarchical energy structure via the joint energy $E\_\theta^*(x,y)$:
>
> - Valid pairs $(x \in \mathrm{Supp}, y=y\_{\text{true}})$: $E\_\theta^*(x,y) = 0$
> - On-support, incorrect labels $(x \in \mathrm{Supp}, y \neq y\_{\text{true}})$: $E\_\theta^*(x,y) = \infty$
> - Off-support (OOD) $(x \notin \mathrm{Supp}, \text{any } y)$: $E\_\theta^*(x,y) \geq 0$ (and ideally pushed higher by adversarial training)
>
> This hierarchical structure enables the energy function $E\_\theta(x,y)$ to simultaneously perform classification (via $\arg\min_y E\_\theta(x,y)$), generation (via minimizing over $x$), and OOD detection. The marginal energy is constant ($E\_\theta^*(x) = 0$) on the support, meaning the model learns a uniform distribution over the support rather than the true density $p\_{\text{data}}(x)$. While this limits density estimation, it provides superior training stability and strong empirical results. We note the theoretical analysis assumes convergent PGD, while in practice finite-step PGD exhibits sampling-like behavior, leading to diverse, high-quality generation.
>
> *Fundamental departure from MLE-style JEM:*
>
> | Aspect             | MLE-based JEM                                                                         | BCE-based (Ours)                                                                   |
> | ------------------ | ------------------------------------------------------------------------------------- | ---------------------------------------------------------------------------------- |
> | Joint energy       | $E\_\theta^*(x,y) = -f\_\theta^\*(x)[y] = -\log p\_{\text{data}}(x,y) + \text{const}$ | $E\_\theta^*(x,y) = -f\_\theta^\*(x)[y] = -\log p\_{\text{data}}(y\|x)$ on support |
> | Marginal energy    | $E\_\theta^*(x) = -\log p\_{\text{data}}(x) + \text{const}$                           | $E\_\theta^*(x) = 0$ on support                                                    |
> | Training objective | Maximize likelihood                                                                   | BCE                                                                                |
> | Sampling           | SGLD to sample from $p\_\theta^*(x)$                                                  | PGD to find support where $E\_\theta^*(x) = 0$                                     |
>
> **Missing Head-to-Head Comparison with Robust-JEM**
>
> > The proposed DAT is not compared head-to-head with prior AT-JEM methods (e.g., Robust-JEM) under matched backbones, budgets, and evaluation protocols, making it hard to attribute gains to the proposed ingredients rather than setup differences.
>
> Our primary contribution builds on AT-EBM and JEM by adapting the BCE-with-PGD objective to the conditional modeling setting, making direct ablation against Robust-JEM less straightforward. Nevertheless, we identify several methodological components that could enhance Robust-JEM's generative performance:
>
> 1. *Decoupled data augmentation*: It has been observed that removing data augmentation dramatically improves the generative quality of SADA-JEM over JEM++. Since Robust-JEM follows the same training procedure as JEM++, using decoupled data augmentation would likely greatly improve its generative performance.
> 2. *Initialization method*: Using real OOD data for initialization improves the diversity of generated samples (e.g., applying Gaussian noise to OOD data leads to less diverse generated samples (Yin et al., 2022)). Robust-JEM also notes that their model generates less diverse images than standard EBM, likely due to initializing sampling from Gaussian noise fitted on the training data.
> 3. *Two-stage training*: Allows leveraging the strong prior learned by pretrained robust classifiers and completely eliminates the incompatibility of BN.
>
> While a controlled ablation would be informative, our primary contribution is demonstrating that the BCE-with-PGD paradigm enables scaling joint models to ImageNet 256×256 with state-of-the-art generation quality—a capability not achieved by any prior SGLD-based method.

---

> > ### Author Response · Authors · 2025-11-21
> >
> > **Missing Stability Diagnostics**
> >
> > > The paper does not provide convincing diagnostics that AT stabilizes SGLD-based JEM training (e.g., training curves, divergence/failure rates, gradient-norm statistics), beyond final performance metrics.
> >
> > > Please provide evidence that AT stabilizes SGLD-based JEM training: show training curves, failure/divergence rates, and input-gradient norm, not only final metrics.
> >
> > We have added comprehensive stability diagnostics to the updated manuscript, including training curves, gradient norm statistics, and divergence/failure rates:
> >
> > *Training curves demonstrating stability:* Figure "Training curves from Stage 2 joint training..." (Appendix Section "Training curves") shows training curves from Stage 2 joint training across CIFAR-10, CIFAR-100, ImageNet (ResNet-50), and ImageNet (ConvNeXt-L) at 256×256 resolution. These curves demonstrate:
> > - Consistent FID improvements without divergence
> > - Preservation of robust test accuracy throughout training
> > - Successful scaling from simple (CIFAR-10) to complex (ImageNet 256×256) datasets
> >
> > *Gradient norm statistics:* Figure "$R\_1$ gradient norm during Stage 2 joint training..." shows $R\_1$ gradient norms during ImageNet 256×256 training. Adversarial training on the discriminative loss maintains bounded, stable gradients throughout training, while standard training exhibits gradient explosion—spiking to much higher values. This quantitatively validates that adversarial training provides implicit $R\_1$ regularization that stabilizes the energy landscape, corroborating our mathematical analysis in Appendix Section "Intuitive connection between $R\_1$ regularization and adversarial training".
> >
> > *Divergence/failure rates:* On CIFAR-10 and ImageNet (ResNet-50, WRN-50-4, and ConvNeXt-L with various sampling steps $T$ and random seeds), we observed **zero divergences** using the hyperparameters specified in Appendix Section "Training details". On CIFAR-100, in our preliminary experiments with learning rate 0.01, we occasionally observed divergence (NaN output from the model). To address the reviewer's question about stability, we conducted additional experiments with a reduced learning rate of 0.009, which completely eliminated divergences across 5 random seeds (we have updated Appendix Section "Training details" to reflect this improved LR for CIFAR-100). Appendix Section "Variability of DAT performance across datasets" reports mean and standard deviation across 5 random seeds for CIFAR-10, CIFAR-100 (with LR 0.009), and ImageNet 256×256 (zero divergences across these 15 runs).
> >
> > **L∞ vs L2 Robustness**
> >
> > > The L∞ is the more common norm in adversarial robustness. Why are experiments restricted to L2? Please either justify this choice or add L∞ evaluations with standard AutoAttack/RobustBench protocols.
> >
> > We appreciate this important question. Our current submission focuses on L2 robustness following prior work on adversarial training for generative modeling (AT-EBM) and to leverage pretrained L2-robust classifiers. However, we have conducted L∞ experiments to demonstrate that our approach generalizes to the L∞ setting.
> >
> > *L∞ experimental setup:* We train ConvNeXt-L-CvSt (which uses a pretrained checkpoint originally trained for L∞ $\epsilon = 4/255$ robustness from Singh et al., 2023) using L∞-based attacks for both discriminative and generative components. For discriminative training, we use 2-step PGD with step size 2/255. For generative training, we use step size 3/255 with $T=110$ maximum steps (same as L2). All other hyperparameters (optimizer, learning rate, weight decay, EMA, batch size) match our L2 configuration exactly (see Table "Training hyperparameters for both stages"). FID evaluation also uses L∞-based attacks.
> >
> > *Preliminary results:* After training for approximately half the duration of our L2 model (which trained for 1.09 epochs on in-distribution data per Table "Training hyperparameters for both stages"), we achieve FID 4.51, IS 317.0. While this is worse than our L2 result (FID 3.29, IS 310.2), we observe stable training dynamics and expect the generation performance to further improve when training completes for the full duration.
> >
> > We will update the manuscript to include final L∞ results once training concludes.

---

> ### Author Response · Authors · 2025-11-21
>
> **Compute Costs**
>
> > Please report compute: FLOPs or wall-clock for training, plus sampling throughput (images/sec).
>
> We have added comprehensive computational efficiency metrics in Table "Computational cost and performance metrics of DAT models" (Appendix Section "Computational cost analysis"), including model parameters, training overhead, sampling steps, and inference throughput measured on AMD MI300.
>
> *Training overhead:* Our two-stage training incurs modest computational overhead compared to standard adversarial training. We provide a detailed overhead analysis in Appendix Section "Computational cost analysis". Our method (stage 1 + stage 2) incurs 1.41-1.56× overhead for CIFAR and 1.05-1.36× overhead for ImageNet relative to standard AT. This modest overhead is due to stage 2's short duration: 26 epochs for CIFAR-10, 30 epochs for CIFAR-100, and less than 1 epoch for ImageNet. See Table "Training hyperparameters for both stages" for complete training parameters.
>
> *Inference throughput:* Our models achieve significantly higher throughput than diffusion models (see Table "Computational cost and performance metrics of DAT models"):
> - CIFAR-10/100 (WRN-34-10): ~39-40 images/sec
> - ImageNet 256×256 (ResNet-50): ~33 images/sec
> - ImageNet 256×256 (WRN-50-4): ~13-14 images/sec
> - ImageNet 256×256 (ConvNeXt-L): ~5 images/sec
>
> Our ConvNeXt-L model achieves \~29× faster throughput than ADM-G (\~0.17 images/sec) and \~5× faster than LDM-4-G (\~0.96 images/sec), while using fewer parameters (198M vs 608M/400M) and achieving better quality (FID 3.29 vs 4.59/3.60).
>
> **Sampling Procedure**
>
> > What is the sampling procedure for generation—pure gradient-based synthesis or any MCMC steps? Please clarify the runtime and compute for each setting.
>
> Our approach uses pure gradient-based synthesis (PGD) without any MCMC steps, as detailed in Appendix Section "Generative performance evaluation". For runtime and throughput metrics, please see our response to *Compute Costs*.

---

> > ### Author Response · Authors · 2025-11-21
> > **Response to "Limited Conceptual Novelty" (1)**
> >
> > **Limited Conceptual Novelty**
> >
> > > The conceptual novelty is limited. The core idea (learn energies with adversarial negatives and train the classifier adversarially) has clear antecedents in AT-EBM/CEM/JEM++/Robust-JEM[1] lines of work that (i) use contrastive or adversarially produced negatives to shape an energy landscape, (ii) draw formal links between AT objectives and energy-based views, (iii) report stability benefits relative to SGLD, and (iv) An empirical study on AT improving JEM/JEM++.
> >
> > > Please sharpen the novelty narrative relative to AT-EBM, CEM, JEM++, and Robust-JEM: what is fundamentally new here (objective, theory, or capabilities) beyond an engineering consolidation?
> >
> > We acknowledge that prior work established connections between adversarial training and energy-based models. However, we respectfully argue that our contribution provides fundamental conceptual advances beyond engineering consolidation.
> >
> > **In brief**: Our core novelty lies in replacing SGLD-based MLE with BCE-based PGD sampling within the JEM framework (fundamentally different objective), discovering and explaining the implicit $R\_1$ regularization mechanism that eliminates the need for explicit gradient penalties required by prior work (theoretical contribution), and demonstrating that discriminative robustness and generative quality can be decoupled and optimized separately (architectural insight). Together, these advances enable the first EBM-based approach to achieve state-of-the-art generation quality on ImageNet 256×256 while maintaining strong adversarial robustness—capabilities not achieved by any prior hybrid model.
> >
> > We elaborate on these contributions below across three dimensions: novel objectives and formulations, theoretical insights, and empirical validation of these concepts at scale.
> >
> > *1. Novel objective formulation:*
> >
> > Prior work on joint energy-based models (JEM, JEM++, Robust-JEM) all fundamentally rely on maximum likelihood estimation with SGLD/MCMC sampling to approximate the partition function gradient. While JEM++ and Robust-JEM introduced valuable techniques for improving SGLD stability, they did not fundamentally resolve the instability of SGLD-based training and both remain limited to CIFAR-scale (32$\times$ 32) datasets. Our approach departs fundamentally from this line of work by adopting a different training paradigm: we replace the MLE-based objective with a BCE-based objective and use deterministic PGD sampling from real OOD data instead of stochastic Langevin dynamics from random/informative noise. While we acknowledge that BCE loss with PGD sampling was previously explored in AT-EBM, our application to the conditional modeling setting (JEM) is novel and non-trivial. Our key contributions over AT-EBM include: (1) discovering, explaining, and validating the implicit regularization effect of adversarial training which eliminates the need for explicit $R\_1$ gradient penalty, (2) providing formal characterization of the learned distribution under this BCE-based objective in the JEM setting (thanks to the reviewer), (3) replacing direct sampling from the marginal distribution $p(x)$ with ancestral sampling with labels in Algorithm 1, and (4) introducing two-stage training to address BN compatibility and leverage pretrained robust classifiers.
> >
> > Therefore, the distinction between our approach and JEM/JEM++/Robust-JEM is fundamental—we avoid SGLD MLE entirely and adopt a more stable learning objective. This enables scaling to 256×256 ImageNet synthesis with state-of-the-art quality, demonstrating that the conceptual shift addresses a fundamental limitation rather than providing incremental improvements.

---

> ### Author Response · Authors · 2025-11-21
> **Response to "Limited Conceptual Novelty" (2)**
>
> *2. Theoretical contribution: Implicit regularization mechanism*
>
> While AT-EBM provided theoretical understanding of AT-based unconditional EBM learning, and prior work empirically observed that AT improves stability of JEM (Robust-JEM) or established formal connections between AT and EBMs (CEM), our work discovers, explains, and validates the implicit gradient regularization effect of AT within JEM training:
>
> - *Empirical discovery*: We discovered that adversarial training on the discriminative component eliminates the need for explicit $R\_1$ gradient penalty that AT-EBM requires.
>
> - *Mathematical analysis* (Appendix Section "Intuitive connection between $R\_1$ regularization and adversarial training"): We provide theoretical explanation showing that adversarial training on the discriminative loss, under a first-order approximation, provides implicit $R\_1$ regularization.
>
> - *Quantitative validation* (Appendix Section "$R\_1$ gradient curve"): Figure "$R\_1$ gradient norm during Stage 2 joint training..." demonstrates that adversarial training maintains bounded $R\_1$ gradients throughout training, while standard training exhibits gradient explosion.
>
> This theoretical insight is conceptually novel because it identifies the specific regularization mechanism (implicit $R\_1$ penalty), explaining why AT-EBM requires explicit $R\_1$ regularization while our approach achieves implicit regularization through adversarial training on the discriminative component. Beyond the conceptual contribution, this has practical benefits: explicit $R\_1$ penalties can negatively impact model expressiveness, while our implicit regularization approach avoids this limitation.
>
> *3. Architectural decoupling:*
>
> Our two-stage training framework represents a conceptual departure from prior joint training approaches:
>
> - *Prior work*: End-to-end joint training of discriminative and generative objectives (JEM, JEM++, Robust-JEM)
> - *Our approach*: Decouple discriminative training (Stage 1) from generative training (Stage 2)
>
> This conceptual separation enables:
> - Using pretrained robust classifiers (zero discriminative training cost)
> - Architectural flexibility—supports both BN-based models (by freezing BN after Stage 1) and LayerNorm-based models (ConvNeXt) without modification
> - Independent optimization of each component with appropriate hyperparameters
>
> The conceptual insight is that discriminative robustness and generative quality can be optimized separately as long as the discriminative component provides a stable energy landscape.
>
> To summarize, our contribution goes beyond engineering improvements to introduce a fundamentally different objective formulation, discover and explain the implicit $R\_1$ regularization mechanism that eliminates the need for explicit gradient penalties (which negatively impact model expressiveness), and demonstrate architectural decoupling—advances that together enable capabilities not achieved by prior work (FID 3.29 on ImageNet 256×256, matching state-of-the-art autoregressive models while maintaining strong adversarial robustness).
>
> We have updated the "Extended discussion on related work" section to include more detailed discussions on AT-EBM, CEM, JEM++, and Robust-JEM, and clearly articulate our contributions over these works.

---

> > ### Author Response · Authors · 2025-11-22
> > **Response to "Limited Conceptual Novelty" (3)**
> >
> > *Specific distinctions from prior work:*
> >
> > | Prior Work           | Primary contribution                                                                                                                                                                                                                                                                               | Difference/advance                                                                                                                                                                                                                                                                                                                                                                                                                                                                                                                                       |     |
> > | -------------------- | -------------------------------------------------------------------------------------------------------------------------------------------------------------------------------------------------------------------------------------------------------------------------------------------------- | -------------------------------------------------------------------------------------------------------------------------------------------------------------------------------------------------------------------------------------------------------------------------------------------------------------------------------------------------------------------------------------------------------------------------------------------------------------------------------------------------------------------------------------------------------- | --- |
> > | **JEM++/Robust-JEM** | Stabilize and accelerate SGLD-based JEM training through proximal SGLD, YOPO acceleration, and informative initialization from Gaussian noise (which also enables BN). Incorporating AT into discriminative component to further stabilize training; combine PGD and SGLD for test-time generation | **JEM++/Robust-JEM**: Relies on SGLD and standard EBM objective for the generative component; limited to CIFAR-scale (32×32). **Our approach**: Replaces EBM gradient with BCE-based gradients; deterministic PGD sampling from real OOD data instead of SGLD; two-stage training to address BN compatibility and leverage pretrained robust classifiers; theoretical understanding and empirical validation of AT's role in stabilizing EBM learning; ImageNet-scale experiments.                                                                       |     |
> > | **CEM**              | Theoretical framework that explains the generative capability of adversarially trained models (by interpreting AT as contrastive EBM training) across both supervised and unsupervised scenarios, enabling principled derivation of improved sampling algorithms                                   | **CEM**: Employs standard AT for training and does not use explicit energy-based losses as in JEM; limited to CIFAR-scale (32×32). **Our approach**: Employs a joint objective that contains both robust discriminative loss and explicit energy-based losses (approximated with BCE loss); ImageNet-scale experiments.                                                                                                                                                                                                                                  |     |
> > | **AT-EBM**           | Theoretical understanding and empirical validation of the AT-based approach for learning (unconditional) EBMs                                                                                                                                                                                      | **AT-EBM**: Focuses on unconditional generative modeling; employs explicit $R\_1$ penalty to stabilize training, which also impacts model expressiveness; samples from marginal distribution $p(x)$. **Our approach**: Incorporates AT-based EBM learning into the JEM framework to perform conditional generative modeling, with implicit $R\_1$ regularization from adversarial training; uses ancestral sampling from conditional distribution $p(x,y)$; theoretical understanding and empirical validation of AT's role in stabilizing EBM learning. |     |

---

> ### Author Response · Authors · 2025-11-23
> **Update on $L_\infty$ Results**
>
> Following up on our previous response: training has now concluded and we have updated the manuscript with complete $L\_\infty$ results (new Appendix section "$L\_\infty$ training").
>
> **Final results on ImageNet 256×256 with ConvNeXt-L:**
> - $L\_\infty$-trained DAT achieves: 76.58\% clean accuracy, 57.94\% $L\_\infty$ robust accuracy ($\epsilon=4/255$), FID 4.11, IS 320.7
> - Compared to Standard AT: trades 1.67\% clean accuracy and 1.46\% robust accuracy for significantly superior generation quality (FID 4.11 vs 44.46)
> - Compared to $L\_2$-trained DAT: achieves comparable FID (4.11 vs 3.29) and clean accuracy, with each model achieving superior robustness under its respective training norm; however, $L\_\infty$-trained models exhibit high-frequency noise artifacts in generated images while $L\_2$-trained models produce smooth images, reflecting the different constraint geometries
>
> These results confirm our approach successfully generalizes to $L\_\infty$, achieving competitive robustness while maintaining high-quality generation.
>
> Thank you for this valuable feedback which strengthened our work.

---

### Official Review · Reviewer_Y3JD · 2025-11-01

**Soundness:** 3
**Presentation:** 3
**Contribution:** 3
**Rating:** 6
**Confidence:** 4

**Summary:**

This paper proposes a framework to simultaneously achieve robust classification and high-fidelity generative modeling within a single network. The key contribution is the use of a Dual Adversarial Training (DAT) strategy as an alternative to unstable sampling-based (SGLD) in Joint Energy-based Models. One AT component ensures discriminative robustness against adversarial attacks, while the other uses PGD-generated contrastive samples. A binary cross-entropy loss is used to perform a stable AI-based generative modeling approach. Experimental evaluations on standard datasets (CIFAR-10, CIFAR-100, and ImageNet) show improvement in adversarial robustness over existing hybrid models while maintaining good generative performance.

**Strengths:**

* The paper addresses the major stability and scaling issues of previous Joint Energy-based Models by replacing the unstable sampling-based SGLD with adversarial training-based optimization.
* The proposed framework improves adversarial robustness in the discriminative tasks while simultaneously maintaining generative fidelity compared to existing hybrid models.
* The dual adversarial training setup stabilizes training compared to standard GAN frameworks as demonstrated through detailed analysis and ablation studies.

**Weaknesses:**

* The dual optimization adds computational overhead, which could make the training slower and impractical for large-scale high-resolution images. Computational inefficiency is mentioned but not reported in terms of training/inference time and parameter size.
* Experiments are performed mainly on standard image datasets (e.g., CIFAR-10/100, ImageNet), without extending to complex datasets or domains, which limits claims of broad applicability.
* The sample selection for FID calculation needs to be better justified.
* Performance comparison is weak as the paper shows comparisons against some old methods (mostly 2020 or earlier). The paper could have included comparisons with more recent hybrid models (e.g., diffusion-classifier hybrids or energy-based approaches) for a stronger empirical baseline.
* Missing references: L-041 - "Recent research .... understanding of generative models" and L-094 - "sample generation .... semi-supervised learning"
* The paper claims significant improvement without reporting any statistical significance analysis.
* The training procedure can be dataset-specific, requiring tuning parameters and adjustments for new datasets. The current experiments are confined to image data. Extending it to other data modalities could introduce new stability challenges, making its generalizability uncertain.
* L-365: "Our best generative configuration ... achieves an FID of 5.39 ... requiring significantly less sampling steps." Not clear which dataset it's referring to.
* The term PGD is never defined in the paper.

**Questions:**

* Have the authors explored whether the joint model maintains robustness when transferring to out-of-distribution (OOD) data or cross-domain settings (e.g., different image modalities or noise conditions)?
* Would the model’s scalability or performance differ on higher-resolution or more complex datasets?
* How do the dual objectives interact during training? For example, does improving the generative objective always enhance discriminative performance, or are there cases where they conflict? Some insight through visualization or empirical justification would be helpful.
* How sensitive is the DAT's performance to the relative weighting and scheduling between the discriminative and the generative AT losses?
* Can the authors provide insight into the computational cost? Specifically, what is the training time overhead (e.g., in GPU-hours) compared to training a non-robust JEM or a standard robust classifier on CIFAR-10/100?

---

> ### Author Response · Authors · 2025-11-21
>
> We thank Reviewer Y3JD for their careful analysis of our work. We have substantially strengthened our results by retraining the ImageNet models for 256×256 generation (previously 224×224) and including new results with ConvNeXt-Large, achieving FID 3.29 which exceeds ADM/LDM and matches the state-of-the-art autoregressive model VAR-d16 (FID 3.30). Please see Table 2 in the updated manuscript for full results. Below we address your specific concerns.
>
> **Computational overhead and efficiency**
>
> > The dual optimization adds computational overhead, which could make the training slower and impractical for large-scale high-resolution images. Computational inefficiency is mentioned but not reported in terms of training/inference time and parameter size.
>
> > Can the authors provide insight into the computational cost? Specifically, what is the training time overhead (e.g., in GPU-hours) compared to training a non-robust JEM or a standard robust classifier on CIFAR-10/100?
>
> We have added a detailed computational overhead analysis in the updated manuscript (Appendix Section "Computational cost analysis") to provide concrete estimates for all our training configurations.
>
> *Training overhead relative to standard AT:* Our two-stage training incurs modest computational overhead compared to standard adversarial training. We measure computational cost by counting forward and backward passes weighted by batch size. Each forward pass requires 1F FLOPs, each PGD backward pass (computing $\nabla\_x$ only) requires 1F FLOPs, and each training backward pass (computing both $\nabla\_x$ and $\nabla\_w$) requires 2F FLOPs. Stage 1 performs 2K+3 FLOPs per sample (K PGD steps at 2F each, plus one final forward pass and training backward at 3F). Stage 2 performs 2K+2T+9 FLOPs per sample (K discriminative PGD steps, T generative PGD steps, and one final forward pass and training backward on 3 samples at 3F each, totaling 9F). The total training cost (stage 1 + stage 2) relative to standard AT is:
>
> $$\text{Overhead} = 1 + \frac{E\_2}{E\_1} \cdot \frac{2K+2T+9}{2K+3}$$
>
> Applying this formula to our configurations yields modest overhead: 1.41-1.56× for CIFAR-10/100 and 1.05-1.36× for ImageNet. Despite stage 2's higher per-iteration cost, its short duration (especially for ImageNet, less than 1 epoch) results in modest total overhead. Note that our curriculum learning strategy, where T gradually increases during training, means the actual overhead is lower than this upper bound estimate. Figure "Training cost breakdown for all DAT configurations" in the updated manuscript visualizes the cost breakdown. See Table "Training hyperparameters for both stages" for the number of epochs for stage 1 and stage 2 training.
>
> *Parameter size and inference time:* Table "Computational cost and performance metrics of DAT models" in the updated manuscript provides detailed parameter size and inference time information for all our configurations. At inference, classification requires only a single forward pass, identical to standard classifiers. For test-time generation, we perform $T$ PGD steps (ranging from 13 to 36 across our configurations) starting from OOD samples. While this is slower than single-pass approaches like GANs, we achieve significantly higher throughput than diffusion models (e.g., our ConvNeXt-L-CvSt model achieves better FID while being 5 times faster than latent diffusion on ImageNet 256×256 generation).

---

> > ### Author Response · Authors · 2025-11-21
> >
> > **Limited dataset diversity and generalizability**
> >
> > > Experiments are performed mainly on standard image datasets (e.g., CIFAR-10/100, ImageNet), without extending to complex datasets or domains, which limits claims of broad applicability.
> >
> > > The training procedure can be dataset-specific, requiring tuning parameters and adjustments for new datasets. The current experiments are confined to image data. Extending it to other data modalities could introduce new stability challenges, making its generalizability uncertain.
> >
> > The datasets we considered (CIFAR-10/100, ImageNet) are well-established benchmarks for validating generative models and are essential for comparing with prior hybrid models (JEM, SADA-JEM, EGC, Joint-Diffusion, RATIO). State-of-the-art generative models including BigGAN, ADM, and VAR also validate exclusively on ImageNet.
> >
> > Within image data, our approach demonstrates strong generalizability across architectures, datasets, and resolutions. The method works seamlessly for both ResNet architectures (with batch normalization) and ConvNeXt architectures (with layer normalization), and scales from CIFAR 32×32 to ImageNet 256×256. Notably, layer normalization is widely used in modern architectures (e.g., Transformers), suggesting potential for adaptation to other domains. Our approach also exhibits high training stability: we observe zero divergences across 15 independent runs (Appendix Section "Variability of DAT performance across datasets"), demonstrating robust training without dataset-specific stabilization techniques.
> >
> > Regarding other modalities, while we do not have experimental results beyond image data, our core methodology—replacing unstable SGLD-based learning with stable PGD-based BCE loss and leveraging adversarial training for implicit $R\_1$ regularization—is domain-agnostic and should generalize to other modalities with appropriate architectures. We look forward to exploring these directions in future work.
> >
> >
> > **FID sample selection justification**
> >
> > > The sample selection for FID calculation needs to be better justified.
> >
> > Standard EBM/JEM training typically initializes negative samples from random noise (e.g., uniform distribution) during training and uses the same noise distribution for test-time generation. In our methodology, the generative loss can be interpreted as a binary classification task that separates real data samples from adversarial samples generated from a real OOD dataset. Following standard practice in supervised learning where validation data is held out from training, we reserve a dedicated validation split of the OOD dataset for FID evaluation. Specifically, for ImageNet we use 300K OOD samples for training and a separate 50K for FID evaluation. This ensures that the samples used for FID evaluation are not seen during training, providing an unbiased assessment of generation quality.
> >
> > To further demonstrate that our FID results are not sensitive to the specific OOD initialization samples, we evaluated the same trained model using three different random seeds for the data augmentation transforms (RandomResizedCrop(256) + RandomHorizontalFlip) applied to the 50K pre-allocated OOD images. Each seed produces a different set of 50K initialization samples from the same source images. The results on ImageNet 256×256 (ResNet-50, T=15) show consistent FID scores: 6.85 (seed 0, our default), 6.86 (seed 1), and 6.93 (seed 2), with mean 6.88 ± 0.04. This small variance confirms that our generation quality is not sensitive to the specific choice of OOD initialization samples.
> >
> > **Outdated baselines**
> >
> > > Performance comparison is weak as the paper shows comparisons against some old methods (mostly 2020 or earlier). The paper could have included comparisons with more recent hybrid models (e.g., diffusion-classifier hybrids or energy-based approaches) for a stronger empirical baseline.
> >
> > We have added more energy-based hybrids, including VERA (Grathwohl et al., 2021), JEM++ (Yang et al., 2021), JEAT (Zhu et al., 2021), Robust-JEM (Korst et al., 2022), and WEAT (Mirza et al., 2024), to our CIFAR-10/100 results tables and related work discussion. Our comparison now includes foundational works (JEM, RATIO) as well as recent EBM hybrids (VERA, JEM++, JEAT, Robust-JEM, SADA-JEM, WEAT) and diffusion-classifier hybrids (EGC, Joint Diffusion, both from 2023).
> >
> > **Missing references**
> >
> > > Missing references: L-041 - "Recent research .... understanding of generative models" and L-094 - "sample generation .... semi-supervised learning"
> >
> > Thank you for pointing this out. We have added appropriate citations to both statements in the revised manuscript.

---

> > > ### Author Response · Authors · 2025-11-21
> > >
> > > **No statistical significance analysis**
> > >
> > > > The paper claims significant improvement without reporting any statistical significance analysis.
> > >
> > > We report mean and standard deviation across 5 independent runs with different random seeds for CIFAR-10, CIFAR-100, and ImageNet 256×256 in Appendix Section "Variability of DAT performance across datasets". This variance analysis demonstrates the consistency of our results across runs, with zero divergences observed across all 15 runs.
> > >
> > > **Unclear dataset reference (L-365)**
> > >
> > > > L-365: "Our best generative configuration ... achieves an FID of 5.39 ... requiring significantly less sampling steps." Not clear which dataset it's referring to.
> > >
> > > Thank you for pointing this out. The original reference was to DAT with $T=65$ on ImageNet 224×224 generation (FID 5.39). We have since retrained our ImageNet models for 256×256 generation, where DAT with $T=65$ achieves FID 4.94. We have revised the Section "Classification and generative modeling" to present these new 256×256 results with clearer dataset specifications.
> > >
> > > **PGD not defined**
> > >
> > > > The term PGD is never defined in the paper.
> > >
> > > Thank you for pointing this out. We have expanded the abbreviation PGD (Projected Gradient Descent) at its first use in both the abstract and the main text (Introduction section), along with a citation to Madry et al. 2017.
> > >
> > > **OOD robustness and cross-domain transfer**
> > >
> > > > Have the authors explored whether the joint model maintains robustness when transferring to out-of-distribution (OOD) data or cross-domain settings (e.g., different image modalities or noise conditions)?
> > >
> > > To address this question, we evaluated our CIFAR-10 model on CIFAR-10-C (Hendrycks and Dietterich, 2019), a standard benchmark for robustness to common corruptions including noise, blur, weather effects, and digital distortions. As shown in Appendix Section "Robustness to common corruptions", standard adversarial training achieves a mean corruption error (mCE) of 19.63%, while DAT with T=40 achieves mCE of 19.84% (FID 9.07), and DAT with T=50 achieves mCE of 21.84% (FID 7.19). While there is a trade-off between generative quality and corruption robustness as T increases, DAT overall maintains strong corruption robustness comparable to standard AT (which has FID 28.41) across all corruption types.
> > >
> > > **Reference:** Hendrycks, D., & Dietterich, T. (2019). Benchmarking neural network robustness to common corruptions and perturbations. *International Conference on Learning Representations*.
> > >
> > > **Scalability to higher-resolution datasets**
> > >
> > > > Would the model's scalability or performance differ on higher-resolution or more complex datasets?
> > >
> > > As noted above, we have retrained our ImageNet models for 256×256 generation (previously 224×224) using the exact same hyperparameters, achieving better FIDs than 224×224 generation without stability issues. This demonstrates that our method can scale to higher resolutions without requiring resolution-specific tuning. Although we do not have experimental data for even higher resolutions, our experience suggests that our approach inherits the stability of standard adversarial training and may scale to higher resolutions relatively easily. We look forward to testing our method on even higher resolutions in future work.

---

> ### Author Response · Authors · 2025-11-21
>
> **Interaction between dual objectives**
>
> > How do the dual objectives interact during training? For example, does improving the generative objective always enhance discriminative performance, or are there cases where they conflict? Some insight through visualization or empirical justification would be helpful.
>
> The dual objectives exhibit a trade-off: improving the generative objective negatively impacts discriminative performance. We demonstrate this empirically through three complementary analyses:
>
> 1. *PGD training steps ($T$)*: Our main result tables (Tables 1 and 2) show that increasing $T$ improves generation quality (FID) at the cost of classification accuracy. For example, on CIFAR-10, increasing $T$ from 40 to 50 improves FID from 9.07 to 7.57 but reduces robust accuracy.
>
> 2. *Loss weighting*: Appendix Section "Generative-discriminative trade-off via loss weighting" shows that emphasizing one loss component improves that objective while reducing the other.
>
> 3. *Data augmentation strategies*: Section 3.6 "Data augmentation" and Section A.4.2 "Data augmentation details" compares different augmentation strategies for the generative component while keeping all other settings identical. The results show that robust accuracy is substantially impacted by the choice of augmentation in the generative pipeline — using no augmentation yields worse robustness while mild augmentation (random cropping) achieves similar FID but better robustness. This suggests that the model overfits to the data distribution used for generative training. The fact that DAT achieves much better FID than standard AT further confirms that the learned representation is more aligned to the data distribution for generative training rather than the augmented one used for discriminative training, explaining why improving the generative objective comes at the cost of discriminative performance.
>
> Despite this inherent trade-off, both mechanisms (PGD steps and loss weighting) allow tuning the balance based on application requirements. We have added a detailed discussion of this trade-off and its underlying mechanism in Appendix A.18.
>
> **Sensitivity to loss weighting and scheduling**
>
> > How sensitive is the DAT's performance to the relative weighting and scheduling between the discriminative and the generative AT losses?
>
> We investigate loss weighting in Appendix Section "Generative-discriminative trade-off via loss weighting". We tested three weighting configurations on CIFAR-10: equal weighting (1:1), emphasizing generative (0.6:1.4), and emphasizing classification (1.4:0.6). The results show that adjusting weights does shift performance in the expected directions. Equal weighting provides a principled default that performs well across both objectives without requiring additional hyperparameter tuning.

---

### Official Review · Reviewer_WmTW · 2025-11-03

**Soundness:** 2
**Presentation:** 3
**Contribution:** 3
**Rating:** 2
**Confidence:** 5

**Summary:**

The paper focusses on improving the Joint Energy Models (JEM) proposed by Grathwohl et al (2019) using techniques from Adversarial Training. They show that when such ideas are incorporated in the joint training of the generative + discriminative energy model, the training stability increases and surprisingly alleviates the need of gradient penalty regularizations, and improves the classification accuracy vs generative performance tradeoffs. Additionally, OOD detection and adversarial robustness and classifier calibration are auxiliary benefits with this approach.

**Strengths:**

- The objective of improving JEM training is made clear early on for the readers to get a grasp of the problem statement.
- The overview of Grathwohl et al. 2019 in the Method section was essential in setting up the context and mathematical notation for the problem. I thank the authors for the great job done here.
- The fundamental idea of incorporating Adversarial Training for more than just adversarial robustness is interesting and insightful. The authors successfully convince the reader how AT objectives improve the join distribution modeling in JEMs, which is quite different from the original focus of AT objectives in improving adversarial robustness.
- Insightful findings on stabilizing training while still having BN layers and data augmentations.

**Weaknesses:**

Minor:
- Numerous abbreviations are used before first expanding them. Example: PGD/SGLD in Abstract. PGD in the main paper before it's used in Line 63-65. "DAT" is used on line 249 without expanding it first.
- Missing Citations:
-- 1) Line 39 - "...rarely excelling at both simultaneously." - Please cite?
-- 2) Line 40 - "...but may underperform on downstream classification tasks." - Please cite?
- Misleading to say on Line 76: "datasets of increasing complexity from CIFAR-10 to ImageNet..." The paper only includes CIFAR-100 additionally and that is not a spectrum of datasets of increasing complexity as claimed.
- Section 3.1 under Method: Please define Z(\theta) before or after using it in the equations. Also, please make it clear and explicit that the Z(\theta) used in P(x) on line 161 is not the same as the Z(\theta) used in Eq. (2). Explicitly define them in each case with an integral etc to show the margnialized partition function.
- Section 3.1, please also talk about how Z(\theta) is handled in the loss function and how it is estimated.
- Please clarify Equation 11: Are the authors only using the adversarial loss on samples x_adv for classification objective and No regular CE loss on the input samples "x"? Or are both losses turned on and the AT-CE loss is just an auxiliary loss to a regular CE classification loss on input samples "x"?
- Unclear why OOD data is required in Line 328. Is it only for Eval? Or is it used during training too (which RATIO method requires)?

Major:
- The *primary* focus of the paper is unclear. Is the focus on improving the discriminative-generative performance tradeoffs? Is the focus on improving the training stability of JEMs mainly? Is the focus on improving the adversarial robustness or OOD detection? It is quite confusing despite novel insights presented in the paper.
- Continuing with the prev major weakness, the results in Table 1 are not clear to any extent where the method presented excels in relation to the other methods. For example, for CIFAR-100 hybrid models the DAT method is worse both in classification accuracy as well as FID than EGC method (sure, there is an extra benefit of Adversarial Robustness -- which comes back to the question of what aspect being the primary focus of the paper). Please BOLD the numbers of your method that you think are excelling at a certain aspect in relation to the rest of the models.
- The datasets presented CIFAR-10/100 are not sufficient in convincing the readers of the merits of the work. These low res datasets are suited for a quick PoC run. The only substantial dataset used is ImageNet. Please consider including other datasets that compare to ImageNet in res like the LSUN dataset and/or CelebA faces etc. I think the results are insufficient to make a clear conclusion at the current form of the paper.
- Line 195 - "Preventing numerical overflow/ underflow" -- Please either cite or show a toy example as evidence that this indeed happens while training. You can follow the plots used in this paper (https://openaccess.thecvf.com/content/WACV2022/papers/Bhaskara_GraN-GAN_Piecewise_Gradient_Normalization_for_Generative_Adversarial_Networks_WACV_2022_paper.pdf) where to convince that the gradient explodes, an explicit plot of the gradient norm is presented in Fig 3.
- Line 199 "The grad formulation stabilizes training at the cost of limiting the EBM to modeling the support of p_data...." -- This is not clear to the reader how the cost here is limiting the EBM to modeling the support but not the full dentsity. Please include a citation or a toy experiment to prove this is true.
- Line 215 - The paper mentions how the authors' method impoves the stability of training, however, no experiment is presented to back up this claim. For example, out of a random 10 experiments for each model, what fraction destabilize and diverge during training. Is the new method better in this quantitatively. See Fig 2 of (https://openaccess.thecvf.com/content/WACV2022/papers/Bhaskara_GraN-GAN_Piecewise_Gradient_Normalization_for_Generative_Adversarial_Networks_WACV_2022_paper.pdf) where a large FID/KID score implies training instability out of 5 random runs.

**Questions:**

Please see weaknesses.

Additional Questions:
1. - Line 319-320 - The authors say they use the RATIO pretrained models. However, they also show in Eq 13 how RATIO objective is different from their AT-CE objective. This totally changes their method fundamentally for CIFAR-10 and ImageNet models since the pretraining objective is RATIO objective which they elucidate how it's different from theirs in Eq 13. Since the datasets used in the paper are no where close to being considered large, I highly recommend the authors not use any pretrained models and train them from scratch using their proposed formulation without departure to other objectives like RATIO that makes it quite confusing to the reader on the exact proposal for training in this paper.
2. - The weighted loss function modification in Eq (7) introduced by the authors is similar to the Focal Loss (https://arxiv.org/abs/1708.02002) albeit with gamma=1. Please compare this paper & suggest the readers why such a specific weight form is chosen (is it empirical? or is there a theoretical argument?)

Please also see weaknesses.

---

> ### Author Response · Authors · 2025-11-21
>
> We thank Reviewer WmTW for their detailed feedback. We have substantially strengthened our results by retraining the ImageNet models for 256×256 generation (previously 224×224) and including new results with ConvNeXt-Large, achieving FID 3.29 which exceeds ADM/LDM and matches the state-of-the-art autoregressive model VAR-d16 (FID 3.30). Please see Table 2 in the updated manuscript for full results. Below we address your specific concerns.
>
> **Abbreviations not expanded before use**
>
> > Numerous abbreviations are used before first expanding them. Example: PGD/SGLD in Abstract. PGD in the main paper before it's used in Line 63-65. "DAT" is used on line 249 without expanding it first.
>
> Thank you for pointing this out. We have expanded all abbreviations at their first use: SGLD (Stochastic Gradient Langevin Dynamics) and PGD (Projected Gradient Descent) in the abstract, and PGD again at first use in the main text. DAT (Dual Adversarial Training) is introduced in Section 1 before subsequent uses.
>
> **Missing Citations**
>
> > Line 39 - "...rarely excelling at both simultaneously." - Please cite?
> > Line 40 - "...but may underperform on downstream classification tasks." - Please cite?
>
> Thank you for pointing this out. We have added appropriate citations to both statements in the updated manuscript.
>
> **Misleading claim about datasets of increasing complexity**
>
> > Misleading to say on Line 76: "datasets of increasing complexity from CIFAR-10 to ImageNet..." The paper only includes CIFAR-100 additionally and that is not a spectrum of datasets of increasing complexity as claimed.
>
> Thank you for this feedback. We have revised the text to simply state "Experiments on CIFAR-10, CIFAR-100, and ImageNet demonstrate that our approach scales effectively" without the misleading claim of a spectrum.
>
> **Section 3.1: Define Z(θ)**
>
> > Please define Z(\theta) before or after using it in the equations. Also, please make it clear and explicit that the Z(\theta) used in P(x) on line 161 is not the same as the Z(\theta) used in Eq. (2). Explicitly define them in each case with an integral etc to show the margnialized partition function.
>
> We have added an explicit definition: $Z(\theta) = \sum\_{y\'} \int \exp(f\_{\theta}(x\')[y\']) dx'$ after Eq. 2 in the updated manuscript. We note that the same $Z(\theta)$ is used for both the joint distribution $p\_\theta(x,y)$ and the marginal distribution $p\_\theta(x)$—this is correct because the marginal is obtained by summing over $y$ in the numerator while the partition function remains unchanged.
>
> **Section 3.1: How Z(θ) is handled in the loss function**
>
> > Section 3.1, please also talk about how Z(\theta) is handled in the loss function and how it is estimated.
>
> In standard EBM training, the partition function $Z(\theta)$ is handled implicitly through the gradient formulation. Since $\nabla\_\theta \log Z(\theta) = \mathbb{E}\_{x \sim p\_\theta(x)}[-\nabla\_\theta E\_\theta(x)]$ (a well-known result in the EBM literature, so we omit the derivation; see, e.g., Chapter 18 of Goodfellow et al., 2016), the gradient of the log-likelihood $\nabla\_\theta \mathbb{E}\_{x \sim p\_{\text{data}}} [\log p\_\theta(x)]$ naturally decomposes into two terms (Eq. 5): one involving data samples (positive phase) and one involving model samples (negative phase). This is why SGLD sampling is required—to approximate the expectation over $p\_\theta(x)$. We have added a citation to Eq. 5 to make this clearer.
>
> **Reference:** Goodfellow, I., Bengio, Y., & Courville, A. (2016). Deep Learning. MIT Press.
>
> **Clarify Equation 11: AT-CE loss formulation**
>
> > Please clarify Equation 11: Are the authors only using the adversarial loss on samples x_adv for classification objective and No regular CE loss on the input samples "x"? Or are both losses turned on and the AT-CE loss is just an auxiliary loss to a regular CE classification loss on input samples "x"?
>
> We only use the adversarial loss on $x\_{adv}$, with no separate CE loss on clean inputs $x$. This is standard adversarial training as introduced by Madry et al. (2017)—the model is trained exclusively on worst-case perturbed inputs within the $\epsilon$-ball.

---

> ### Author Response · Authors · 2025-11-21
>
> **Primary focus of the paper unclear**
>
> > The primary focus of the paper is unclear. Is the focus on improving the discriminative-generative performance tradeoffs? Is the focus on improving the training stability of JEMs mainly? Is the focus on improving the adversarial robustness or OOD detection? It is quite confusing despite novel insights presented in the paper.
>
> Thank you for this feedback. We have revised the abstract, introduction, and results section to make this clear. Our primary focus is demonstrating that **hybrid models are no longer a compromise in either generative quality or classification performance**. We present the first hybrid model to simultaneously achieve:
>
> 1. **State-of-the-art generative quality**: FID 3.29 on ImageNet 256×256, matching the recent autoregressive model VAR-d16 (FID 3.30)
> 2. **Strong adversarial robustness**: Comparable to standard AT
>
> This addresses fundamental limitations of prior hybrids: EBM-based approaches (IGEBM, JEM, SADA-JEM) cannot scale beyond low resolution and achieve competitive generative performance on ImageNet-level datasets, while other scalable hybrids like the diffusion-based EGC do not reach state-of-the-art generative quality and lack adversarial robustness.
>
> Our work advances hybrid models in three ways: (1) we present the first EBM-based hybrid that scales to high-resolution complex datasets; (2) we uniquely combine generative quality with adversarial robustness, enabling critical applications like robust counterfactual explanations; and (3) our approach functions as a competitive standalone generative model with unique versatility. These results demonstrate that hybrid models are no longer a compromise—they can match specialized models in both dimensions while providing unique capabilities neither can provide alone.
>
>
> **Table 1 results unclear**
>
> > Continuing with the prev major weakness, the results in Table 1 are not clear to any extent where the method presented excels in relation to the other methods. For example, for CIFAR-100 hybrid models the DAT method is worse both in classification accuracy as well as FID than EGC method (sure, there is an extra benefit of Adversarial Robustness -- which comes back to the question of what aspect being the primary focus of the paper). Please BOLD the numbers of your method that you think are excelling at a certain aspect in relation to the rest of the models.
>
> We have highlighted both tables. In Table 2 (ImageNet 256×256), we highlight our best result—the ConvNeXt-L model achieving FID 3.29, matching VAR-d16 while maintaining strong adversarial robustness. In Table 1 (CIFAR-10/100), we have bolded the robust accuracy and FID values for our DAT models, highlighting our unique combination: strong adversarial robustness with substantially better generative quality (FID 7.57-9.96 vs. 23-28 for AT). DAT excels at the combination of capabilities rather than individual metrics. We clarify this across three dimensions:
>
> *1. First EBM-based hybrid that scales to high-resolution complex datasets:* Prior EBM-based hybrid approaches (IGEBM, JEM, SADA-JEM) could not scale beyond low resolution or achieve competitive generative performance on ImageNet-level datasets due to SGLD instability. Our approach is the first to overcome these limitations, achieving competitive generative quality and strong classification performance on ImageNet 256×256, demonstrating that EBM-based hybrid models can scale to complex, high-resolution datasets.
>
> *2. Generative quality and adversarial robustness for critical applications:* While other scalable hybrids like the diffusion-based EGC achieve reasonable performance, they do not reach state-of-the-art generative quality and lack adversarial robustness. On ImageNet, EGC achieves 13.56% robust accuracy compared to our 56.40% (ConvNeXt-L), while we also achieve better FID (3.29 vs 6.05). (Robustness data for EGC on CIFAR-10/100 is unavailable due to lack of public checkpoints, but ImageNet results provide sufficient evidence of limited robustness.) Our dual capability enables robust counterfactual explanations that are substantially more faithful than non-robust or robustness-only methods (Section 4.3.2), essential for interpretable ML in high-stakes domains (medical diagnosis, legal/financial decisions).
>
> *3. Flexibility as a competitive standalone generative model:* As a standalone generative model, our ConvNeXt-L matches the state-of-the-art autoregressive model VAR-d16 and surpasses leading diffusion models (ADM, LDM) in generative quality on ImageNet 256×256. Beyond this competitive quality, our energy-based approach offers unique versatility for diverse synthesis tasks (inpainting, super-resolution, image manipulation) and compositional generation (Du et al., 2020)—capabilities that emerge more naturally from energy-based models than GANs, diffusion, or autoregressive models.
>
> **Reference:** Du et al. (2020). Compositional Visual Generation with Energy Based Models. NeurIPS.

---

> > ### Author Response · Authors · 2025-11-21
> >
> > **OOD data usage unclear**
> >
> > > Unclear why OOD data is required in Line 328. Is it only for Eval? Or is it used during training too (which RATIO method requires)?
> >
> > The requirement for OOD data is clarified in Section 3.2, paragraph 2 ("In addition to the above gradient reformulation..."), where we explain that OOD samples serve as the source distribution for PGD-generated contrastive samples, which are then used to estimate $\mathbb{E}\_{x \sim p\_{\theta}(x)}[\cdot]$ in the BCE gradient. OOD data is used for both training and evaluation: during Stage 2 training for contrastive sample generation (Eq. 9), and during evaluation as initialization for conditional generation via PGD (see Section "Generative performance evaluation"). We use the 80 Million Tiny Images dataset for CIFAR-10/100 and construct an OOD dataset from Open Images for ImageNet (details in Section "Training setup").
> >
> > **Insufficient datasets**
> >
> > > The datasets presented CIFAR-10/100 are not sufficient in convincing the readers of the merits of the work. These low res datasets are suited for a quick PoC run. The only substantial dataset used is ImageNet. Please consider including other datasets that compare to ImageNet in res like the LSUN dataset and/or CelebA faces etc. I think the results are insufficient to make a clear conclusion at the current form of the paper.
> >
> > We thank the reviewer for this suggestion. CIFAR-10/100 are essential for comparing with prior hybrid models (JEM, SADA-JEM, EGC, Joint-Diffusion, RATIO), which all report results on these datasets. For high-resolution conditional generation, ImageNet is the standard benchmark: BigGAN, ADM, and VAR validate exclusively on ImageNet. Our ImageNet experiments are also comprehensive, demonstrating the robustness and generalizability of our approach:
> >
> > - 3 architectures: ResNet-50, WRN-50-4, ConvNeXt-Large
> > - State-of-the-art results: FID 3.29 matching VAR-d16
> > - Multiple ablation studies: Calibration, OOD detection, Effect of PGD sampling steps, effect of OOD dataset size, multiple random seeds demonstrating result stability
> >
> > Regarding LSUN and CelebA: to our knowledge, these are unconditional generation benchmarks without class labels, which does not fit our conditional JEM framework (e.g., they are used as unconditional benchmarks in LDM and EGC).
> >
> >
> > **Line 195: Evidence for numerical overflow/underflow**
> >
> > > Line 195 - "Preventing numerical overflow/ underflow" -- Please either cite or show a toy example as evidence that this indeed happens while training. You can follow the plots used in this paper (https://openaccess.thecvf.com/content/WACV2022/papers/Bhaskara_GraN-GAN_Piecewise_Gradient_Normalization_for_Generative_Adversarial_Networks_WACV_2022_paper.pdf) where to convince that the gradient explodes, an explicit plot of the gradient norm is presented in Fig 3.
> >
> > We appreciate the reviewer's suggestion for toy experiments; however, EBM instability tends to manifest primarily in high-dimensional settings, which may limit the informativeness of such experiments. Instead, we provide evidence from prior work.
> >
> > According to Grathwohl et al. (2019), the standard EBM gradient estimator (Eq. 5) is prone to instability because samples with high energy produce gradients that are orders of magnitude larger than typical, which causes divergence. In addition, Yin et al. (2022) showed that replacing the BCE gradient with the standard EBM gradient (while keeping PGD sampling) causes the energy difference between real data samples and contrastive samples to overflow even with very small learning rates. Since the only difference is the scaling factors $\alpha(x)$ and $\beta(x)$, this demonstrates that stability comes from their gradient attenuation effect. Our high training stability across extensive experiments (see our response to your stability diagnostics question) provides additional empirical evidence.

---

> ### Author Response · Authors · 2025-11-21
>
> **Line 199: Cost of limiting EBM to modeling support**
>
> > Line 199 "The grad formulation stabilizes training at the cost of limiting the EBM to modeling the support of p_data...." -- This is not clear to the reader how the cost here is limiting the EBM to modeling the support but not the full dentsity. Please include a citation or a toy experiment to prove this is true.
>
> Proposition 1 of Yin et al. (2022) proves that the BCE-with-PGD objective learns the support of the data distribution rather than the full density. To extend this result to the JEM (conditional modeling) setup, we have added a formal characterization in Appendix Section "Formal characterization of the learned distribution", where we derive the optimal class logits under our joint objective. In summary, our approach learns a hierarchical energy structure via the joint energy $E\_\theta^*(x,y)$:
> - Valid pairs $(x \in \mathrm{Supp}, y=y\_{\text{true}})$: $E\_\theta^*(x,y) = 0$
> - On-support, incorrect labels $(x \in \mathrm{Supp}, y \neq y\_{\text{true}})$: $E\_\theta^*(x,y) = \infty$
> - Off-support (OOD) $(x \notin \mathrm{Supp}, \text{any } y)$: $E\_\theta^*(x,y) \geq 0$
>
> This hierarchical structure enables the energy function $E\_\theta(x,y)$ to simultaneously perform robust classification (via $\arg\min_y E\_\theta(x,y)$), generation (via minimizing over $x$), and OOD detection (via thresholding). The marginal energy is constant ($E\_\theta^*(x) = 0$) on the support, meaning the model learns a uniform distribution over the support rather than the true density $p\_{\text{data}}(x)$. While this limits density estimation within the support, it provides superior training stability and effective OOD detection. We also note that the theoretical analysis assumes convergent PGD, while in practice finite-step PGD exhibits sampling-like behavior, exploring different regions of the energy landscape and leading to diverse, high-quality generation.
>
>
>
>
> **Line 215: No experiment to back up stability claim**
>
> > Line 215 - The paper mentions how the authors' method impoves the stability of training, however, no experiment is presented to back up this claim. For example, out of a random 10 experiments for each model, what fraction destabilize and diverge during training. Is the new method better in this quantitatively. See Fig 2 of (https://openaccess.thecvf.com/content/WACV2022/papers/Bhaskara_GraN-GAN_Piecewise_Gradient_Normalization_for_Generative_Adversarial_Networks_WACV_2022_paper.pdf) where a large FID/KID score implies training instability out of 5 random runs.
>
> We have added comprehensive stability diagnostics in the updated manuscript.
>
> *Divergence/failure rates:* On CIFAR-10 and ImageNet (ResNet-50, WRN-50-4, and ConvNeXt-L with various sampling steps $T$ and random seeds), we observed **zero divergences** using the hyperparameters specified in Appendix Section "Training details". On CIFAR-100, in our preliminary experiments with learning rate 0.01, we occasionally observed divergence (NaN output from the model). To address the reviewer's question about stability, we conducted additional experiments with a reduced learning rate of 0.009, which completely eliminated divergences across 5 random seeds (we have updated Appendix Section "Training details" to reflect this improved LR for CIFAR-100). Appendix Section "Variability of DAT performance across datasets" reports mean and standard deviation across 5 random seeds for CIFAR-10, CIFAR-100 (with LR 0.009), and ImageNet 256×256 (zero divergences across these 15 runs). This contrasts with SGLD-based JEM, which suffers from multiple stability issues (see discussion in Grathwohl et al., 2019).
>
> Additionally, we provide: (1) **training curves** (Figure "Training curves from Stage 2 joint training...") showing smooth FID improvements without divergence across all datasets, and (2) **gradient norm statistics** (Figure "$R\_1$ gradient norm during Stage 2 joint training...") showing that adversarial training maintains bounded gradients while standard training exhibits gradient explosion.

---

> > ### Author Response · Authors · 2025-11-21
> >
> > **RATIO pretrained models vs proposed AT-CE objective**
> >
> > > Line 319-320 - The authors say they use the RATIO pretrained models. However, they also show in Eq 13 how RATIO objective is different from their AT-CE objective. This totally changes their method fundamentally for CIFAR-10 and ImageNet models since the pretraining objective is RATIO objective which they elucidate how it's different from theirs in Eq 13. Since the datasets used in the paper are no where close to being considered large, I highly recommend the authors not use any pretrained models and train them from scratch using their proposed formulation without departure to other objectives like RATIO that makes it quite confusing to the reader on the exact proposal for training in this paper.
> >
> > We apologize for the confusion. We do not use models trained with the RATIO objective. The RATIO codebase provides standard adversarial training (AT) checkpoints, and we use these standard AT checkpoints for Stage 1. Specifically:
> > - CIFAR-10: Standard AT checkpoint from the RATIO codebase
> > - CIFAR-100: Trained our own standard AT model
> > - ImageNet: Standard AT checkpoints from Salman et al. (2020) and Singh et al. (2023)
> >
> > We have revised the text in Section "Training setup" to clarify that we use "a standard AT checkpoint from the RATIO codebase" rather than "a pretrained CIFAR-10 model from RATIO" to avoid this confusion.
> >
> > Regarding training from scratch: since Stage 1 is equivalent to standard adversarial training, using pretrained standard AT checkpoints is fully consistent with our proposed formulation—there is no departure to other objectives. Leveraging pretrained models is a practical advantage of our two-stage approach, as it demonstrates that our method can improve existing robust classifiers without retraining from scratch. We note that we do train CIFAR-100 from scratch, showing our method works in both scenarios.
> >
> >
> >
> > **Weighted loss function similar to Focal Loss**
> >
> > > The weighted loss function modification in Eq (7) introduced by the authors is similar to the Focal Loss (https://arxiv.org/abs/1708.02002) albeit with gamma=1. Please compare this paper & suggest the readers why such a specific weight form is chosen (is it empirical? or is there a theoretical argument?)
> >
> > We appreciate the reviewer's observation regarding the similarity to Focal Loss. While the scaling factors in Eq. 7 share a similar form with Focal Loss ($\gamma=1$), they arise from different motivations. Focal Loss introduces $(1-p\_t)^\gamma$ as a design choice applied directly to the loss function, with $\gamma$ as a tunable hyperparameter to down-weight easy examples for class imbalance. In contrast, our scaling factors appear in the gradient formulation, not in the loss itself, and are not a design choice but arise directly from taking the derivative of the standard BCE loss (Eq. 9). When we compute $\nabla\_\theta \mathcal{L}\_{\text{BCE}}$, the terms $\alpha(x) = 1 - \sigma(-E\_\theta(x))$ and $\beta(x) = \sigma(-E\_\theta(x))$ naturally appear as part of the gradient computation—this is a mathematical consequence of the BCE formulation, not an empirical or heuristic modification. The standard BCE loss has a principled interpretation for EBM learning: it trains a binary classifier to distinguish real data from generated samples, providing a theoretically grounded approach to learning the data support (Yin et al., 2022). We have added a note after Eq. 9 clarifying that $\nabla\_\theta \mathcal{L}\_{\text{BCE}}$ equals the right-hand side of Eq. 7.

---

### Author Response · Authors · 2025-11-21
**Manuscript Updates in Response to Reviewer Feedback**

We sincerely thank all reviewers for their thoughtful and constructive feedback. We have substantially revised our manuscript to address your concerns. Below we highlight the major updates:

**Empirical Results and Baselines**

- **New ImageNet 256×256 results with ConvNeXt-Large:** We have retrained our ImageNet models at 256×256 resolution (previously 224×224) and added results with ConvNeXt-Large, achieving FID 3.29—exceeding ADM/LDM and matching the state-of-the-art autoregressive model VAR-d16 (FID 3.30) while maintaining strong adversarial robustness. The abstract, introduction, Section 3.5 (Two-stage training), and all ImageNet tables have been updated to reflect these results.

- **Expanded related work (feF3, Y3JD):** Section 2 and Appendix A.1 now include enhanced comparison with AT-EBM, CEM, JEM++, Robust-JEM, and discussions of additional work connecting adversarial robustness and EBMs (Zhu et al., 2021; Mirza et al., 2024). Table 1 adds 5 additional baseline comparisons on CIFAR-10 (JEM++, JEAT, Robust-JEM, VERA, WEAT). We now more clearly differentiate our approach from prior SGLD-based methods and AT-EBM.

**Theoretical Analysis and Formalization**

- **Formal characterization of the learned distribution (WmTW, feF3):** Appendix A.15, new. We provide a formal analysis with Proposition A.1 proving that our BCE-with-PGD objective learns optimal class logits $f\_\theta^*(x)[y] = \log p\_{\text{data}}(y|x)$ on support with constant marginal energy $E\_\theta^\*(x) = 0$. This creates a unified compatibility function where minimizing $E\_\theta(x,y)$ simultaneously performs robust classification, generation, and OOD detection.

- **Empirical validation of implicit $R_1$ regularization (WmTW, feF3):** Appendix A.13, new. Figure 10 shows $R_1$ gradient norms during training, demonstrating that adversarial training maintains bounded gradients while standard training exhibits gradient explosion, validating our theoretical analysis.

**Additional Empirical Analysis**

- **Analysis of discriminative-generative trade-off (Y3JD):** Appendix A.18, new. We provide empirical analysis across multiple dimensions (PGD steps, loss weights, augmentation strategies) revealing an inherent trade-off between generative and discriminative performance, with insights into the underlying mechanism and practical tuning mechanisms.

- **Robustness to common corruptions (Y3JD):** Appendix A.16, new. We evaluate on CIFAR-10-C, achieving mean corruption error comparable to standard AT (19.84% vs 19.63%). See Table 18.

- **$L\_\infty$ training (feF3):** Appendix A.17, new. We completed $L\_\infty$ adversarial training on ImageNet 256×256 with ConvNeXt-L, achieving FID 4.11 and IS 320.7 while maintaining competitive robustness—demonstrating successful generalization to the more commonly used $L\_\infty$ norm. See Table 17.

**Stability, Reproducibility, and Computational Efficiency**

- **Training stability diagnostics (WmTW, feF3, Y3JD):** Appendix A.11-A.13, updated/new. We report zero divergences across 15 independent runs (5 seeds × 3 datasets), with mean and standard deviation demonstrating reproducibility. Figure 9 shows smooth training curves for all datasets. Additionally, Figure 10 shows $R_1$ gradient norms during training, demonstrating that adversarial training maintains bounded gradients while standard training exhibits gradient explosion.

- **Computational cost analysis (Y3JD, MpyS):** Appendix A.14, new. We provide detailed efficiency metrics showing modest training overhead w.r.t. standard AT (1.41-1.56× for CIFAR, 1.05-1.36× for ImageNet) with actual training times of 2.4-20 hours for ImageNet on AMD Instinct accelerators. Inference is ~29× faster than ADM-G and ~5× faster than LDM-4-G. See Table 17 and Figure 11.

- **Code and reproducibility:** Complete code for ImageNet 256×256 training/evaluation with ConvNeXt-Large has been provided as supplementary material to ensure full reproducibility of our results.

**Writing and Presentation**

- **Revised abstract, introduction, and results section to clarify our primary contribution and its significance (WmTW, MpyS):** We have substantially revised the abstract, introduction, and results section to clarify our primary contribution: (1) first EBM-based hybrid to scale to high-resolution datasets with high training stability, simultaneously achieving state-of-the-art discriminative and generative performance; (2) uniquely combines generative quality with adversarial robustness for critical applications; and (3) functions as a competitive standalone generative model, achieving significantly higher throughput than diffusion models while offering unique versatility.

- **Notation and references (WmTW, Y3JD):** We have added missing citations and improved clarity in mathematical formulations.

---

We believe these updates substantially strengthen the paper and address the reviewers' concerns. We remain committed to further revisions based on additional feedback.

---

> ### Author Response · Authors · 2025-11-24
> **Additional Update: Wall-clock Training Times**
>
> *Note: We have revised these training times to report effective training duration (excluding FID/accuracy evaluation overhead), which provides a more accurate measure of the actual training cost.*
>
> We have added effective wall-clock training times (excluding FID/accuracy evaluation) to Table 17 (Appendix Section "Computational cost analysis") measured on AMD Instinct accelerators:
>
> **CIFAR-10/100 (Stage 2 only):**
> - CIFAR-10, T=40: 10 hours (4×MI210)
> - CIFAR-10, T=50: 10 hours (4×MI210)
> - CIFAR-100, T=45: 12 hours (4×MI210)
> - CIFAR-100, T=50: 12 hours (4×MI210)
>
> **ImageNet 256×256 (Stage 2 only):**
> - ResNet-50, T=15: 2.4 hours (4×MI210)
> - ResNet-50, T=30: 3.8 hours (4×MI210)
> - WRN-50-4, T=30: 4.7 hours (4×MI250)
> - WRN-50-4, T=65: 8.2 hours (4×MI250)
> - ConvNeXt-L, T=110: 20 hours (8×MI300)
>
> **Baseline models (converted to 8×MI300 equivalent, from Rombach et al., 2022):**
> - BigGAN-deep: 52-104 hours (from 128-256 V100-days)
> - ADM-G: 390 hours (from 962 V100-days)
> - LDM-4-G: 110 hours (from 271 V100-days)
>
> These times represent effective training duration (excluding evaluation overhead) for Stage 2 training (our models) or total training time (baselines). Our ConvNeXt-L Stage 2 training (20 hours) is \~5 faster than LDM-4-G (110 hours). Baseline conversions use MI300/V100 = 7.40× based on benchmark performance ratios.

---

### Meta-Review · Area_Chair_bMxn · 2026-01-06

**Summary:**

This paper introduces a dual adversarial training framework to address stability issues in joint discriminative-generative energy-based models. The approach replaces Stochastic Gradient Langevin Dynamics (SGLD) with adversarial training principles. The paper shows both robust classification and high-fidelity generation.

**Reviewer Concerns:**

The reviewer raised the following concerns initially.

1. Technical contribution (feF3, MpyS): The reviewer believes that the conceptual novelty is limited, regarding the proposed approach as a known adversarial training and energy-based model techniques. In addition, the individual components also borrowed from established ones (BCE loss, projected gradient descent)
2. Stability (feF3, WmTW): The reviewer requested stability analysis beyond final metrics (using gradient norms and failure rates)
3. Additional comparisons (Y3JD, feF3): The reviewer mentioned that the outdated baselines
4. Mathematical Formalization (feF3, WmTW): The reviewer requested clarification about the distribution, and another reviewer questioned the partition function
5. Computational cost (Y3JD, MpyS): The reviewers requested training/inference overhead of the proposed approach.
6. Non L2 norms (feF3): the reviewer requested the use of L2 norms
7. Dataset: the initial submission has experiments using smaller datasets (CIFAR10), which questions the possibility of extending the idea to larger datasets.

**Reviewer Scores:**

Initial reviews are leaning towards rejection 2,4,4,6 due to incremental contribution, insufficient stability analysis, lack of high-resolution results, and missing comparison to modern approaches.

However, AC noted that the authors provided an extensive rebuttal that answers most of the major concerns raised by reviewers. To summarize how the authors provided feedback, for 7, they provided an ImageNet 256x256 experiment showing compelling FIDs. For 6 and 2, the authors provided an extensive comparison and an ablation study. For 3, authors added experiments that compare the proposed approach with AT-EBM, CEM, JEM++, and Robust-JEM. For 4, the authors added a formal characterization and a proposition showing that the BCE-with-PCD objective learns optimal class logits. For 5, the authors formally explain the computational overhead, showing 1.05~1.36x overhead. Regarding 1, AC agrees with reviewer feF3 and MpyS's comments stating that the proposed approach is built upon a well-established idea. However, AC notes that successful demonstrations and propositions are valuable, so the technical contribution should not be the sole criterion for acceptance.

AC confirms that a significant update has been made in the revision after properly requesting clarifications. Therefore, even though the initial score was negative, AC voted for accepting this paper. However, AC's decision confidence is not firm because the revision included extensive updates that may require a full review by the original reviewers, even though AC carefully re-revisited the paper.

---

### Decision · Program_Chairs · 2026-01-26

Accept (Poster)